# A topological mechanism for robust and efficient global oscillations in biological networks

Chongbin Zheng [1,2] & Evelyn Tang [1,2] ✉

Long and stable timescales are often observed in complex biochemical networks, such as in emergent oscillations. How these robust dynamics persist remains unclear, given the many stochastic reactions and shorter time scales demonstrated by underlying components. We propose a topological model that produces long oscillations around the network boundary, reducing the system dynamics to a lower-dimensional current in a robust manner. Using this to model KaiC, which regulates the circadian rhythm in cyanobacteria, we compare the coherence of oscillations to that in other KaiC models. Our topological model localizes currents on the system edge, with an efficient regime of simultaneously increased precision and decreased cost. Further, we introduce a new predictor of coherence from the analysis of spectral gaps, and show that our model saturates a global thermodynamic bound. Our work presents a new mechanism and parsimonious description for robust emergent oscillations in complex biological networks.

The reduction of the full system response to a much lower dimensional description has been observed in many complex biological systems, where the system dynamics or behavior reduces to a much smaller phase space[1–3]. However, we still lack good models that can mechanistically account for this dimensionality reduction, or that remain stable under noise or structural heterogeneity. This is exemplified in computational models of memory, that describe specific attractor states which represent persistent memories[4,5]. However, attractors tend to drift or lose accuracy with noise, and it remains an area of open research on how to retain encoded information in these models[4]. Another example is that of long oscillations, such as the circadian rhythm, which are crucial for the regulation of many processes such as metabolism and replication[6,7]. Previously proposed models, such as feedback loops of chemical reactions, typically involve either small reaction networks consisting of reactions on a similar timescale as the oscillation itself or a large number of system-dependent parameters[8–10]. Simple conditions are still lacking for explaining how such oscillations with their long timescales can emerge from a large phase space of faster chemical reactions. Understanding the necessary and sufficient principles that govern robust oscillations is crucial as

disruptions in biological clocks lead to decreased health and reproductive fitness in multiple organisms[7,11–14].

As biological networks, including those mentioned above, typically have a large phase space of possible reactions, this renders unfeasible exhaustive searches using other approaches like experiments or numerical simulation[15], underscoring the need for simple conceptual methods to provide insight[16–18]. The development of rigorous theory would also shed light on simple design principles for targeted dynamics in synthetic biological systems[19,20] or in the engineering of reconfigurable materials, e.g., through dissipative self-assembly[21,22]. Yet, biology presents challenges for the development of suitable theory due to being stochastic, heterogeneous, and strongly non-equilibrium[15,23–25]. Hence, the few successful models that exist in biology are often heavily dependent on specific system parameters.

Towards dimensional reduction onto the network boundary, topological models have been proposed in stochastic systems[26–28]. These are a generalization of topological invariants studied in quantum systems[29–32], which show physical responses on the system edge or boundary. Powerfully, this response is insensitive to various types of disorder or noise[26,28,33]. It would be desirable to demonstrate how

[1]Center for Theoretical Biological Physics, Rice University, Houston, TX 77005, USA. [2]Department of Physics and Astronomy, Rice University, Houston, TX 77005, USA. ✉e-mail: e.tang@rice.edu

topology can be realized in a biological system, given the many attractive properties of topology, such as its robust response. However, it has not yet been shown to relate to a biological system. Previous works do not explicitly connect to experimentally tunable parameters or known molecular reactions, giving little guidance on how to measure or access the topological properties[26–28]. Our manuscript responds to this lack in the field, by providing detailed biophysical mechanisms towards experimental verification of topological invariants.

Here, we provide the first case study of a topological mechanism in a concrete biological system—that of the KaiABC system for the circadian rhythm of cyanobacteria[34,35]. The topological edge currents naturally reproduce the kinetic ordering of KaiC phosphorylation cycles, first proposed in ref. 28, where the biophysical mechanisms remained abstract. These missing mechanisms are made concrete in this work, where we demonstrate how the biochemical reactions[36–39] interact via the separation of timescales to produce the resulting edge currents. This provides the necessary and sufficient conditions for the observed oscillatory cycle. We analyze the parameter space to show how a variation of the transition rates (e.g., by changing ATP concentration[40] or using a mutant[41]) affects the coherence and dissipation of the oscillation, by tuning the system into and out of the topological transition. This yields key insights into an important regulatory system that had until now required rather complicated models, especially to reproduce the observed kinetic ordering of T and S phosphorylation[8–10].

On the theoretical level, we characterize the coherence of the resulting cycle, showing that it satisfies theoretical bounds[42] for the most coherent oscillator equivalent to that of a unicycle network—without the fine-tuning needed for a unicycle[42]. The topological model shows high coherence compared to other available models while producing the global day–night cycle with unusually few free parameters. In addition, we explore the coherence and energetic cost of the oscillation using tools from non-equilibrium stochastic thermodynamics, to reveal an efficient regime where coherence increases while cost simultaneously decreases. Lastly, a new indicator of oscillation coherence from spectral gaps in band theory is introduced, to study the saturation of this model on global thermodynamic bounds. Overall, this analysis can explain long-standing puzzles in biology, such as how dimensional reduction is achieved in a robust and flexible manner to produce emergent oscillations.

Our work provides an alternative mechanism for oscillations from the prevailing paradigm of Monod–Wyman–Changeux (MWC)[43]. This paradigm assumes cooperative all-or-none conformational changes for protomers in an oligomer upon ligand binding, which acts as a molecular switch that changes the affinity of all binding sites. The model has been useful in describing systems such as hemoglobin[43], ligand-gated ion channels[44], and bacterial chemotaxis[45]. Still, it remains unclear if the MWC model is the dominant mechanism for many other systems. Using the MWC view, models for KaiABC typically assume highly cooperative conformational changes for KaiC monomers in order to obtain oscillations[9,10,46,47]. However, new structural studies suggest that the positive cooperativity between monomer conformational states is fairly weak[48]. Hence, it is timely to examine alternative models that can generate emergent oscillations for macromolecules in the presence of strong internal fluctuations or weak positive cooperativity. Overall, our work proposes a new pathway for the emergence of high coherence despite stochasticity and strong fluctuations distinct from the typical paradigm of strong cooperativity, that could be relevant for biological oscillations more generally.

## Results
### Topological model for emergent oscillations
We consider discrete stochastic processes that operate in a two-dimensional configuration space. The state of the system is completely specified by three variables $(x, y)_s$. The "external" variables $x$ and $y$ are independent dynamical variables. Based on the widespread presence of non-equilibrium cycles in biological systems[49–51], we propose that the "external" transitions modifying these variables form reaction cycles. The external processes have transition rates $\gamma_{ex}$ and slower reverse rates $\gamma'_{ex} \ll \gamma_{ex}$ (black solid arrows), as shown in Fig. 1a. In the KaiABC system, $x$ and $y$ represent the number of phosphorylated T- and S-sites, respectively. T and S are two residues on each monomer of the hexameric KaiC molecule[52], where T phosphorylation is denoted with orange spheres and S phosphorylation with pink spheres in Fig. 1a. Hence, phosphorylation occurs in the left and top arrows of the cycle in Fig. 1a, and dephosphorylation in the remaining right and bottom arrows—this four-state motif contains two futile cycles[49,50].

The "internal" state variable $s$, given by compass directions N–E–S–W, labels which of the four external transitions the system is primed for. For example, in the W state, the system is most likely to go through the westward external transition that decrements $x$. Within each phosphorylation level $(x, y)$, we model transitions between internal states in a cyclic manner, with transition rates $\gamma_{in}$ and slower reverse rates $\gamma'_{in} \ll \gamma_{in}$ (gray dashed arrows in Fig. 1b). In the KaiABC system, the vertical internal transitions stem from conformational changes of the A-loop[36,38], where the A-loop is denoted as a blue square when exposed and as a purple circle when buried (Fig. 1b). The top (N → W) and left (W → S) transitions are facilitated by KaiB binding and unbinding, respectively[53,54]. The bottom transition (S → E) is catalyzed by KaiA[55,56], represented by a curved arrow. As a single KaiA dimer binds and unbinds several times during each phosphorylation[57,58], it acts as a catalyst, consistent with the treatment of KaiA in previous models[9,59]. The intermediate KaiA-bound states are coarse-grained out of the model and not shown.

These reactions are repeated for each monomer and hence can be laid out as a lattice. As shown in Fig. 1c, the phosphorylation/dephosphorylation cycles in Fig. 1a (green box) and the internal cycles in Fig. 1b (blue box) both repeat along the $x$ and $y$ axes of T and S phosphorylation. Such a lattice will have edges representing the physical constraints of the system, i.e., $0 \le x \le N_x$ and $0 \le y \le N_y$. In our case, $N_x = N_y = 6$ since there are 6 sites available on a KaiC hexamer for each T and S phosphorylation. Note that our model only keeps track of the number of T and S phosphorylated monomers, while the specific location of each monomer does not matter.

Our model in Fig. 1c is mathematically equivalent to the model introduced in ref. 28 but given a novel interpretation based on a more realistic description of KaiABC. Specifically, in ref. 28, the external variables $x$ and $y$ represent the number of phosphorylated KaiC monomers and the number of monomer conformational changes. In our paper, these variables are interpreted as the number of T and S phosphorylated monomers. Rather than occurring independently, the conformational change is instead hypothesized to change the system's internal state. In addition, we include interactions with KaiA and KaiB molecules along with ADP/ATP turnover as reactions that change the internal state of the model.

Towards analyzing the model, we simplify our parametrization for the four transition rates $\gamma_{ex}, \gamma'_{ex}, \gamma_{in}, \gamma'_{in}$ down to three parameters. First, $\mu$ is the thermodynamic force defined by $e^{\mu/k_B T} \equiv \gamma_{ex}/\gamma'_{ex} = \gamma_{in}/\gamma'_{in}$. We analyze $\mu$ in units of $k_B T$. For an arbitrary cycle in the network, the sum of $\mu$ along each transition in the cycle is the energy input into that cycle from external driving such as ATP hydrolysis[60]. The system obeys detailed balance when the total $\mu$ along every cycle is 0 and is out of equilibrium otherwise[61]. Here we assume the same $\mu$ for every transition in the model, which removes one free parameter. Detailed balance then simply corresponds to $\mu = 0$. Second, $\rho$ varies the ratio of external to internal transitions, as defined by $e^\rho \equiv \gamma_{ex}/\gamma_{in} = \gamma'_{ex}/\gamma'_{in}$. It quantifies the separation of timescales between the external and internal transitions. Lastly, $\gamma_{tot}$ controls the overall timescale of all transitions, i.e., $\gamma_{tot} \equiv \gamma_{ex} + \gamma'_{ex} + \gamma_{in} + \gamma'_{in}$.

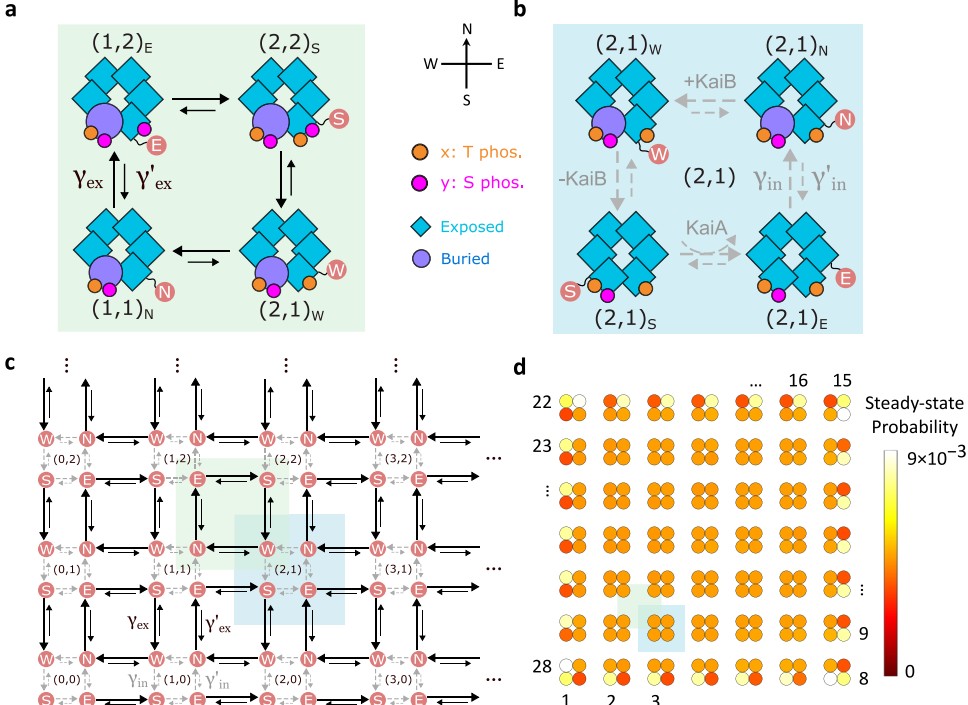

**Fig. 1 | Topological model for emergent oscillations, illustrated with KaiABC that regulates the circadian rhythm. a** Based on observations of autophosphorylation in the literature[51,57,59], it is thought that monomers undergo phosphorylation and dephosphorylation cycles (black arrows $\gamma_{ex}$ and slower reverse transitions $\gamma'_{ex}$). Two types of phosphorylation T and S are shown with the addition of orange and pink circles and the numbers of each are given in brackets. **b** Within a given phosphorylation level, internal transitions (gray arrows, $\gamma_{in}$ and $\gamma'_{in}$) take place due to conformational changes (illustrated by circles vs. squares) or interaction with proteins like KaiA or KaiB. The internal state (N, E, S, W) labels the direction of the next external transition that it primes. KaiA promotes phosphorylation catalytically[57] and hence is denoted with a curved arrow. **c** These cycles can be laid out in a lattice, with T phosphorylation along the horizontal axis and S phosphorylation along the vertical axis, where each set of four internal transitions (highlighted in blue) repeats along these axes. **d** This lattice allows probing of its topological properties. In the ordinary case with similar rates throughout, i.e. $\gamma_{ex} \sim \gamma_{in}$, the system will perform a random walk ergodically through the phase space. In the topological regime, when $\gamma_{ex} \gg \gamma_{in}$, once the system hits an edge, it will continue around the edge, as can be verified by inspection. We plot the steady state in the topological regime, which lies on the system edge, taking $\mu = 1, \rho = 2, \gamma_{tot} = 1$. There are 28 states along the edge with high probability, labeled by the order in which they are traversed in a typical trajectory. These trajectories form a global current along the edge of the state space.

There are two dynamical regimes for the system, determined by tuning the parameter $\rho$. In the trivial regime, when $\rho < 0$, the system tends to go through local counterclockwise cycles via internal transitions $\gamma_{in}$, interspersed with slower external transitions $\gamma_{ex}$ that break out of these cycles. Over a long time, the system displays diffusive dynamics and will ergodically explore the whole state space. In the topological regime, when $\rho > 0$, the system supports an edge state[28]. In the bulk of the state space, the system would similarly go through local clockwise cycles via external transitions $\gamma_{ex}$, interspersed with internal transitions $\gamma_{in}$, which are now slower. However, once the system reaches the edge, it will continue around the edge. This can be verified by inspection in Fig. 1c (also see Supplementary Movie 1). Over long times, the steady-state distribution will hence lie on the system edge, forming a global current that flows counterclockwise along the boundary of the lattice (see Fig. 1d). The edge state and the associated dynamical regime have a topological origin, as their emergence is governed by a topological invariant, the 2D Zak phase[28,62]. For our topological model, the 2D Zak phase takes the trivial value (0, 0) when $\rho < 0$ but becomes $(\pi, \pi)$ in the topological regime $\rho > 0$ (see Supplementary Section II for more details). As $\rho$ becomes larger in the topological regime, the Zak phase $(\pi, \pi)$ is preserved, while the edge state becomes more localized on the system boundary[63]. The edge state further inherits the useful property of topological protection from inaccessible states[28] or perturbations in transition rates (see Supplementary Figs. 1 and 2). Because of these unusual properties, we focus on the topological regime and investigate the properties of our system, assuming $\rho > 0$ from now on.

## Biophysical mechanisms for the topological model

In the topological regime, our model exhibits a separation of timescales ($\gamma_{ex} \gg \gamma_{in}$). This finds experimental support in the KaiABC system, where the faster phosphorylation/dephosphorylation reactions are primed by other slower processes. As shown in Fig. 2a, phosphorylation involves two main steps. The slow transition priming phosphorylation is the KaiA-induced ADP release and ATP binding[64]. This process occurs in the nucleotide-binding pocket on the CII domain[65] (gray oval on the upper lobe of KaiC in Fig. 2a). The fast phosphorylation reaction occurs only after the ADP/ATP exchange, as it transfers the phosphate group $P_i$ from ATP to the phosphorylation site[66,67]. Similarly, dephosphorylation also involves two main steps. As shown in Fig. 2b, the slow transition priming T dephosphorylation is KaiB binding[56,57,68] and ATP hydrolysis at CII[66,67]. With ADP at the CII domain, fast dephosphorylation can proceed by transferring $P_i$ back to ADP[66,67]. Such separation of timescales is essential for the emergence of global oscillations in phosphorylation level due to nontrivial topology.

Moreover, non-equilibrium driving powers key reactions in the KaiABC phosphorylation cycle. In particular, phosphorylation is powered by the conversion of ATP to ADP[66,67], a process that dissipates free energy in relevant experimental conditions. Indeed, experiments show that the ATP consumption rate increases by 75% during the phosphorylation phase compared to the average basal rate[69]. In addition, KaiB binding is powered by ATP hydrolysis in the CI domain (lower lobe of KaiC in Fig. 2), which leads to subsequent dephosphorylation[40]. CI ATP hydrolysis proceeds slowly but continuously throughout the

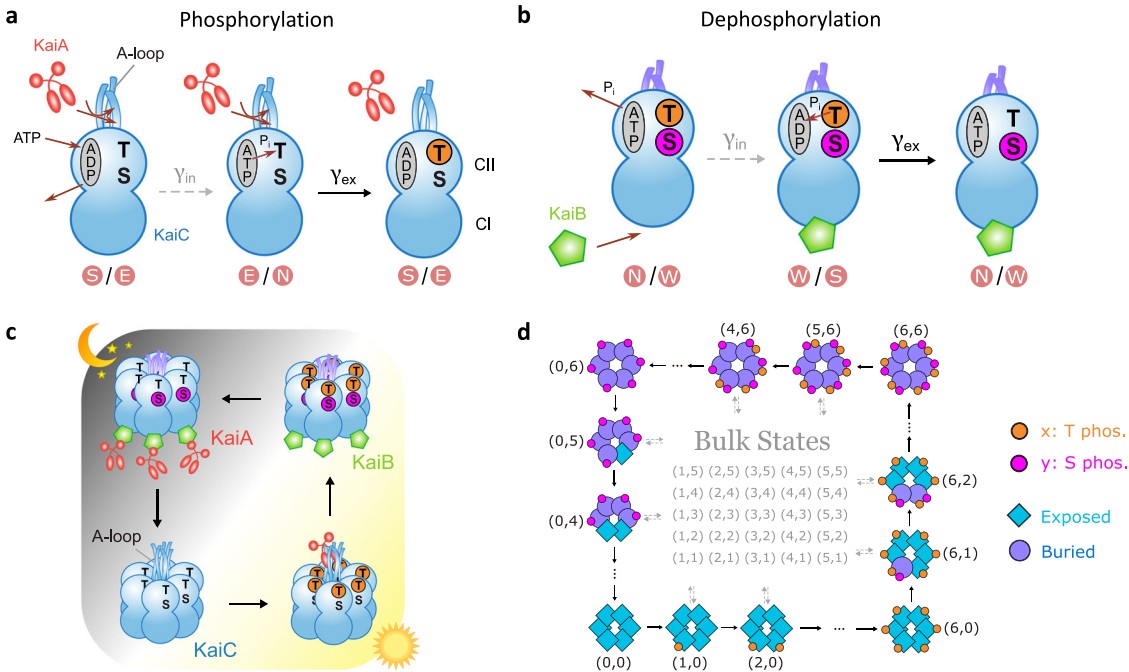

**Fig. 2 | Biophysical mechanisms for the topological model, which reproduces the KaiABC circadian rhythm. a** Phosphorylation of a monomer relies on two main steps. *Left*: Slow reaction $\gamma_{in}$ where KaiA promotes ADP release and ATP binding in the CII nucleotide-binding pocket (gray oval)[64,66,67], priming the monomer for phosphorylation. *Right*: Fast reaction $\gamma_{ex}$ where a phosphate group $P_i$ is transferred from ATP to the phosphorylation site[66,67]. **b** Dephosphorylation similarly has two main steps. *Left*: Slow reaction $\gamma_{in}$ where KaiB binds to KaiC[56,68] and ATP hydrolyzes to ADP at CII[66,67], priming the monomer for dephosphorylation. *Right*: Fast reaction $\gamma_{ex}$ where $P_i$ is transferred back to ADP[66,67]. In both (**a,b**), we illustrate what happens for T-sites; additional distinctions between T and S phosphorylation/dephosphorylation are discussed in the main text. **c** KaiABC exhibits

oscillations via a concerted global cycle of phosphorylation and dephosphorylation. During the day, all six KaiC monomers get phosphorylated at the T-sites, and then at the S-sites. Phosphorylation is promoted by interaction with KaiA molecules[55]. By night, phosphorylated KaiC binds to KaiB, which sequesters KaiA from the solution. In the absence of KaiA, all the T-sites get dephosphorylated, followed by the S-sites[56]. Since individual monomers can independently phosphorylate[51], it is unclear why they would perform a concerted phosphorylation cycle that is robust. **d** A possible solution lies in the topological phase of the model, in which a global cycle emerges that recapitulates the experimentally observed phosphorylation sequence.

phosphorylation cycle, at a rate of only ~10 ATP molecules per day[40,69]. Such consistent external driving is required for a global reaction cycle where each transition is biased toward the same direction and detailed balance is broken[60].

In Fig. 2a and b, we illustrate the case of T phosphorylation/dephosphorylation. The same ADP/ATP exchange and phosphotransfer mechanisms are used for S phosphorylation/dephosphorylation. Additionally, S phosphorylation is accompanied by A-loop burial while S dephosphorylation is accompanied by A-loop exposure in our model, consistent with experimental results[38]. The role of S-sites (rather than T-sites) in regulating KaiC conformational change is further highlighted in a structural study, where S phosphorylation/dephosphorylation drives conformational changes of a local phospho-switch between the CI and CII domain[70]. In particular, we assume that monomer conformational changes occur after S phosphorylation of the same monomer. It follows that the states $(x, y)_E$ and $(x, y)_S$ (for $y > 0$) have $y-1$ circles and $7-y$ squares, while the states $(x, y)_W$ and $(x, y)_N$ have $y$ circles and $6-y$ squares. Moreover, KaiB unbinding rather than KaiB binding tends to occur during S dephosphorylation[54]. We note this tendency using "-KaiB" in Fig. 1b. Note that the internal processes illustrated in Fig. 2a and b form reaction cycles as shown in Fig. 1b. This is consistent with experimental evidence that A-loop exposure facilitates KaiA catalysis[36] and inhibits KaiB binding[37], while A-loop burial facilitates KaiB binding and inhibits KaiA catalysis[37]. Such interactions between A-loop conformations and KaiA/KaiB, therefore, restrict the allowed internal transitions to a single cyclic pathway.

In experimental conditions that support oscillations, KaiC forms a stable hexameric structure[71]. Each monomer in the hexamer can

independently go through the biochemical reactions illustrated in Fig. 2a and b. Experimentally, it is well established that KaiC molecules exhibit 24-h cycles via a concerted phosphorylation sequence in the presence of KaiA and KaiB[35]. As illustrated in Fig. 2c, during the day, all six KaiC monomers get phosphorylated at the T-sites, and then at the S-sites. By night, all the T-sites get dephosphorylated, again followed by the S-sites[72]. The phosphorylation phase is facilitated by KaiA catalysis[55], while the dephosphorylation phase is promoted by KaiB binding[53]. The robustness of this concerted phosphorylation sequence is surprising since KaiC monomers within the hexamer can independently phosphorylate and change conformations[48]. Given this very large phase space of possible reactions available to individual monomers, why would all the monomers phosphorylate in a concerted global cycle? Further, since phosphorylation in both T and S are promoted during the day, why does phosphorylation proceed in the specific order where all six T-sites get phosphorylated before S phosphorylation begins (Fig. 2c)?

To answer this question, models[9,10,46,47] have typically relied on the cooperative or Monod-Wyman-Changeux paradigm[43] of allosteric regulation, which restricts the configuration space such that either all or none of the monomers in a complex undergo a conformational change. However, recent cryo-EM data shows that monomers can demonstrate independent conformational changes in the same hexamer[48], challenging this strong restriction. Further, these models often put in by hand the specific ordering of T phosphorylation occurring before S phosphorylation.

Our topological model presents an alternative way to account for the experimental facts and explain the emergence of the KaiC phosphorylation cycle. As previously discussed, in the topological regime

where $\rho \gg 0$, the system supports a propagating edge current. Figure 2d shows a coarse-grained picture of the edge current. One out of four internal states is shown for each phosphorylation level $(x, y)$ along the edge, specifically the last state along the edge (e.g. E states for the bottom edge or S states for the left edge). As we can see, the edge current is equivalent to a global cycle of concerted phosphorylation of the T-sites, followed by the S-sites, then dephosphorylation of the T-sites and, lastly, of the S-sites (also see Supplementary Movie 1). This provides a mechanism that allows for monomers to undergo conformational and other changes individually, while still producing a global cycle and the experimentally observed phosphorylation sequence that emerges with less fine-tuning.

## Model thermodynamics: Precision vs. cost

While this topological model provides a unique alternative mechanism to experimentally observed oscillatory dynamics, how precise or efficient are the oscillations produced? In this section, we quantify the thermodynamics and entropy production of this model and compare its performance to that of other KaiC models[42,73,74]. Further, we identify a new predictor for oscillator coherence and analyze the saturation of thermodynamic bounds for different models. We begin by analyzing the master equation that describes stochastic systems,

$$\frac{d\mathbf{p}}{dt} = \mathcal{W}\mathbf{p},$$

(1)

where $\mathbf{p}(t)$ is a vector that describes the probability distribution of system states. $\mathcal{W}$ is the transition matrix, whose elements $\mathcal{W}_{ij}$ specify the transition rates from state $j$ to $i$. The dynamics of oscillations are typically dominated by the first non-zero eigenvalue of $\mathcal{W}$, which is the eigenvalue with the smallest modulus in the real part[42]. This eigenvalue is denoted as $\lambda_1 = -\lambda_R + i\lambda_I$ for the real and imaginary parts $\lambda_R$ and $\lambda_I$, respectively; $\lambda_R, \lambda_I \geq 0$. In general, $\mathbf{p}(t)$ relaxes to the steady-state distribution through damped oscillations, with a decay time $\lambda_R^{-1}$ and oscillation period $\mathcal{T} = 2\pi/\lambda_I$ [42] (also see Supplementary Fig. 3). Following ref. 42, we define coherence as the ratio

$$\mathcal{R} \equiv \frac{\lambda_I}{\lambda_R},$$

(2)

which quantifies the robustness of sustained oscillations before stochastic noise destroys the coherence (more details in Supplementary Section III). We would like to see how our model performs using this metric and what factors contribute to increased coherence.

In typical oscillator models, coherence can be increased by dissipating more free energy. For a general oscillator model described by a master equation, the free energy cost for maintaining the non-equilibrium steady state under constant temperature can be quantified by the entropy production per period $\Delta S$. Denoting the steady-state probability distribution as $\mathbf{p}^s$ and the oscillation period as $\mathcal{T}$, $\Delta S$ is given by[75,76]

$$\Delta S = \frac{\mathcal{T}}{2} \sum_{i,j} (p_j^s \mathcal{W}_{ij} - p_i^s \mathcal{W}_{ji}) \ln\left(\frac{p_j^s \mathcal{W}_{ij}}{p_i^s \mathcal{W}_{ji}}\right).$$

(3)

In the MWC-type model of KaiC studied in ref. 42, for example, increasing $\Delta S$ is necessary to increase coherence. Both quantities increase when the external driving $\mu$ is stronger, although coherence starts to decrease when $\mu$ is increased still further (see Supplementary Fig. 6). This leads to even worse performance for the oscillator as it maintains less coherent oscillations with increasing energetic cost.

On the contrary, our model in the topological regime displays an unusual regime where coherence increases while entropy production per period becomes lower. Figure 3a shows the coherence of our

model as a function of the two parameters $\mu$ and $\rho$ (also see Supplementary Fig. 4 featuring both positive and negative $\rho$). Increasing the thermodynamic force increases coherence monotonically, as expected for typical oscillator models[42]. Going deeper into the topological regime by increasing $\rho$ also leads to higher coherence, as the global currents become more localized on the system edge[63]. However, the entropy production per period $\Delta S$ does not change monotonically with $\mu$. As illustrated in Fig. 3b, in a smaller lattice, the system response becomes localized to the edge as $\mu$ increases. This causes the entropy production on the edge (blue arrows) to increase, while the entropy production in the bulk (orange arrows) decreases, in the region $1.5 < \mu < 7$. Since the bulk contribution typically dominates the edge contribution, i.e., $\mathcal{O}(N^2) \gg \mathcal{O}(N)$ where $N$ is the typical system size, the sum of their contributions also decreases (green curve in Fig. 3c). This negative slope of $\Delta S$ with respect to $\mu$ implies that the system dissipates less free energy overall even when the external driving $\mu$ supplied to each reaction is stronger. In other words, we "get more from pushing less"[77–80]. This unusual regime has a topological origin, since the decrease in the bulk entropy production results from the localization of the steady state onto the system edge. This unusual negative slope leads to a unique experimental signature. Where $\Delta S$ has a negative slope, increasing the ATP concentration (increasing $\mu$) leads to a decrease in total ADP production (decreasing $\Delta S$). This is a striking prediction of our topological model, as shown by the dash-dotted arrow in Fig. 3c. In addition, since coherence increases monotonically with $\mu$, this leads to an efficient regime with simultaneously increasing coherence and decreasing cost. The cost-effectiveness in terms of expending free energy for coherent oscillations can be measured by the ratio $\Delta S/\mathcal{R}$ [81]. See Fig. 3d for the region in parameter space with low $\Delta S/\mathcal{R}$, indicating a highly efficient oscillator.

We can further compare the coherence between different families of KaiC models. We include a simple MWC-type model[42] and a bilayer model that has a lattice structure more similar to ours, adapted from Li et al.[74] (details of each in Supplementary Section IV). Each layer in the bilayer lattice represents the T and S phosphorylation levels along its $x$ and $y$ coordinates, similar to our topological model, with the possibility to switch between the two layers that denote unbound KaiC and KaiB-bound KaiC, respectively (see Supplementary 5b and c). KaiC is more likely to bind to KaiB on the upper right half of each lattice and more likely to unbind on the lower left half. In order to aid comparison, we simplify this lattice model using our $(\mu, \rho)$ parameters (see Supplementary Section IV).

By sampling random parameters in the three models, we find a regime of high thermodynamic driving where the topological model has the highest coherence (Fig. 3e). In the same plot, we also indicate a thermodynamic bound for coherence[42] for each model (dashed lines), which depend on the system size and thermodynamic force $\mu$ (see the "Methods" section). We also plot the bound for a 28-state unicyclic model (purple dashed line in Fig. 3e), which our model approaches in the limit of high $\mu$ and high $\rho$. This is because there are 28 slow internal transitions on the edge that form the effective bottleneck and dominate over the other fast external transitions, resulting in 28 high-probability states, as noted in Fig. 1d. This shows that our topological model approaches the bound set by the most coherent cycle, which is the unicycle with uniform rates ($\gamma_{in}$ and $\gamma'_{in}$ in our case)[42], deep in the topological regime with high external driving.

Given the high coherence of our model, we would like to identify the factors that determine high coherence. Here, we introduce a new predictor of coherence, which is the spectral gap (or band gap) of the system, inspired by the band theory of solids[82]. As the spectral gap measures the separation between modes with different timescales[75], a larger gap predicts greater separation between modes and, hence, the stability of longer-lived modes, as they mix less with transient ones. See Fig. 4a and b for the spectral gap in both imaginary and real space,

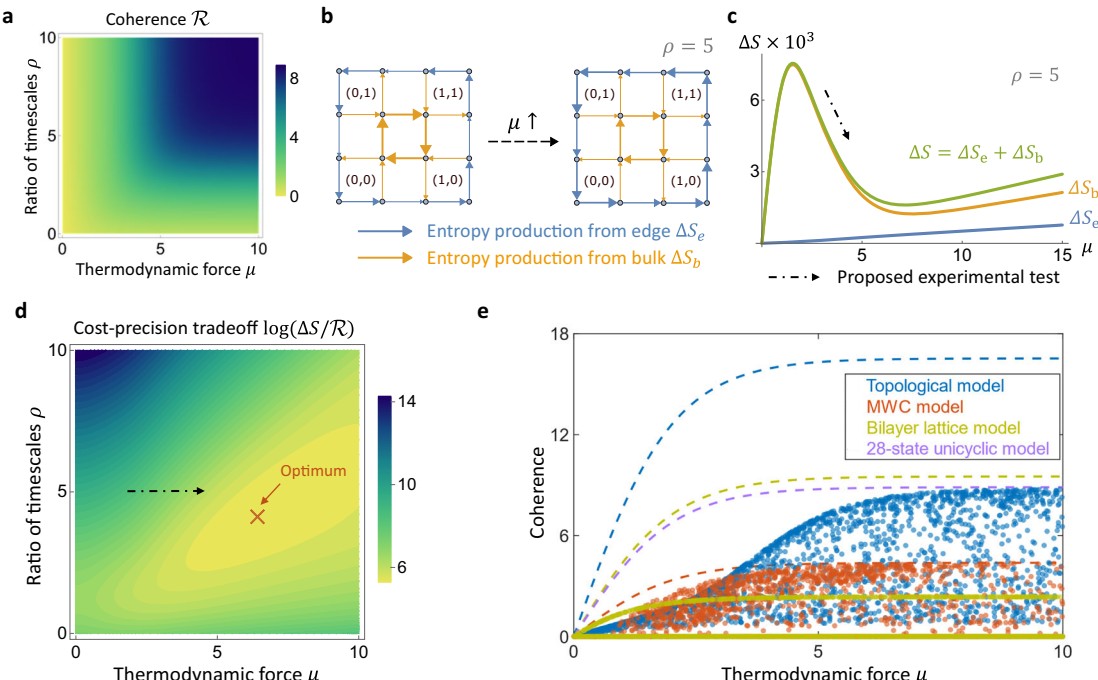

**Fig. 3 | Coherence and an efficient regime with simultaneously increased precision and decreased cost. a** Phase diagram of coherence $\mathcal{R}$ for the topological model, which increases with respect to thermodynamic force $\mu$ and the ratio of timescales $\rho$, as expected. **b** Entropy production per period $\Delta S$ moves from the bulk to the boundary of the system with increasing $\mu$, illustrated on a smaller lattice. The system is in the topological regime; $\rho = 5$. Blue arrows represent entropy production on the edge $\Delta S_e$, and orange arrows represent entropy production in the bulk $\Delta S_b$. Arrow thickness corresponds to the magnitude of the entropy production for the corresponding transitions, and arrows point in the direction of the probability flux. **c** Entropy production per period $\Delta S$ for the topological model ($N_x = N_y = 6$) is decomposed into a bulk and an edge contribution. Due to the localization effects in (**b**), the entropy production on edge $\Delta S_e$ (blue) increases while entropy production in the bulk $\Delta S_b$ (orange) decreases with $\mu$ around $1.5 < \mu < 7$. Typically, the bulk contribution dominates the edge contribution, hence their sum $\Delta S$ (green) also

decreases with increasing $\mu$. Experimentally, the negative slope in this regime predicts a decreased overall ATP consumption when ATP concentration is increased, as indicated by the black dash-dotted arrow. **d** Phase diagram of the cost-precision tradeoff $\Delta S/\mathcal{R}$[81]. The black dash-dotted arrow, similar to that in (**c**) indicates increasing $\mu$ in the energy-efficient regime. This leads to a lower $\Delta S/\mathcal{R}$, which implies a more cost-effective oscillator. There is a global optimum for $\Delta S/\mathcal{R}$ at $\mu = 6.5$, $\rho = 4.1$. **e** Comparison of coherence for different KaiC models with randomly sampled parameters (individual points): there is a strongly driven regime where the topological model has the highest coherence. The dashed lines represent the upper bounds of coherence for the corresponding models in terms of the thermodynamic force $\mu$[42] (see the "Methods" section). The purple dashed line is the upper bound for the 28-state unicyclic model, which is approached by the edge state in the strongly topological regime. Values of $\Delta S$ in all panels are given in units of $\gamma_{tot}k_B$.

$\Delta_I$ and $\Delta_R$, respectively, in the reciprocal space version of our topological model (see Supplementary Section VI). The spectral gap is defined as the difference between the minimum of the topmost band and the maximum of the band below it. Comparing $\Delta_I$ with the imaginary part $\lambda_I$ of the first non-zero eigenvalue $\lambda_1$ in Fig. 4c, we see that they track each other well as $\mu$ is increased. Since coherence $\mathcal{R}$ is the ratio of $\lambda_I$ to its corresponding real part $\lambda_R$ of the same eigenvalue (Eq. (2)) and $\lambda_R$ remains roughly constant with increasing $\mu$ (see Supplementary Fig. 8), $\Delta_I$ tracks $\mathcal{R}$ (Fig. 3a) monotonically in $\mu$ as well. Along the other axis, increasing $\rho$ decreases the spectral dispersion, i.e. it compresses the top band in Fig. 4a. The topmost point of the top band also increases with $\rho$ while the middle green bands in Fig. 4a remain flat, leading to a widened spectral gap $\Delta_I$ for increasing $\rho$. Moreover, $\lambda_R$ decreases more quickly than $\lambda_I$ with increasing $\rho$, so that their ratio $\mathcal{R}$ increases (see Supplementary Fig. 8). Therefore, $\Delta_I$ tracks coherence in $\rho$ as well. Hence, both the imaginary spectral gap and coherence track each other monotonically, as can be seen in their phase diagrams Figs. 4d and 3a, respectively, where both saturate to a maximum following a rapid increase.

Lastly, we examine how the different models perform compared to a conjectured global thermodynamic bound on the full spectrum of $\mathcal{W}$[83]. For a given thermodynamic force, the spectrum is found to lie within an ellipse in the complex plane (purple line in Fig. 4e). The most coherent cycle, a unicyclic network with uniform rates[42], saturates this bound[83]. While the other two families of KaiC models do not saturate

the bound for any sampled parameters, our model approaches this bound as $\rho \to \infty$. This is consistent with our model approaching a uniform unicyclic network when $\rho$ is large, and also with our previous analysis of the spectral gap contributing to high coherence. Similar to the results in the reciprocal space model discussed above, the transition matrix $\mathcal{W}$ also shows a larger imaginary spectral gap as $\rho$ increases. The larger gap moves the topmost eigenvalue upward in the complex plane to saturate the global spectral bound while increasing coherence (see Supplementary Section VI and Supplementary Fig. 7 for details).

## Discussion

We have proposed a topological model that generates coherent oscillations, which supports an unusual regime with increased coherence and simultaneously decreased energetic cost. The mathematical model, which first appeared in ref. 28, is interpreted in a biological context and applied to the KaiABC system. Compared to other KaiABC models, our model has high coherence and more closely saturates a global spectral bound, similar to the most coherent unicyclic models. We also find that the imaginary spectral gap can be used to predict oscillation coherence. Further, the kinetic ordering of the KaiABC phosphorylation cycle arises naturally as an edge current in our model.

In contrast to typical MWC-type models, which usually involve a large number of system-dependent parameters[9,10], our model is parsimonious in that it captures the same phosphorylation dynamics with only a few parameters. Moreover, it also does not require fine-tuning of

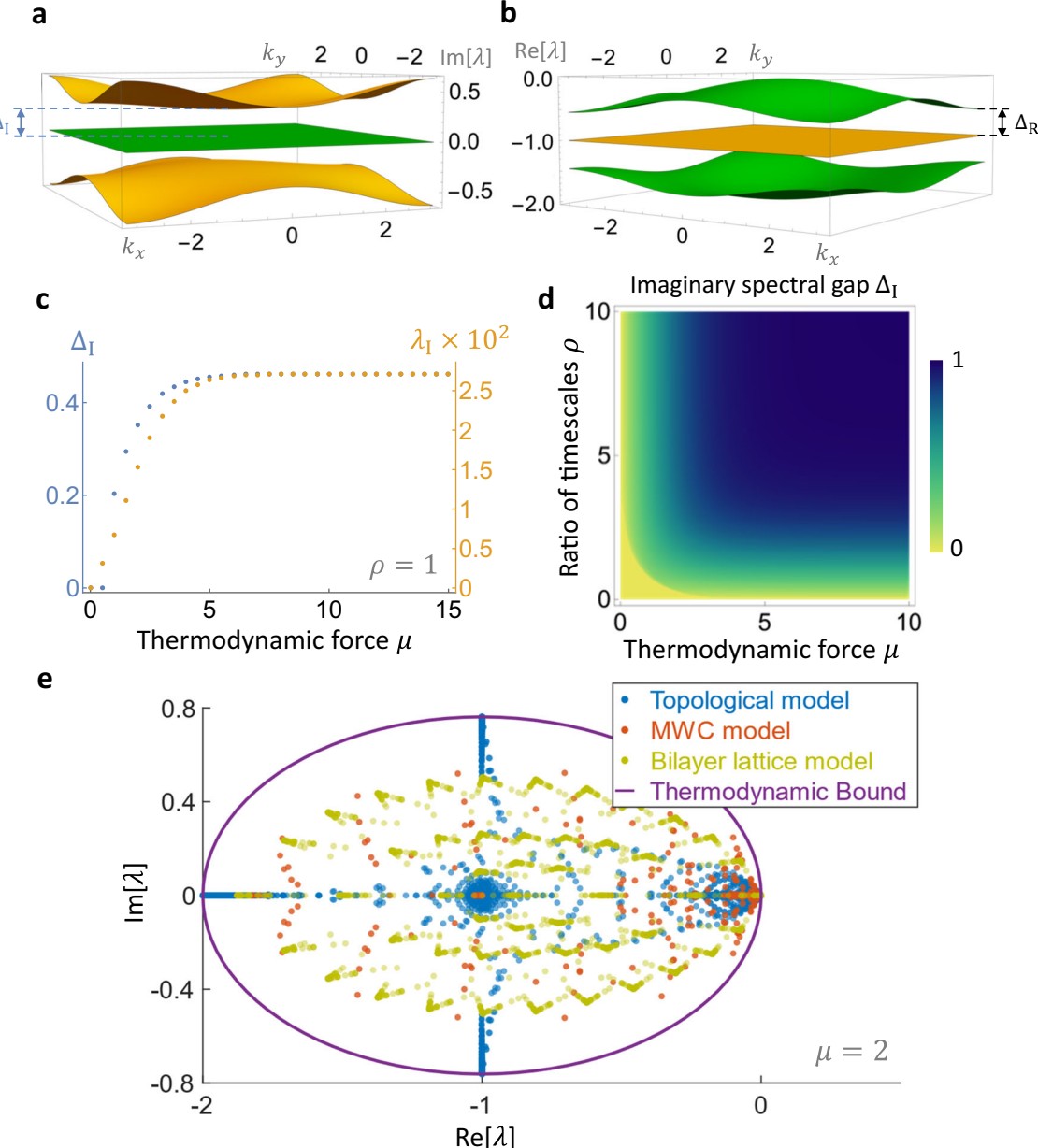

**Fig. 4 | New predictor of coherence and saturation of a global thermodynamic bound. a, b** Spectrum for the topological model in reciprocal space in imaginary space (**a**) and real space (**b**) for $\mu = 1.5$ and $\rho = 1$. The spectral gaps $\Delta_I$ and $\Delta_R$ are defined as the difference between the minimum of the topmost band and the maximum of the second band. **c** The imaginary spectral gap $\Delta_I$ closely tracks the imaginary part $\lambda_I$ of the first non-zero eigenvalue $\lambda_1$, as a function of $\mu$. **d** Phase diagram for the imaginary spectral gap $\Delta_I$ in $\mu$ and $\rho$. $\Delta_I$ and coherence (Fig. 3a) track each other monotonically, and both saturate to a maximum value after a rapid increase. **e** The global spectral bound for a fixed $\mu$ is shown by a purple ellipse in the complex plane[83], while points represent the spectra for the transition matrix $\mathcal{W}$ for different oscillator models. The topological model approaches this bound in the limit $\rho \to \infty$, which is saturated by the most coherent cycle (the unicycle)[83]. Here $\mu = 2$ and other parameters are randomly sampled. For panels, (**a–d**), eigenvalues are given in units of $\gamma_{tot}$. For (**e**), each transition matrix is scaled by an overall factor such that $\min_i[\mathcal{W}_{ii}] = -1$ (see the "Methods" section).

the reaction rates or restriction of the configuration space to an all-or-none conformational change, an assumption that is not supported by emerging experimental evidence[48]. Indeed, recent cryo-EM images of KaiC hexamers suggest that KaiC conformational changes are weakly coupled[48]. From experimental data, the two states consistent with the MWC picture (all buried or all exposed A-loops) are not the most abundant species observed. Rather, hexamers with mixed A-loop conformations make up more than 80% of the entire population. Notably, the coupling constant $J$ from the Ising model fit is very small ($J = 0.086$), and the correlation length $\xi$ is short ($\xi_{AA} \approx \xi_{EE} \approx 0.5$ monomer length)—demonstrating weak cooperativity between monomers.

We further propose three testable experimental signatures for our topological model in the KaiABC system. First, we predict a response unique to our topological model where decreasing ATP concentration leads to an increase in ADP consumption. This is the opposite of what would typically be expected and is due to the system being driven from the edge localized state into bulk dynamics that simultaneously increase energy consumption despite a decrease in coherence. This is unique to our topological model and discussed in Fig. 3c. We expect a 5-fold increase in ADP consumption for a 10-fold decrease in ATP concentration, which can be measured via tracking ADP levels[69] or alternatively with heat dissipation[84]. Second, we predict that KaiC

hexamers should have a wide distribution of different A-loop conformational patterns when imaged over the period of an oscillation. This would be a similar pattern to what has been observed for non-oscillating KaiC mutants[48], which would distinguish our model from MWC-type models[9,10,59], where all six A-loops are expected to be mostly buried or exposed. Lastly, mutants that do not oscillate[85] can be used to avoid potential confounds due to synchronization effects. As our model couples S phosphorylation with A-loop burial (while still allowing independent conformational change), this predicts that mutants mimicking phosphorylated S-sites would correlate with hexamers having mostly buried A-loops with the highest frequency. Meanwhile, mutants mimicking unphosphorylated S-sites would correlate with hexamers having mostly exposed A-loops.

Our model gives rise to the observed sequence of phosphorylation reactions in KaiABC without having to tune as many free parameters as in typical models[9,10]. This is due to the repetition of simple motifs in our network, which allows the flexible generation of emergent timescales. Further, we note that population effects in KaiABC are known to be important, e.g., by promoting the synchronization of many molecules in the KaiABC system[9,35,56]. Still, the oscillation and coherence of individual molecules are generally presumed to be building blocks for sustained oscillations at the population level, even while the extent of the single-molecule contribution remains unclear. Hence, our work focuses on the oscillations and coherence of single molecules as a first step towards the understanding of population-level coherence, with the exploration of multiple molecules and their synchronization to be left for future work. See Section VII in Supplementary Information for preliminary work on generalizing our single-molecule model to many molecules and incorporating population-level effects such as competition for KaiA molecules.

By rigorously embedding topological methods within non-equilibrium statistical physics, our work generalizes their usage for various biological and chemical systems. Our results suggest a new mechanism that utilizes dissipative cycles to produce emergent oscillations or attractor states in biological systems. This mechanism can be tested by introducing perturbations or mutant proteins[86], in order to jointly analyze how these changes modify the robustness and coherence of the global cycle. Even though we have mostly discussed our model in the context of the KaiABC system, the model is more general. It can be mapped to various other biological systems where $x$ and $y$ represent other possible types of molecular modifications, such as polymerization[28]. More broadly, our model provides a blueprint for the design of synthetic oscillators, which is becoming increasingly feasible due to new experimental developments[87,88]. While designing synthetic oscillators that are robust across different parameters or changes in the environment remains a challenge[89,90], this project provides new models for robust oscillators and continuous attractor dynamics in various biochemical scenarios and changing conditions.

## Methods

### Simulation of the system steady-state
The steady-state probability distribution in Fig. 1d is obtained by simulating the system dynamics with the Gillespie algorithm[91]. The simulation is run for $10^8$ steps with a random initial condition, and the probability for each state is given by the fraction of time the system spends in that state.

### Thermodynamic bound for coherence
Ref. 42 conjectures an upper bound for coherence for any stochastic system. Suppose that a cycle $\kappa$ has $N_\kappa$ states labeled by $\kappa_1, \kappa_2, ..., \kappa_{N_\kappa}$ such that $\kappa_1$ is connected to $\kappa_{N_\kappa}$ and $\kappa_2$, $\kappa_2$ is connected to $\kappa_1$ and $\kappa_3$, etc. Define the affinity of $\kappa$ as $\mathcal{A}_\kappa \equiv \ln \prod_{i=1}^{N_\kappa} \frac{\mathcal{W}_{\kappa_{i+1}\kappa_i}}{\mathcal{W}_{\kappa_i \kappa_{i+1}}}$, where $\kappa_{N_\kappa+1}$ is the same

as $\kappa_1$. For an arbitrary stochastic model, we look at all possible cycles in the underlying network. The upper bound for coherence in Fig. 3e is given by

$$\mathcal{R} \le \max_\kappa \left\{ \cot(\pi/N_\kappa) \tanh[\mathcal{A}_\kappa/(2N_\kappa)] \right\}.$$

For the topological model, the thermodynamic force per state $\mathcal{A}_\kappa/N_\kappa$ is always $\mu$, and the cycle that maximizes the right-hand side of the bound is the global cycle going around the boundary with $N_\kappa = 52$.

### Global spectral bound for a driven stochastic system
The global bound in Fig. 4e is conjectured by Uhl and Seifert[83] for the spectrum of any transition matrix for a master equation. To obtain the bound, we look for a cycle that maximizes the affinity per state $\mathcal{A}_\kappa/N_\kappa$. Denote this maximum by $\mathcal{A}_\mathcal{C}/N_\mathcal{C}$. We also define $w_0 = \max_i[|\mathcal{W}_{ii}|]$. The spectrum is hypothesized to lie entirely in the ellipse given by

$$g(x) = w_0 \left\{ -1 + \cos(2\pi x) + i \tanh[\mathcal{A}_\mathcal{C}/(2N_\mathcal{C})] \sin(2\pi x) \right\}.$$

For the topological model, we have $\mathcal{A}_\mathcal{C}/N_\mathcal{C} = \mu$. The bound for $\mu = 2$ is plotted in purple in Fig. 4e. Each model has a different $w_0$ for their corresponding transition matrix. To plot all spectra under a common bound, we rescale each transition matrix by a constant factor such that $w_0 = 1$ for all transition matrices considered.

### Sampling random parameters
In Fig. 3e, for the topological model and the bilayer lattice model, we randomly select the parameters $\mu \in [0, 10]$ and $\rho \in [0, 7]$ from uniform distributions on each interval. For the MWC model in ref. 42 (see Supplementary Section IV for parameter definitions), we select from the uniform distributions $\gamma \in [3, 7]$, $E \in [5, 15]$ and $\eta \in [0, \frac{70}{3}]$. The parameterization of the MWC model is such that the maximum affinity per state is $\mathcal{A}/N = \frac{3}{7}\eta$. Because we keep $\mathcal{A}/N$ (which we also call thermodynamic force per state and simply denote as $\mu$) the same in Fig. 3e, $\eta \in [0, \frac{70}{3}]$ exactly corresponds to $\mu \in [0, 10]$. For Fig. 4e, we fix the thermodynamic force per state $\mu = 2$ (which is $\eta = \frac{14}{3}$ for the MWC model) and sample the remaining parameters $\rho$, $\gamma$, and $E$ in the same way as above.

### Reporting summary
Further information on research design is available in the Nature Portfolio Reporting Summary linked to this article.

## Data availability
All data used to produce the figures in this manuscript can be generated from the computer codes deposited at https://doi.org/10.5281/zenodo.12593308.

## Code availability
Mathematica and MATLAB codes used to generate the data and figures in this manuscript have been deposited on Zenodo at https://doi.org/10.5281/zenodo.12593308.

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

## Acknowledgements

We are grateful to Jordan Horowitz, David Lubensky, Jaime Agudo-Canalejo, Yuhai Tu, and Michael Rust for helpful discussions. In addition, we thank Pankaj Mehta, Peter Wolynes, Ulrich Schwarz, and Oleg Igoshin for their thoughtful comments. C.Z. and E.T. acknowledge support from the NSF CAREER Award (DMR-2238667), and E.T. acknowledges support from the NSF Center for Theoretical Biological Physics (PHY-2019745).

## Author contributions

E.T. conceived the research project. C.Z. performed the simulations and calculations. C.Z. and E.T. connected biophysical mechanisms in KaiABC with the theoretical model, and wrote the manuscript. E.T. supervised the project.

## Competing interests

The authors declare no competing interests.
