## [Peer Review File · Nature Communications]

REVIEWER COMMENTS

Reviewer #1 (Remarks to the Author):

This paper proposes a new model to explain the robustness of circadian rhythms, based on a network view of transitions between states of a hexameric KaiC molecule. I believe the underlying molecular biology is well understood in principle. However I don't have the expertise to comment onto whether the model reduction is appropriate for the biology in question. I will instead comment on the dynamical systems aspect of the model itself.

The principal result of the paper is that (tuned by rather general parameters) there is a boundary between a regime of chaotic bulk dynamics where all configurations are observed, and a "topological" state where the system transitions between a small subset of states that can be seen as at the boundary of the lattice. I think there is good evidence given in the paper that this is true.

I query - and I think this is not just nit-picking - whether this should be called "topological" in the modern sense of condensed matter physics (since this comparison is made in the paper, I feel it is fair to quibble). Topology is not just the existence of an edge state - it implies that there is some conserved quantity (typically the ratio of an edge current to a driving field) that is fixed in some set of basic units. I don't see that here. If I am missing this point, I'd like to see it derived clearly. Edge states can arise in more banal circumstances - e.g. a type I superconductor in a magnetic field below H_{c1} . In this case they reflect some collective state of the bulk.

These results feel to me like something slightly different. The system of equations is manifestly non-reciprocal and also has explicitly PT(parity-time)-broken symmetry. We know that (as a function of parameters) in models like this one can have a transition between a time-independent state (stasis), a disordered state (time-dependent but random) and a chiral state. The transition boundary is an exceptional point of the dynamics, and it has recently been getting some attention. Because this model has a conservation law, the chiral state (I think) will have net currents only at the boundary. If the authors disagree with my conclusion they should refute it.

But it is an important distinction between explicitly topological effects, and explicitly bulk transitions in a finite system with a conservation law.

I find the results about entropy generation and energy consumption interesting. Obviously this will be a consideration for the biology.

Overall I think that this paper raises interesting questions about an important biological process. I think it is not quite finished.

Some narrower comments, courtesy of a graduate student who I asked to read this paper.

1) Why is this topological? In my mind, topological protection means there is some conserved number that depends only on the topology, and the transition rate to other states is zero for a topologically protected one. Is it because it is 2d? The SI has a model with 2 layers – so now the topology is different. So it seems to only be a property of having an edge.

2) I still don't understand why the negative values of ρ and μ don't produce the same phases.

3) Is the entropy in the paper defined only for their model, or for general ones? May be restructure the paragraph around eq. 3 in the paper.

4) In the SI they model with periodic boundary conditions – then there is actually topology, but the results are similar. Why?

5) In figure S6, how do they know which states are bulk vs edge?

6) Why are the imaginary axes for the eigenvalues at $\text{Re}[\lambda] = -1$

Reviewer #2 (Remarks to the Author):

The idea of using topology to produce stable cyclic behaviors in an unstable world is an attractive one. We might expect to see this arise if there is a 2D lattice of states, and the dynamics are in a regime which favors the edge. And we might then hope that it will give more stable oscillations than competing mechanisms.

The present paper claims to have such a model, and to produce such stability. The model is for a circadian clock, and it works by drawing a grid of possible phosphorelation states, on which the dynamics prefer to walk around the edge. This echoes what is observed in reality. It is claimed that this state has a high coherence (which is a sort-of Q-value) with moderate "thermodynamic force".

However, I have reservations about the model, about its novelty, and about the presentation. I believe these can be addressed, but will require substantial changes.

The model presented is similar to one in an earlier paper (2021 PRX, reference [31], E. Tang et. al.) which I had to read to understand it. But crucially, this variant seems at first to have one more dimension -- a 3-dimensional state space, not 2. The boundary of a cube is topologically quite different to that of a square, with no protected 1-cycles. Deep in the description, there might be a constraint introduced which reduces it to 2 dimensions — this took several readings to uncover, and I'm still not sure.

This now seems like the central question for applying topological ideas to state space: How many dimensions are there?

In solid state physics, the number of dimensions is that of physical space: 3 for a block of metal, 2 if it's really thin (or a magnetic field strongly separates 1 from the rest). The edge states literally live on edges you can touch with your finger.

Here, the topological state is instead a particular walk in a molecular state space. It was claimed in the earlier paper that this state space should have 2 bounded dimensions (a phosphorelation level, and a conformational change) as well as 1 circular one (some kind of ligation state). This paper expands the list to include 1 more bounded dimension, another phosphorelation level. The second phosphorelation state seems to have better experimental grounding than the cyclic ligation state. Perhaps one could go further, and include other details. Would it still work? Is the topological state generic? Or does it rest completely on a hard constraint to exactly lock the conformational state to one of the phosphorelation levels, to remove one bounded dimension?

This question is what I feel the paper ought to address. I'm not quite sure how. If the answer for now is that the dimension is fixed to exactly 2 bounded dimensions by hand, then this needs to be stated very

clearly as a major assumption. If 2 dimensions is an emergent property, then how that works sounds interesting.

Aside from this major question, I think that the presentation should be more straight-forward. At present it muddles together what is actually done with all sorts of analogies, and is unclear about the differences from earlier work. I have more detailed comments below but the big three are:

1). The model needs to be clearly described on its own terms (i.e. as a mathematical model). The complete list of what specifies its state, and the complete list of parameters, must be written down clearly, at once, in the main paper. Not trickled in over 4 pages, not in the supplement. The motivating analogy to biological mechanisms is also interesting, but should never be used to obscure what's actually being done.

2). In particular, states ABCD need to be re-named, to avoid confusion with KaiA, KaiB, KaiC. (Perhaps compass directions N-E-S-W would be natural, or just numbers.) This cyclic 4-state system is important to the model, and deserves its own sub-figure. If there is some real cyclic 4-state system, any discussion of the analogy must start off with clearly distinct names for the things it wishes to compare. Similarly, Fig 1(e) needs to be moved out to be an independent figure with its own caption, as it's describing the real system, not the model.

3). The exact relation to the earlier model (2021 PRX) must be described clearly. This one introduces extra complexity (in the name of realism?) in a way which looks like it may destroy the good properties. But perhaps extra constraints are added, to restore them. Is the final result mathematically equivalent (up to a change of co-ordinates) or not? If not, would the old one look different on Figure 2, say? (Then it should probably appear on most plots.) The fact that this is a variant of an earlier model needs to be clearly stated in the introduction.

What follows is a description of how the model works. Below that are more detailed presentation comments.

///// 2021 model

To understand this paper's model I had to read an earlier paper. My understanding of how that works is as follows:

The state is specified by (x,y) in a compact subset of Z^2 (the integer plane), together with what I'll call s in Z_4 (integers modulo 4), whose elements are named A,B,C,D in the paper. The complete state is written with a subscript for s , like $(1,2)_C$.

The dynamics always moves to a nearest neighbor in both (x,y) and in s , with s cyclic. The position (x,y) is rigidly coupled to s as follows:

[SEP] If s is decremented, then (x,y) does not change. This is called an "internal" transition.

[SEP] If s is incremented, then (x,y) must change in a particular compass direction, determined by s , clockwise in the plane, N-E-S-W. Thus increment $s=1 \rightarrow s=2$ goes with $y \rightarrow y+1$, North, while $s=2 \rightarrow 3$ goes with $x \rightarrow x+1$, East, and so on. Such transitions are called "external". If the resulting (x,y) is outside the allowed subset, then the transition is impossible, s cannot be incremented.

The essence of the topological state is this: After an "external" step along a straight boundary of the allowed (x,y) region, with the disallowed region to the right, the next "external" will be impossible: The clockwise increment would make it point into the wall. Thus an "internal" step is the only possibility, and brings s back to what it was, allowing another "external" step in the same direction. This is most likely when "internal" moves are rare, i.e. when the rate constant for "external" is much larger. Only in this limit is the state topological.

Section IV of the 2021 paper proposes this system as a model for circadian rhythms, the KaiABC system. This involves a hexamer i.e. a complex of 6 similar proteins, monomers. Each has two conformational states, and the mapping is that $x = 0,1,\dots,6$ counts how many of the monomers are in one conformation (ignoring order), while $y = 0,1,\dots,6$ counts how many of them are phosphorelated (ignoring which ones).

What is attractive is that the phosphorelation states are not constrained to be occupied in some order; instead, the fact that this happens is an emergent property of the edge state, which walks the boundary of the square in (x,y) state space. That the internal state s is cyclic is, however, put in by hand. This cyclic choice of compass direction of the allowed "external" step is what produces the topological state.

////// Changes in the 2023 model

The goal of the present paper seems to be to elaborate this model, to make it more realistic.

The principal change made is to allow for two kinds of phosphorelation sites, T and S, instead of one. They map these to the mathematical model's x and y parameters. However, they also still include the conformation state, giving another variable which I'll name $z = 0, 1, \dots, 6$, the "number of diamonds" in the figures. Thus it appears that the state is now $(x, y, z), s$, with (x, y, z) lying in the cube $\{0, 1, \dots, 6\}^3$. The cube's boundary is topologically quite different, it does not have an incontractable 1-cycle, so at first glance it seems no previous results should be expected to survive.

However, three other changes are introduced:

**1. When s is decremented, then (x, y) does not change, but z sometimes must change: $B \rightarrow A$ increments z, while $D \rightarrow C$ decrements z, other steps do not change it. This is still called "internal", but is presumably forbidden when z is 6 or 0 respectively. With this change alone (from the model above) some states could never be escaped. ^[1]_{SEP}

**2. These two reactions are also allowed to run backwards, with smaller rates (γ -prime). I believe there are now 4 independent rate constants (although the figure confusingly labels some arrows with hypothesized KaiB relations, instead of rates). ^[1]_{SEP}

**3. Finally, a few pages later, there may or may not be another connection between z (the "number of diamonds") and y. If I have parsed this correctly, it seems to say that when $s=B$ or C , they fix $z=y-1$, while when $s=A$ or D , instead $z=y$. (Page 5, line 262, 2nd last paragraph before section III.) But this contradicts Figure 1a.

At present it's unclear to me whether the end z (conformation) is an independent dynamical variable, or not. If not, the complete state of the system is specified by $(x, y)_s$ as before.

(The 2021 paper eventually allows point 2 too, but the basic concept can be grasped without it.)

So the new paper adds more complexity to the model's state, in a way which changes the topology, and also adds more complexity to the dynamics. The desired outcome seems to be only not to break the old model's edge states -- to pick a regime in which this 4-parameter model has the same emergent behavior as the previous 2-parameter one. This is the principal technical change and its motivation, and consequences (or lack thereof) seem to be barely discussed.

Are there really 4 rate constants? Later on page 5 (line 324, “Before analyzing...”) some equations write extra constraints, such that there are only 3.

///// Presentation comments

The writing is polished, but it is not easy to see where the contribution starts. I feel it should spend less time on what is hoped for, and more time on what is actually done.

The introduction can be shorter. That would make space for saying by what measures success will be judged. Is the metric Q-value per energy consumption, or what? Is “thermodynamic force” just some parameter combination or something real?

I think it is confusing to first say that (x,y) could be anything (line 133, “They could also represent the state of a biopolymer assembled...”) before describing this paper's actual model. The model is confusing enough on its own, without hypothetical variants. This can be reversed — point out in the conclusion that much the same mathematics could perhaps be mapped to other systems.

///// Fig 1 / section 2

Figure 1 introduces the model. It is, I think, impossible to parse without reading the 2021 paper. By my count there are 10 different meaningful colors, 5 distinct kinds of arrows (with many rate constants), and 5 differently shaped objects, in panels (a) and (b) alone. It is fantastically hard to puzzle out the correspondence between these pictures and the caption text below.

The model's pieces need to be clearly, *individually*, described before putting them together. Starting with the cyclic part of the state (my s above) in isolation, and the important fact that it's constrained to move $A \leftrightarrow B \leftrightarrow C \leftrightarrow D \leftrightarrow A$. (New sub-figure, showing a compass, N-E-S-W?) This cycle is an important ingredient. Then the conformation and the phosphorelation of the monomers, x,y,z. (Another new sub-figure? Not sure.) Then their transitions. (Existing a,b with simpler arrow labels?)

It must be abundantly clear exactly what numbers specify the complete state of the model. It must be abundantly clear exactly how many different rate constants are present, and what is assumed about their magnitudes. It must be abundantly clear whether _which_ monomer is phosphorelated matters or not, whether the positions of the monomers matters or not. Whether the attachment point of the curly

line connecting the A to the monomer matters or not. The reader should not have to guess these things by playing sudoku.

Pg 5 (line 324) says it is "simplifying the notation". But in fact it seems to impose one new constraint on the 4 rate constants, so that only 3 are independent. Were they coupled this way before, does it matter? Why not just tell us how many parameters there are at the beginning?

It's very confusing, deceptive even, to call the internal cyclic state ABCD, and I think this ought to be changed. They are drawn as if they are somehow subsidiary components, but their constraint to change only in a cyclic order $A \leftrightarrow B \leftrightarrow C \leftrightarrow D \leftrightarrow A$ is really key to the whole model. It's not clear to me whether they map to any known components of the real system. While KaiA and KaiB are real things, I don't see the analogy, as the rest of the chain seems to be missing. Clearly C cannot be KaiC as that's already something else.

Figure 1(c) highlights the block $(x,y) = (1,1)$, why not the exact same block $(2,1)$ as Fig. 1(b)? (And perhaps, why not highlight where 1(a) fits in, too?)

Figure 1(d) changes the arrangement of these blocks to group them more tightly. Can the two be drawn more similarly? At least the thing highlighting where the $(2,1)$ block from 1(b) lives, could be drawn in both.

The grouping here puts the slow transitions tightly together, and the fast transitions further apart. So it has 24 blocks on the edge, but 28 yellow sites. Instead of introducing one more kind of arrow around the cube for the cyclic motion, perhaps numbering the high-probability sites 1,2,3... 27,28 around the perimeter would help explain where this number comes from.

If Fig 1(d) is data not a cartoon, it should show the what y_s are used. And should probably pick less extreme values, so that some intermediate colors are shown. ("Occupancy" might be clearer than "Prob." too? Or "steady-state probability"?)

Fig 1(f) shows 24 edge states, not 28. This is wrong, surely.

In Figure 1(e), I think that a 45 degree line dividing day from night is confusing. The text claims that the T and S phosphorelation stages both occur during the day. Perhaps we are meant only to notice what color the small arrows are on, not what color the large diagrams are on. Surely this can be done in a less

confusing way. It would help if this wasn't muddled up with the model -- probably it should be a new figure, with its own caption.

I do not know what we are to take from KaiA and KaiB on this figure. They are not mentioned in the caption. Note that KaiC is labelled only once on this figure, as if to hint at an ABCX cycle... but in fact KaiC applies equally to every part of the figure. This will be less confusing once the internal cycle is not called ABCD anymore.

////// Fig 2 / section 3

Below (1), "first nonzero eigenvalue". I think this mean the nonzero eigenvalue of smallest absolute value? "first" doesn't make much sense unless you have told us how you order them.

" ΔS_e (blue) increases while ΔS_b (orange) decreases with μ ". This seems to describe only $1 < \mu < 5$ or so, which is confusing.

Figure 2(b) has at least 4 kinds of arrows. I think one black arrow is the word "increases". I do not understand what the blue and orange arrows are. The axes are not labelled but I assume they are x, y.

I am not sure that I have puzzled out why the sign of the slope of ΔS with μ is interesting. Can the goals here be made obvious?

Related SI:

Same comment about "first non-zero eigenvalue" below (S2).

The MWC model presented in (S4) (S5) is a pretty complicated parameterization with lots and lots of researcher degrees of freedom. Apparently this is taken from a reference, so not changing it is probably the right thing to do.

Presentation: if $k=1$, then why introduce it at all? Since χ depends on i , why not write χ_i with a subscript? Or just immediately write the rates as being either 1 or $\exp(E^*(i-3)/3)$? Also, why call the phosphorelation level i here, not x or y as elsewhere?

Fig S4: This has 4 colors & 3 styles of arrows for rate constants. I'm not sure why some are red. I think I eventually puzzled out why some are green, but this does not have to be a sudoku puzzle. Panel (a) could have axes arrows phosphorelation & activity like (b1).

Fig S5: Is the green y-axis $\Delta S/T$? Or maybe we are to puzzle out that it is $\Delta S/(T k_B \gamma_{tot})$? I think the symbols should appear in the label. (Likewise Coherence R, as before.) All x-axes are μ , I think, although their ranges differ and only the right hand ones are labelled.

///// Fig S1

I'm unsure that this idea of deleting some states from the grid makes sense for a state-space model. It seems to be borrowed from solid state physics, where obviously one may chip a bit off a crystal. Here, why would the allowed S-phosphorelation states depend on how many T-phosphorelations are present?

Reviewer #3 (Remarks to the Author):

In this manuscript the authors propose a mechanism for coherent biochemical oscillations that relies on the emergence of a "topological phase" where the system gets trapped into a region of the space states. In particular, their model is 2D model and this region corresponds to the edges of a 2D lattice. The authors also discuss the applicability of their model to the KaiC system, one the best understood circadian clocks. They also discuss the relation between energy dissipation and coherent oscillations in the light of known thermodynamic bounds, showing that it is possible for the model to saturate such bound in a limit of timescale separation.

I think the idea proposed by the authors connecting topology concepts from "traditional" condensed matter physics with biochemical oscillations is interesting, independent of the speculation that their model might be

a good model for the specific KaiC system. I am inclined to recommend acceptance of the manuscript.

1) In the intro: "It remains unclear how such oscillations ... emerge". Not the best sentence. There are many mechanisms, such as certain feedback loops in chemical reactions, that demonstrate that such oscillations can occur. While it is valuable to propose a new mechanism it is not correct that it is unclear how such oscillations emerge. Of course there is much to understand in this field.

2) In Fig. 1f the authors connect their 2D model with a single KaiC molecule. How does the connection work with more KaiC molecules? Is the network still 2D if we have two KaiC molecules?

3) One feature that is found in many biochemical oscillators is the "competition for a scarce resource". In the case of the KaiC system this scarce resource would be the KaiA molecules. Roughly the number of KaiC molecules that "synchronizes" is equal to the number of KaiA molecules, which must be smaller than the number of KaiC molecules. The typical models for the KaiC system mentioned by the authors have this feature. Does the present 2D model have this feature? If we were to remove the KaiA molecules from the system what would happen to the present model?

4) In models for biochemical oscillators with "competition for a scarce resource" the network of states is typically very complicated. But features such as this competition and feedback loops will make the system get trapped in some sort of "limit cycle". Well there might be more than one cycle. Would the authors say that this is a similar thing to what is happening in their model? In other words, is their model a nice and simple way to express this idea of dimension reduction

that probably also happens in other models for biochemical oscillators or do the authors think that this emergence of a topological cycle is new feature in their model different from other models for biochemical oscillators?

5) Ref 43. I think the authors mean to cite Phys. Rev. E 106, 014106 2022, which contains a bound in terms of the thermodynamic cost ΔS and not the thermodynamic affinity.

Response to the Referees for “A topological mechanism for robust and efficient global oscillations in biological networks”

Chongbin Zheng^{1,2}, Evelyn Tang^{1,2,*}

**1 Center for Theoretical Biological Physics
Rice University, Houston, Texas 77005, USA**

**2 Center for Theoretical Biological Physics
Rice University, Houston, Texas 77005, USA**

*** E-mail: Corresponding e.tang@rice.edu**

I. REVIEWER 1

This paper proposes a new model to explain the robustness of circadian rhythms, based on a network view of transitions between states of a hexameric KaiC molecule. I believe the underlying molecular biology is well understood in principle. However I don't have the expertise to comment onto whether the model reduction is appropriate for the biology in question. I will instead comment on the dynamical systems aspect of the model itself.

The principal result of the paper is that (tuned by rather general parameters) there is a boundary between a regime of chaotic bulk dynamics where all configurations are observed, and a "topological" state where the system transitions between a small subset of states that can be seen as at the boundary of the lattice. I think there is good evidence given in the paper that this is true.

We thank the reviewer for their positive feedback.

I query - and I think this is not just nit-picking - whether this should be called "topological" in the modern sense of condensed matter physics (since this comparison is made in the paper, I feel it is fair to quibble). Topology is not just the existence of an edge state - it implies that there is some conserved quantity (typically the ratio of an edge current to a driving field) that is fixed in some set of basic units. I don't see that here. If I am missing this point, I'd like to see it derived clearly. Edge states can arise in more banal circumstances - e.g. a type I superconductor in a magnetic field below H_{c1} . In this case they reflect some collective state of the bulk.

The reviewer is right to point out that in modern condensed matter physics, topological systems often have a conserved quantity that comes about due to their topological invariant. For instance, quantum Hall systems have a conserved Hall conductance σ_{xy} due to their Chern number n , where $\sigma_{xy} = n \frac{e^2}{h}$. However, whether or not there is a clear physical signature of the topological invariant, as in the quantum Hall effect, topological systems are distinguished by their topological invariant, whereas the physical response may or may not be detectable. We agree that the topological invariant in our system has not been derived clearly before this, and have added these lines in the sixth paragraph of the main text: "The edge state and the associated dynamical regime have a topological origin, as their emergence is governed by the presence of a topological invariant, the 2D Zak phase [1, 2]. The Zak phase is zero when $\rho < 0$ but becomes nontrivial in the topological regime $\rho > 0$ (see Supplementary Information)."

We continue to explicitly write the mathematical definition of the 2D Zak phase and explain it in a new section of the Supplementary Information titled "The 2D Zak phase"; reproduced below.

- Our model topology is characterized by the topological invariant known as the 2D Zak phase [1, 2]. It is defined as an integral over the 2D Brillouin zone:

$$\Phi_i^c = \frac{1}{2\pi} \int_{BZ} \mathbf{A}_i(k_x, k_y) dk_x dk_y, \quad (1)$$

where $\mathbf{A}_i(k_x, k_y)$ is the Berry connection for the i -th band, given by $\mathbf{A}_i(k_x, k_y) = i \langle \phi_i | \partial_{\mathbf{k}} | \psi_i \rangle$ for left and right eigenvectors $\langle \phi_i |$ and $|\psi_i\rangle$, respectively. For our stochastic system, we focus on the Zak phase Φ_h^c for the highest band h in real space (upper green band in Fig. 4b in the main text), which is the band closest to the steady state eigenvalue of 0. It turns out that when our system is in the trivial regime ($\rho < 0$), we have $\Phi_h^c = (0, 0)$. When our system is in the topological regime ($\rho > 0$), we have $\Phi_h^c = (\pi, \pi)$. The emergence of the dynamical regime with edge currents and edge-localized steady state coincides with a nontrivial 2D Zak phase, indicating its topological origin.

The "edge states" in a type I superconductor are, as the reviewer says, reflecting some collective state of the bulk, and hence scale with the full system size. This is qualitatively different from the

edge states in a topological system, that scales with the boundary of the system. For instance, in a 2-dimensional system, the former would scale as L^2 , where L is the system width. In contrast, a topological edge state would scale with L , as the edge state in our system does. This interplay between the system bulk and its effects on the system boundary (which is one dimension smaller) are indicative of unique topological effects via the bulk-boundary correspondence.

These results feel to me like something slightly different. The system of equations is manifestly non-reciprocal and also has explicitly PT(parity-time)-broken symmetry. We know that (as a function of parameters) in models like this one can have a transition between a time-independent state (stasis), a disordered state (time-dependent but random) and a chiral state. The transition boundary is an exceptional point of the dynamics, and it has recently been getting some attention. Because this model has a conservation law, the chiral state (I think) will have net currents only at the boundary. If the authors disagree with my conclusion they should refute it.

But it is an important distinction between explicitly topological effects, and explicitly bulk transitions in a finite system with a conservation law.

Models with a transition between time-independent, disordered, and chiral states have been seen in work by Fruchart et al. [3]. However, their results emerge in a continuous description of many-body systems that interact with each other in a non-reciprocal manner, whereas ours is a discrete description for a single particle system that does not have interactions between different species.

Further, PT symmetry is broken differently in both cases. The chiral state in Fruchart et al. does not break PT symmetry explicitly in the model equations. PT symmetry is broken spontaneously. Therefore, it is possible for the chiral state to circulate in either direction. However, our master equation explicitly breaks detailed balance so the edge current in our system can only circulate in one direction, which is counterclockwise for $\mu > 0$. These differences further illustrate the unique properties of our topological edge states, which are only a property of the edge and are unrelated to bulk effects.

I find the results about entropy generation and energy consumption interesting. Obviously this will be a consideration for the biology.

Overall I think that this paper raises interesting questions about an important biological process. I think it is not quite finished.

Indeed, we do hope that the paper will generate interest in biological comparison and testing. We thank the reviewer for their positive evaluation.

Some narrower comments, courtesy of a graduate student who I asked to read this paper.

1) Why is this topological? In my mind, topological protection means there is some conserved number that depends only on the topology, and the transition rate to other states is zero for a topologically protected one. Is it because it is 2d? The SI has a model with 2 layers – so now the topology is different. So it seems to only be a property of having an edge.

Indeed, topological protection means there is a conserved number, i.e., the topological invariant of the system. For our model, the invariant is the 2D Zak phase. As mentioned above, we have added relevant discussions of the Zak phase in the main text and Supplementary Information. A corollary of this is that in our network no individual state is topologically protected. The nontrivial topology, which gives rise to the edge localization in the steady state, is a global property of the system that is not reflected in any local set of states. This global property can be identified via the Zak phase calculation, as discussed above.

Our particular topological model is indeed set in 2D. However, topology is a broader concept that extends over several dimensions. There are topological models in other dimensions, such as the 1D

SSH chain or the 3D topological insulators. While topological systems typically possess nontrivial properties on their edges, having edges generically does not mean that a system is topological. For instance, the bilayer lattice model in the SI has no nontrivial topological invariant and also no edge states, even though the model does have an edge.

2) *I still don't understand why the negative values of rho and mu don't produce the same phases.*

Actually, the system is mirror-symmetric around $\mu = 0$, so negative values of μ do produce the same phases as positive values of μ – agreeing with the reviewer's intuition there. However, the topological phase transition boundary occurs exactly at $\rho = 0$, which separates the trivial phase at $\rho < 0$ and the topological phase at $\rho > 0$.

We have included more detailed descriptions of the dynamics in both regimes in the sixth paragraph of the main text, which we hope makes the distinctions between $\rho < 0$ and $\rho > 0$ more transparent. The first half of this paragraph, which is relevant for this discussion, is reproduced here:

- There are two dynamical regimes for the system, determined by tuning the parameter ρ . In the trivial regime when $\rho < 0$, the system tends to go through local counterclockwise cycles via internal transitions γ_{in} , interspersed with slower external transitions γ_{ex} that break out of these cycles. Such behavior is observed both in the bulk and on the edge of the state space. Over long times, the system displays diffusive dynamics and will ergodically explore the whole state space. On the other hand, in the topological regime when $\rho > 0$, the system supports an edge state [2]. In the bulk of the state space, the system would similarly go through local clockwise cycles via external transitions γ_{ex} , interspersed with internal transitions γ_{in} which are now slower. However, once the system reaches the edge, it will continue around the edge, which can be verified by inspection in Fig. 1c (also see Supplementary Movie). At long times, the steady-state distribution will hence lie on the system edge, forming a global current that flows counterclockwise along the boundary of the lattice; see Fig. 1d.

3) *Is the entropy in the paper defined only for their model, or for general ones? May be restructure the paragraph around eq. 3 in the paper.*

The expression for entropy production per period ΔS (Eq. 3 in the paper) is applicable to any oscillator model described by a master equation with a non-equilibrium steady state [4, 5]. We have clarified the range of applicability for Eq. 3 in the new manuscript, and put the statement in a more prominent place in the relevant paragraph. We have also included more details on how ΔS behaves for the MWC model. The paragraph is reproduced below:

- In typical oscillator models, coherence can be increased by dissipating more free energy. For a general oscillator model described by a master equation, the free energy cost for maintaining the non-equilibrium steady state under constant temperature can be quantified by the entropy production per period ΔS . Denoting the steady-state probability distribution as \mathbf{p}^s and the oscillation period as \mathcal{T} , ΔS is given by [4, 5]

$$\Delta S = \frac{\mathcal{T}}{2} \sum_{i,j} (p_j^s \mathcal{W}_{ij} - p_i^s \mathcal{W}_{ji}) \ln \left(\frac{p_j^s \mathcal{W}_{ij}}{p_i^s \mathcal{W}_{ji}} \right).$$

In the MWC-type model of KaiC studied in [6], for example, increasing ΔS is necessary to increase coherence. Both quantities increase when the external driving μ is stronger, although coherence starts to decrease when μ is increased still further (see Supplementary Fig. S6). This leads to even worse performance for the oscillator as it maintains less coherent oscillations with increasing energetic cost.

4) In the SI they model with periodic boundary conditions – then there is actually topology, but the results are similar. Why?

The spectral gaps in periodic boundary condition (PBC) and in open boundary condition (OBC) are similar (as shown in Fig. S7b in the revised manuscript) because the OBC stochastic matrix can be seen as a perturbation to the PBC one, where only the transition rates along the edges are changed. Such perturbations turn out to not modify the spectral gaps significantly, as we have numerically verified in Fig. S7b.

We indeed define topology only in periodic boundary conditions. This “bulk” topology is generally related to edge states of the corresponding lattice in open boundary conditions. While thinking about the comparison between PBC and OBC, we realized that we have used the same notation for the spectral gap in PBC and in OBC. We apologize for this possible source of confusion, and have changed the notation such that the spectral gaps in OBC are denoted as Δ_I^o and Δ_R^o instead. We have updated the notations and figures in the SI accordingly. Changes are made in:

- Fig. S7a in the revised manuscript, where Δ_I and Δ_R are replaced by Δ_I^o and Δ_R^o . See Fig. 1 in this document.
- Fig. S7b in the revised manuscript, where the legend “line $y = x$ ” is replaced by “line $\Delta_I^o = \Delta_I$.”
- Figure caption for Fig. S7: “**Spectral properties of \mathcal{W} in open boundary conditions (OBC).**” and “Calculated values for the two gaps (Δ_I in PBC and Δ_I^o in OBC, blue dots) almost all lie on the orange line on which the two gaps are equal.”
- Section VI in the Supplementary Information:
 - Last sentence in the second paragraph: “As shown in Fig. S7a, we define the **OBC real spectral gap Δ_R^o** as the shortest distance from the leftmost band to the imaginary axis and the **OBC imaginary spectral gap Δ_I^o** as the shortest distance from the topmost band to the real axis.”
 - Second to last paragraph: “For the top band, for instance, the imaginary parts of all eigenvalues in the band move upwards in the complex plane by roughly the same distance, keeping the bandwidth constant but increasing the distance between the bottom point in the band to the real axis (which is the definition of Δ_I^o).”
 - Last paragraph: “Nevertheless, when λ_I begins to decrease, it approaches zero more slowly than λ_R and gives rise to a monotonically increasing \mathcal{R} that tracks Δ_I^o (and therefore Δ_I) in ρ (Fig. S8d).”

5) In figure S6, how do they know which states are bulk vs edge?

We thank the reviewer for pointing out the lack of explanation on what we mean by bulk and edge states and how to identify them. We do not impose a binary classification on whether a state is a bulk state or an edge state. Rather, we calculate a continuous quantity $\sum_{i \in \text{Edge}} |\psi_i|^2$, the sum of the magnitude squared for entries i in the corresponding eigenvector that lies on the edge of the system. We normalize each eigenstate to have norm 1 so that the value of the sum varies continuously between 0 and 1. The magnitude of this sum provides information on the extent to which eigenstates are localized on the edge. We apologize for the potentially confusing legend in Fig. S6a (Fig. S7a in the revised manuscript), which was meant to show representative states that are highly localized on the edge or bulk. We have removed the word “state” from the legend, and replaced the points with a color bar to clarify that the sum $\sum_{i \in \text{Edge}} |\psi_i|^2$ lies on a continuous spectrum and the states with a higher sum are more localized on the edge. The above clarifications are

FIG. 1. (Fig. S7 in the manuscript.) **Spectral properties of \mathcal{W} in open boundary conditions (OBC).** **a**, Spectrum of \mathcal{W} in OBC. The real spectral gap is defined as the shortest distance between the left band to the imaginary axis, and the imaginary spectral gap is defined as the shortest distance between the top band to the real axis. The color for each dot corresponds to the value of $\sum_{i \in \text{edge}} |\psi_i|^2$, the sum of the magnitude squared of all entries in the steady-state eigenvector that lie on the edge. This quantity measures the extent to which the corresponding eigenvector is localized on the boundary of the system, and lies on a continuous spectrum between 0 and 1. A redder color means more localization. The solid black arrow points at the eigenvalue $\lambda_1 = -\lambda_R + i\lambda_I$, which determines the coherence $\mathcal{R} = \frac{\lambda_I}{\lambda_R}$. The parameters used are $\mu = 2, \rho = 1$. **b**, The imaginary spectral gaps defined in PBC and OBC are nearly identical when both gaps are large and the bands are well separated. ρ is varied from 0.2 to 4.6 in the plot while μ is fixed at $\mu = 2$. Calculated values for the two gaps Δ_I in PBC and Δ_I^o in OBC (blue dots) almost all lie on the orange line on which the two gaps are equal. Eigenvalues and spectral gaps are given in units of γ_{tot} .

included in the figure captions for Fig. S6 (Fig. S7 in the revised manuscript, reproduced as Fig. 1 in this document) and in the second paragraph of Section VI in the Supplementary Information, reproduced below:

- Spectral gaps can be defined for both PBC and OBC. Taking PBC, we can write the transition matrix in reciprocal space as

$$\mathcal{W}_{\mathbf{k}} = \begin{pmatrix} -\gamma_{tot} & \gamma_{in} + \gamma'_{ex} e^{-ik_y} & 0 & \gamma'_{in} + \gamma_{ex} e^{-ik_x} \\ \gamma'_{in} + \gamma_{ex} e^{ik_y} & -\gamma_{tot} & \gamma_{in} + \gamma'_{ex} e^{-ik_x} & 0 \\ 0 & \gamma'_{in} + \gamma_{ex} e^{ik_x} & -\gamma_{tot} & \gamma_{in} + \gamma'_{ex} e^{ik_y} \\ \gamma_{in} + \gamma'_{ex} e^{ik_x} & 0 & \gamma'_{in} + \gamma_{ex} e^{-ik_y} & -\gamma_{tot} \end{pmatrix}.$$

We obtain the spectral gaps for PBC from the continuous spectrum defined by $\mathcal{W}_{\mathbf{k}}$, as illustrated in Figs. 4a and 4b in the main text. Similarly, we can obtain spectral gaps for OBC by plotting the spectrum of \mathcal{W} in the complex plane, as in Fig. S7a. For each eigenstate ψ , we also calculate $\sum_{i \in \text{edge}} |\psi_i|^2$, the sum over the squared magnitudes $|\psi_i|^2$ for all entries i that lie on the edge of the system. This quantity characterizes how localized the corresponding eigenstate is on the edge. We normalize each eigenstate to have norm 1 so that the value of $\sum_{i \in \text{edge}} |\psi_i|^2$ varies continuously between 0 and 1. A larger sum, or a redder color, indicates more edge localization. As we can see from Fig. S7a, the OBC spectrum displays two circle-like shapes with eigenstates highly localized on the edge and four clusters of eigenvalues, lying on

FIG. 2. (**Fig. 4 in the revised manuscript.**) **predictor of coherence and saturation of a global thermodynamic bound.** **a,b**, Spectrum for the topological model in reciprocal space in imaginary space (**a**) and real space (**b**) for $\mu = 1.5$ and $\rho = 1$. The spectral gaps Δ_I and Δ_R are defined as the difference between the minimum of the topmost band and the maximum of the second band. **c**, The imaginary spectral gap Δ_I closely tracks the imaginary part λ_I of the first non-zero eigenvalue λ_1 , as a function of μ . **d**, Phase diagram for the imaginary spectral gap Δ_I in μ and ρ . Δ_I and coherence (Fig. 3a) track each other monotonically, and both saturate to a maximum value after a rapid increase. **e**, The global spectral bound for a fixed μ is shown by a purple ellipse in the complex plane [7], while points represent the spectra for the transition matrix \mathcal{W} for different oscillator models. The topological model approaches this bound in the limit $\rho \rightarrow \infty$, which is saturated by the most coherent cycle (the unicycle) [7]. Here $\mu = 2$ and other parameters are randomly sampled. In all panels, eigenvalues are given in units of γ_{tot} .

the real or imaginary axis, with eigenstates more localized in the bulk. These four clusters are identified as the “bands” in the OBC case, in analogy with the four bands in the PBC spectrum as shown in Fig. 4a and 4b in the main text. As shown in Fig. S7a, we define the OBC real spectral gap Δ_R^o as the shortest distance from the leftmost band to the imaginary axis and the OBC imaginary spectral gap Δ_I^o as the shortest distance from the topmost band to the real axis.

6) *Why are the imaginary axes for the eigenvalues at $Re[\lambda]=-1$.*

We can understand this by looking at the matrix $\mathcal{A}_{\mathbf{k}}$ (see Appendix B in [2]), defined by $\mathcal{A}_{\mathbf{k}} = \mathcal{W}_{\mathbf{k}} + \gamma_{tot}\mathbb{I}$ for identity matrix \mathbb{I} and $\mathcal{W}_{\mathbf{k}}$ as defined in Eq. (S8) in the Supplementary Information. The symmetries of $\mathcal{A}_{\mathbf{k}}$ results in eigenvalues that are often purely real or purely imaginary [2]. Since $\mathcal{W}_{\mathbf{k}} = \mathcal{A}_{\mathbf{k}} - \gamma_{tot}\mathbb{I}$, the eigenvalues of $\mathcal{W}_{\mathbf{k}}$ correspond to the eigenvalues of $\mathcal{A}_{\mathbf{k}}$ shifted to the left by γ_{tot} in the complex plane. Thus, in units of γ_{tot} the purely imaginary eigenvalues are shifted to $Re[\lambda] = -1$.

Meanwhile, the eigenvalues in Fig. S6a (Fig. S7a in the revised manuscript) correspond to the eigenvalues of \mathcal{W} , which is the transition matrix in OBC. \mathcal{W} can be seen as perturbations from the

transition matrix $\mathcal{W}_{\mathbf{k}}$ in PBC. Therefore, the eigenvalues are still distributed very close to the line $Re[\lambda] = -1$, even though the real parts may not be exactly -1 .

Lastly, while thinking about this question, we also noticed a mistake in Fig. 4b in the revised manuscript. The yellow bands in the subfigure should also lie on the plane defined by $Re[\lambda] = -1$. Fig. 4 has been updated and reproduced here as Fig. 2.

II. REVIEWER 2

The idea of using topology to produce stable cyclic behaviors in an unstable world is an attractive one. We might expect to see this arise if there is a 2D lattice of states, and the dynamics are in a regime which favors the edge. And we might then hope that it will give more stable oscillations than competing mechanisms.

The present paper claims to have such a model, and to produce such stability. The model is for a circadian clock, and it works by drawing a grid of possible phosphorelation states, on which the dynamics prefer to walk around the edge. This echoes what is observed in reality. It is claimed that this state has a high coherence (which is a sort-of Q -value) with moderate "thermodynamic force".

However, I have reservations about the model, about its novelty, and about the presentation. I believe these can be addressed, but will require substantial changes.

We thank the reviewer for their careful reading and constructive comments. We address each comment in detail below.

The model presented is similar to one in an earlier paper (2021 PRX, reference [31], E. Tang et. al.) which I had to read to understand it. But crucially, this variant seems at first to have one more dimension – a 3-dimensional state space, not 2. The boundary of a cube is topologically quite different to that of a square, with no protected 1-cycles. Deep in the description, there might be a constraint introduced which reduces it to 2 dimensions — this took several readings to uncover, and I'm still not sure.

We apologize for the obscure presentation and have worked hard to clarify our model – the reviewer's comments on this score have been extremely helpful. Indeed, the model presented in this paper is actually mathematically equivalent to the model introduced in the PRX paper – both are two-dimensional. However, the state space of the model in the PRX has been re-interpreted towards a more realistic description of KaiABC, so some chemical reactions are no longer present while some new ones have been noted.

Specifically, the two independent variables x and y in the PRX represent the number of phosphorylated KaiC monomers and the number of KaiC monomer conformational changes, while in our paper these two variables are interpreted as the number of T and S phosphorylated monomers, respectively. Only one kind of phosphorylation is considered in the PRX, while we consider two kinds of phosphorylation as both are relevant in the phosphorylation cycle [8]. The conformational change does not occur independently in our model. Rather, it serves to change the internal state of the system and facilitate subsequent phosphorylations, which is, for example, consistent with studies on A-loop conformations on KaiC monomers [9, 10]. We have included a paragraph in Section II of the manuscript that discusses the relationship between our model and the PRX model. The paragraph is reproduced below for convenience of the reviewer:

- Our model in Fig. 1c is mathematically equivalent to the model introduced in [2], both featuring a lattice structure in a 2D state space. However, when applying to the KaiABC system, the state space of the model in [2] has been re-interpreted towards a more realistic description. In turn, some chemical reactions are no longer present while others have been noted. Specifically, in [2] the external variables x and y represent the number of phosphorylated KaiC monomers and the number of monomer conformational changes, while in our paper these variables are interpreted as the number of T and S phosphorylated monomers. Rather than occurring independently, the conformational change is instead hypothesized to change the system internal state and prime subsequent phosphorylations. This is consistent, for example, with studies on A-loop conformational changes on KaiC monomers [9, 10]. We also include interactions with KaiA and KaiB molecules as possible biological reactions that change the internal state of the model.

This now seems like the central question for applying topological ideas to state space: How many dimensions are there?

We have tried to explain our model more explicitly above and in the revised manuscript, and hope it is now clear that there are just two dimensions. More generally, there could be other models in other dimensions, but that is a direction for future work.

In solid state physics, the number of dimensions is that of physical space: 3 for a block of metal, 2 if it's really thin (or a magnetic field strongly separates 1 from the rest). The edge states literally live on edges you can touch with your finger.

Here, the topological state is instead a particular walk in a molecular state space. It was claimed in the earlier paper that this state space should have 2 bounded dimensions (a phosphorelation level, and a conformational change) as well as 1 circular one (some kind of ligation state). This paper expands the list to include 1 more bounded dimension, another phosphorelation level. The second phosphorelation state seems to have better experimental grounding than the cyclic ligation state. Perhaps one could go further, and include other details. Would it still work? Is the topological state generic? Or does it rest completely on a hard constraint to exactly lock the conformational state to one of the phosphorelation levels, to remove one bounded dimension?

This question is what I feel the paper ought to address. I'm not quite sure how. If the answer for now is that the dimension is fixed to exactly 2 bounded dimensions by hand, then this needs to be stated very clearly as a major assumption. If 2 dimensions is an emergent property, then how that works sounds interesting.

Yes, we are glad that the reviewer is in agreement that the second phosphorylation state seems to have better experimental grounding than the cyclic ligation state (internal states). We have included more experimental evidence for the internal cycles in the second to last paragraph of Section II: “We also propose directed cycles of internal transitions as in Fig. 1b based on experimental results relating A-loop conformations with KaiA and KaiB. For example, A-loop exposure promotes KaiA binding [9] and prohibits KaiB binding [11] while A-loop burial prohibits KaiA binding and promotes KaiB binding [11].” As discussed earlier, the state space still has 2 bounded dimensions (T and S phosphorylation levels). The conformational changes are re-interpreted as internal changes. One can include other details, and our model is expected to be robust to changes up to an extent because of its topological nature. In Section I of the Supplementary Information (now titled “Robustness of oscillations in our model”), we add an analysis of the spectral gaps and coherence of the model and associated Fig. S2 (reproduced in this document as Fig. 3) under random perturbations to transition rates. The topological state is not generic to any perturbations but to a finite amount of perturbations. The analysis is reproduced below:

- The global oscillations in our model are also robust to perturbations in transition rates. The changes in rates can come from changing external conditions like concentrations of KaiA and KaiB or setting more biologically realistic transition rates for each reaction. They can also come from incorporation of additional KaiC reactions hitherto unconsidered. For instance, ATP association with the nucleotide binding site plays the same role as KaiA interaction in promoting T phosphorylation [12]. This reaction can be combined into the $S \rightarrow E$ internal transition, which results in an effective rate for the combined reaction that is larger than the rates for either reactions. To study the effects of such modifications, we make the rates non-uniform in different directions by multiplying each of $\gamma_{ex}, \gamma'_{ex}, \gamma_{in}, \gamma'_{in}$ in each direction (N,S,E,W) by a different random scaling factor f , taken from a normal distribution with mean 1 and standard deviation ϵ . We look at the real and imaginary spectral gaps and coherence of the resulting model, averaged over different random configurations of transition rates. As shown in Fig. S2, the spectral gaps do not close for ϵ up to 80%, so the network is still in the topological regime. In general, the steady state is still localized on the edge

FIG. 3. (Supplementary Fig. S2 in the manuscript.) **Spectral gaps and coherence for randomized transition rates.** **a**, Each transition rate ($\gamma_{ex}, \gamma'_{ex}, \gamma_{in}, \gamma'_{in}$) in the model is multiplied by a different random scaling factor f in each direction (N,S,E,W). f is taken from a normal distribution with mean 1 and standard deviation ϵ . In the case when a negative value is sampled, we take $f = 0$. Here we plot the real spectral gap Δ_R , averaged over different random realizations of transition rates, as a function of the standard deviation ϵ for randomization of the transition rates. The randomized rates repeat along the x and y axis. For example, any eastward γ_{ex} transition has the same rate after randomization regardless of the (x, y) coordinates. **b**, Average imaginary spectral gap Δ_I as a function of ϵ . **c**, Average coherence \mathcal{R} as a function of ϵ . For all panels, we average over 500 random configurations for each ϵ . The same configurations are used to calculate all three quantities. The yellow shaded area represent one standard error. The initial parameters for the model before multiplying by f are $\mu = 3, \rho = 5$.

of the state space, even though it can have a higher probability in a particular edge, e.g., the left edge, because of the non-uniform transition rates. The coherence of the oscillation is attenuated but some degree of oscillation persists. The robustness of the oscillations obtain from its topological nature, and we expect it to remain robust, up to some extent, for other types of perturbations such as additions of long-range interactions that connect nonadjacent states in the state space.

We point to these new results in the sixth paragraph of the main text: “The edge state further inherits the useful property of topological protection from inaccessible states [2] or perturbations in transition rates (see Supplementary Fig. S1 and S2).”

In addition to perturbations to transition rates, our model is also expected to be robust, up to an extent, to other modifications such as long-range interactions that connect nonadjacent states on the lattice. The topological state does not rest completely on our hard constraint on the conformational state – we model it as such here for simplicity in this most basic model that we present, but the robustness should extend to even when it’s not a hard constraint, similar to the robustness exhibited to perturbations above. Hence, even if the biological space is larger than 2d but has chemical reactions that are not essentially changing the dynamics, the dynamics can effectively be represented in a 2d space. We do see this kind of effective low-dimensional dynamics in KaiABC and other biological systems [13–15]. In this sense, 2 dimensions is an effective dimension and hence emergent.

Aside from this major question, I think that the presentation should be more straight-forward. At present it muddles together what is actually done with all sorts of analogies, and is unclear about the differences from earlier work. I have more detailed comments below but the big three are:

1). *The model needs to be clearly described on its own terms (i.e. as a mathematical model). The complete list of what specifies its state, and the complete list of parameters, must be written down clearly, at once, in the main paper. Not trickled in over 4 pages, not in the supplement. The motivating analogy to biological mechanisms is also interesting, but should never be used to obscure what’s actually being done.*

We appreciate the reviewer’s suggestions on improving the clarity of the manuscript. We have rewritten Section II of the manuscript which introduces the model, spelling out the model details and parameters all at once. However, because our mathematical model is completely equivalent to that in the PRX, we feel that it is repetitive to some degree to only introduce the model abstractly before going into the biology. We have instead introduced the model in the context of the KaiABC system from the beginning, rather than introducing hypothetical biological variants of the model before discussing KaiABC (which can be confusing as the reviewer suggests). We focus on the KaiABC system because part of the new contribution of our paper is providing a more biologically accurate interpretation of the model when it is applied to KaiABC. To give a coherent account of the model in terms of KaiABC in the beginning, we have made the following changes:

- Remove the paragraph “To propose an alternative model, ...” that maps the model to the KaiABC system only in the latter half of Section II. Move the discussion on the nature of KaiA interaction (line 239-243) to the second paragraph in Section II where the internal transitions such as KaiA interactions are introduced.
- Modify the first sentence in the paragraph “Thus far, we have been ...” and turn to discussions of experimental results on KaiABC: “Experimentally, it is well established that KaiC molecules exhibits 24-hour cycles via a concerted phosphorylation sequence in the presence of KaiA and KaiB [16]. As illustrated in Fig. 2a, during the day, all six KaiC monomers get phosphorylated at the T-sites, ...”
- For the second to last paragraph in Section II, we
 - Add a topic sentence: “Our topological model presents an alternative way to account for the experimental facts and explain the emergence of the KaiC phosphorylation cycle.”
 - Move the discussion on KaiC monomer phosphorylation cycle from the previous paragraph (line 227-230) to here.
 - Add experimental support for the internal reaction cycles, in response to a later comment.
 - Move the explanation on the number of different conformations for a state $(x, y)_s$ (line 257-267) to the end of the second paragraph in Section II, where internal transitions are introduced.
 - The modified paragraph is reproduced here: “Our topological model presents an alternative way to account for the experimental facts and explain the emergence of the KaiC phosphorylation cycle. Based on experimental evidence for KaiC monomer autophosphorylation [17, 18], we hypothesize directed monomer reaction cycles of phosphorylation and dephosphorylation as illustrated in Fig. 1a. We also propose directed cycles of internal transitions as in Fig. 1b based on experimental results relating A-loop conformations with KaiA and KaiB. For example, A-loop exposure promotes KaiA binding [9] and prohibits KaiB binding [11] while A-loop burial prohibits KaiA binding and promotes KaiB binding [11]. In addition, our hypothesis that KaiA and KaiB preferentially interact with different phosphorylation configurations is consistent with experimental evidence that activation by KaiA is incompatible with the configuration of KaiC that triggers KaiB binding [18]. In particular, we propose that KaiA preferentially interacts with KaiC during T-site phosphorylation rather than S-site phosphorylation, and the converse for KaiB, which agrees with observed data [19]. Indeed, experiments indicate that KaiB binding (and possibly its unbinding), with rates γ_{in} in our model, are slow processes [8, 18, 20, 21] compared to phosphorylation reactions (with rates γ_{ex} in our model) [22] – consistent with our assumption that there is a separation of timescales.”

- To be clear about what are the system parameters we are using, we move the discussion of the parameters μ and ρ from Section III to Section II and introduce the parametrization together with the model.

To not muddle together the abstract model with the biological interpretation, when introducing each component of the model we always describe the model mathematically first before discussing possible interpretations for KaiABC. We hope this provides a clearer explanation of the model. The paragraphs introducing the model in the beginning of Section II is reproduced below:

- We consider discrete stochastic processes that operate in a two-dimensional configuration space. The state of the system is completely specified by three variables $(x, y)_s$. The “external” variables x and y are independent dynamical variables. Based on the widespread presence of non-equilibrium cycles in biological systems [23, 24], we propose that the “external” transitions modifying these variables form reaction cycles with transition rates γ_{ex} and slower reverse rates $\gamma'_{ex} \ll \gamma_{ex}$ (black solid arrows), as shown in Fig. 1a. For the KaiABC system, x and y correspond to the number of phosphorylated T and S sites, two residues that can independently phosphorylate on each monomer of the hexameric KaiC molecule [25, 26]. Phosphorylation of these two sites are represented by orange and pink circles in Fig. 1, respectively. In our model, we only keep track of the number of T and S phosphorylated monomers, while the specific location of each monomer does not matter.

The “internal” variable s , given by compass directions N-E-S-W, labels the internal state of the system that primes it to go through one of the four possible external transitions next. For example, in the W state, the system is most likely to go through the westward external transition that decrements x . Within each phosphorylation level (x, y) , the system can take any of the four possible internal states N, E, S or W that prime each possible external transition. As shown in Fig. 1b, these internal states are also assumed to transition between each other in a cyclic manner, with internal transition rates γ_{in} and slower reverse rates $\gamma'_{in} \ll \gamma_{in}$ (gray dashed arrows). Unlike the external cycles above, the internal cycles typically do not correspond to reaction cycles in biological systems. For the KaiABC system, the internal transitions could correspond to interactions with KaiA and KaiB proteins [19] and conformational changes of a loop-like structure on KaiC called the A-loop (two conformations represented by blue squares and purple circles in Fig. 1 respectively) [9]. These transitions modify the internal state of the system, changing its tendency to phosphorylate or dephosphorylate and whether in T or S. In particular, we assume that the monomer conformational changes occur after S-site phosphorylation of the same monomer. It follows that the states $(x, y)_E$ and $(x, y)_S$ (for $y > 0$) in our model have $y - 1$ circles and $7 - y$ squares, while the states $(x, y)_W$ and $(x, y)_N$ have y circles and $6 - y$ squares. We represent the internal transition associated with KaiA with a curly arrow, since KaiA promotes phosphorylation through rapid association and dissociation, activating a larger stoichiometric amount of KaiC as compared to the typical one-on-one binding like KaiB [18].

These reactions are repeated for each monomer and hence can be laid out as a lattice. As shown in Fig. 1c, the phosphorylation/dephosphorylation cycles in Fig. 1a (green box) and the internal cycles in Fig. 1b (blue box) both repeat along the x and y axes of T and S phosphorylation. Such a lattice will have edges representing the physical constraints of the system, i.e., $0 \leq x \leq N_x$ and $0 \leq y \leq N_y$. In our case, $N_x = N_y = 6$ since there are 6 sites available on a KaiC hexamer for either T or S phosphorylation.

Our model in Fig. 1c is mathematically equivalent to the model introduced in [2], both featuring a lattice structure in a 2D state space. However, when applying to the KaiABC system, the state space of the model in [2] has been re-interpreted towards a more realistic description. In turn, some chemical reactions are no longer present while others have been noted. Specifically, in [2] the external variables x and y represent the number of phosphorylated KaiC monomers and the number of monomer conformational changes, while in our

paper these variables are interpreted as the number of T and S phosphorylated monomers. Rather than occurring independently, the conformational change is instead hypothesized to change the system internal state and prime subsequent phosphorylations. This is consistent, for example, with studies on A-loop conformational changes on KaiC monomers [9, 10]. We also include interactions with KaiA and KaiB molecules as possible biological reactions that change the internal state of the model.

Before analyzing the model, we simplify our parametrization for the four transition rates γ_{ex} , γ'_{ex} , γ_{in} , γ'_{in} down to three parameters. First, μ is the thermodynamic force defined by $e^{\mu/k_B T} \equiv \gamma_{ex}/\gamma'_{ex} = \gamma_{in}/\gamma'_{in}$. For an arbitrary cycle in the network, the sum of μ along each transition in the cycle is the energy input into that cycle from external driving such as ATP hydrolysis [27]. Detailed balance occurs at $\mu = 0$ while out-of-equilibrium is when $\mu > 0$; we analyze μ in units of $k_B T$. Here we assume the same μ for every transition in the model, which removes one free parameter. Second, ρ varies the ratio of external to internal transitions, defined by $e^\rho \equiv \gamma_{ex}/\gamma_{in} = \gamma'_{ex}/\gamma'_{in}$. It quantifies the separation of timescales between the external and internal transitions. Third, γ_{tot} controls the overall timescale of all transitions, i.e., $\gamma_{tot} \equiv \gamma_{ex} + \gamma'_{ex} + \gamma_{in} + \gamma'_{in}$.

The reviewer is right to point out that our introduction of system parameters is scattered around the manuscript. However, we have introduced every parameter in the main text and not in the SI as the reviewer may be suggesting. The parameters of the model only include μ, ρ, γ_{tot} , and we have moved their introduction to Section II when we discuss the model.

2). *In particular, states ABCD need to be re-named, to avoid confusion with KaiA, KaiB, KaiC. (Perhaps compass directions N-E-S-W would be natural, or just numbers.) This cyclic 4-state system is important to the model, and deserves its own sub-figure. If there is some real cyclic 4-state system, any discussion of the analogy must start off with clearly distinct names for the things it wishes to compare. Similarly, Fig 1(e) needs to be moved out to be an independent figure with its own caption, as it's describing the real system, not the model.*

The reviewer is correct that the current labeling could be potentially confusing. We have relabeled the internal states to N-E-S-W following the suggestion of the reviewer, and added a simple “compass” between Fig. 1a and Fig. 1b in the main text to define directions. The internal labels represent the direction of the next most probable external transition primed by the internal state. Here we show the mapping from the original labels to the new labels: $A \rightarrow N, B \rightarrow E, C \rightarrow S, D \rightarrow W$.

The reviewer is right that the assumption of a cyclic 4-state system is important to the model. However, this 4-state system (with its own subfigure Fig. 1b) does not correspond to real biological systems in isolation. A complete internal cycle rarely occurs in the topological regime $\rho \ll 0$ ($\gamma_{ex} \gg \gamma_{in}$), as an internal transition is much more likely to be followed by a faster external transition rather than by another internal transition. The internal state can be better thought of as a tag that signals what the next external transition is. We also add a clarification in the second paragraph of Section II: “Unlike the external cycles above, the internal cycles typically do not correspond to reaction cycles in biological systems.” Also, following the reviewer’s suggestion, we have split up Fig. 1 so that Fig. 1e and 1f now make up their own figure, which is Fig. 2 in the revised manuscript. The figure is reproduced in this document as Fig. 4.

3) *The exact relation to the earlier model (2021 PRX) must be described clearly. This one introduces extra complexity (in the name of realism?) in a way which looks like it may destroy the good properties. But perhaps extra constraints are added, to restore them. Is the final result mathematically equivalent (up to a change of co-ordinates) or not? If not, would the old one look different on Figure 2, say? (Then it should probably appear on most plots.) The fact that this is a variant of an earlier model needs to be clearly stated in the introduction.*

FIG. 4. (**Fig. 2 in the manuscript**) **Phosphorylation cycle of the KaiABC system, reproduced by the topological model.** **a**, KaiABC exhibits oscillations via a concerted global cycle of phosphorylation and dephosphorylation. During the day, all six KaiC monomers get phosphorylated at the T-sites, and then at the S-sites. **This phosphorylation phase is promoted by interaction with KaiA molecules [28].** By night, **fully phosphorylated KaiC binds to KaiB, which sequesters KaiA from the solution.** In the absence of KaiA, all the T-sites get dephosphorylated, followed by the S-sites [8]. **The A-loop tends to bind KaiA in the exposed state and not bind with KaiA in the buried state [9].** Since individual monomers can independently phosphorylate and shuffle between hexamers [17, 18], why do they perform a concerted phosphorylation cycle that is robust? **b**, In the topological regime, a global cycle emerges similar to the experimentally observed phosphorylation sequence. **Here we only show one representative state (out of four) for each phosphorylation level (x, y) along the edge.**

Following the reviewer’s suggestion, we have added a paragraph in Section II that clarifies the relationship between the current model and the one in the PRX, as discussed previously. Our model is mathematically equivalent to the model in the PRX up to a reinterpretation of coordinates. We include more biological details here when specializing to the KaiABC system for the sake of realism. This is why we presented the model not just as an abstract mathematical model but also interpreted in the context of KaiABC. We have redrawn the cycles in a different way from the PRX to illustrate what the underlying biophysical mechanisms could be.

Apart from providing a more biologically accurate interpretation of the model for KaiABC, in this paper we also expand on how global oscillations, attractor cycles, and longer time scales can emerge from the topological model, which only constitute a small part of the PRX paper. We characterize how robust the oscillations are and evaluate the performance of such oscillators in terms of energy dissipation with the tools of nonequilibrium thermodynamics. This is the novelty of our work.

What follows is a description of how the model works. Below that are more detailed presentation comments.

///// 2021 model

To understand this paper’s model I had to read an earlier paper. My understanding of how that works is as follows:

The state is specified by (x, y) in a compact subset of Z^2 (the integer plane), together with what I’ll call s in Z_4 (integers modulo 4), whose elements are named A, B, C, D in the paper. The complete state is written with a subscript for s , like $(1, 2)_C$.

The dynamics always moves to a nearest neighbor in both (x, y) and in s , with s cyclic. The position (x, y) is rigidly coupled to s as follows:

*** If s is decremented, then (x, y) does not change. This is called an "internal" transition.*

*** If s is incremented, then (x, y) must change in a particular compass direction, determined by s , clockwise in the plane, N-E-S-W. Thus increment $s=1 \rightarrow s=2$ goes with $y \rightarrow y+1$, North, while $s=2 \rightarrow 3$ goes with $x \rightarrow x+1$, East, and so on. Such transitions are called "external". If the resulting*

(x,y) is outside the allowed subset, then the transition is impossible, s cannot be incremented.

The essence of the topological state is this: After an "external" step along a straight boundary of the allowed (x,y) region, with the disallowed region to the right, the next "external" will be impossible: The clockwise increment would make it point into the wall. Thus an "internal" step is the only possibility, and brings s back to what it was, allowing another "external" step in the same direction. This is most likely when "internal" moves are rare, i.e. when the rate constant for "external" is much larger. Only in this limit is the state topological.

Section IV of the 2021 paper proposes this system as a model for circadian rhythms, the KaiABC system. This involves a hexamer i.e. a complex of 6 similar proteins, monomers. Each has two conformational states, and the mapping is that $x = 0, 1, \dots, 6$ counts how many of the monomers are in one conformation (ignoring order), while $y = 0, 1, \dots, 6$ counts how many of them are phosphorelated (ignoring which ones).

What is attractive is that the phosphorelation states are not constrained to be occupied in some order; instead, the fact that this happens is an emergent property of the edge state, which walks the boundary of the square in (x,y) state space. That the internal state s is cyclic is, however, put in by hand. This cyclic choice of compass direction of the allowed "external" step is what produces the topological state.

The reviewer's understanding is correct.

////// Changes in the 2023 model

The goal of the present paper seems to be to elaborate this model, to make it more realistic.

The principal change made is to allow for two kinds of phosphorelation sites, T and S , instead of one. They map these to the mathematical model's x and y parameters. However, they also still include the conformation state, giving another variable which I'll name $z = 0, 1, \dots, 6$, the "number of diamonds" in the figures. Thus it appears that the state is now $(x,y,z),s$, with (x,y,z) lying in the cube $\{0, 1, \dots, 6\}^3$. The cube's boundary is topologically quite different, it does not have an incontractable 1-cycle, so at first glance it seems no previous results should be expected to survive.

However, three other changes are introduced:

**1. When s is decremented, then (x,y) does not change, but z sometimes must change: $B \rightarrow A$ increments z , while $D \rightarrow C$ decrements z , other steps do not change it. This is still called "internal", but is presumably forbidden when z is 6 or 0 respectively. With this change alone (from the model above) some states could never be escaped.

As mentioned previously, the conformational change the reviewer dubs z is not assumed to be an independent variable throughout this paper. Rather, changes in z is encoded in changes in the internal state s , as the conformational change is an internal process that primes subsequent phosphorylation.

Also, we realized that we have been using both "squares" and "diamonds" to refer to the same shape that represents the exposed state of the A-loop. We clean up notation and always call these objects "squares" in the revised manuscript. "Diamonds" have been used at the end of the second to last paragraph in Section II, when we discuss the number of different conformational states for a generic state $(x,y)_s$. They have been replaced with "squares".

**2. These two reactions are also allowed to run backwards, with smaller rates (gamma-prime). I believe there are now 4 independent rate constants (although the figure confusingly labels some arrows with hypothesized KaiB relations, instead of rates).

The reviewer is right that there are four independent rate constants $\gamma_{ex}, \gamma'_{ex}, \gamma_{in}, \gamma'_{in}$ to start with. However, we also impose a constraint when we introduce a new parametrization for the system

such that there are only 3 independent parameters μ, ρ, γ_{tot} . Only these 3 parameters are used throughout the paper. We have moved the paragraph discussing μ, ρ , and γ_{tot} to Section II and explicitly stated the number of parameters involved. This paragraph is reproduced below.

- Before analyzing the model, we simplify our parametrization for the four transition rates $\gamma_{ex}, \gamma'_{ex}, \gamma_{in}, \gamma'_{in}$ down to three parameters. First, μ is the thermodynamic force defined by $e^{\mu/k_B T} \equiv \gamma_{ex}/\gamma'_{ex} = \gamma_{in}/\gamma'_{in}$. For an arbitrary cycle in the network, the sum of μ along each transition in the cycle is the energy input into that cycle from external driving such as ATP hydrolysis [27]. Detailed balance occurs at $\mu = 0$ while out-of-equilibrium is when $\mu > 0$; we analyze μ in units of $k_B T$. Here we assume the same μ for every transition in the model, which removes one free parameter. Second, ρ varies the ratio of external to internal transitions, defined by $e^\rho \equiv \gamma_{ex}/\gamma_{in} = \gamma'_{ex}/\gamma'_{in}$. It quantifies the separation of timescales between the external and internal transitions. Third, γ_{tot} controls the overall timescale of all transitions, i.e., $\gamma_{tot} \equiv \gamma_{ex} + \gamma'_{ex} + \gamma_{in} + \gamma'_{in}$.

As discussed previously, we draw the cycles in a different way, which incorporate the biological interpretations of the model, to avoid repeating the same model and figures introduced in the PRX paper, and to better connect abstract transitions to experimentally observed biophysical mechanisms. The arrows are labeled with molecular reactions as suggestions for what the transition rates could correspond to when applied to a biological system.

***3. Finally, a few pages later, there may or may not be another connection between z (the “number of diamonds”) and y . If I have parsed this correctly, it seems to say that when $s=B$ or C , they fix $z=y-1$, while when $s=A$ or D , instead $z=y$. (Page 5, line 262, 2nd last paragraph before section III.) But this contradicts Figure 1a.*

If z is the number of diamonds, then at B or C (E or S in the new notation) we have $z = 7 - y$ for $y > 0$, while at A or D (N or W) we have $z = 6 - y$, as noted in the sentence pointed out by the reviewer. This should be consistent with Fig. 1a. We have moved this specification to the second paragraph in Section II for a clearer introduction of the internal states.

At present it's unclear to me whether the end z (conformation) is an independent dynamical variable, or not. If not, the complete state of the system is specified by $(x, y)_s$ as before.

z is not an independent dynamical variable. Changes in s capture all the information about changes in z . The system state can be completely specified by the three variables $(x, y)_s$.

So the new paper adds more complexity to the model's state, in a way which changes the topology, and also adds more complexity to the dynamics. The desired outcome seems to be only not to break the old model's edge states – to pick a regime in which this 4-parameter model has the same emergent behavior as the previous 2-parameter one. This is the principal technical change and its motivation, and consequences (or lack thereof) seem to be barely discussed.

(The 2021 paper eventually allows point 2 too, but the basic concept can be grasped without it.)

As noted earlier, mathematically our model is the same as that in the PRX paper with reversible rates. Both models have 3 free parameters, (c, r, γ_{tot}) in the PRX and $(\mu, \rho, \gamma_{tot})$ in our paper. The added complexity only fills in more details as to what the system states in the network can represent when applied to the KaiABC system. Because the models are mathematically equivalent, all discussions on the model with reversible rates in the PRX paper also apply to our model. The same emergent behavior of edge dynamics follows. Here, we are investigating a specific aspect of the model behavior, the global oscillations, which is only touched upon briefly in a subsection of the PRX article. We expand the analysis to analyze the factors affecting oscillation coherence and evaluate oscillator performance in terms of energy dissipation.

Are there really 4 rate constants? Later on page 5 (line 324, “Before analyzing...”) some equations write extra constraints, such that there are only 3.

The reviewer is right that because of the constraints that accompany the introduction of μ , there are really only 3 independent parameters. We have clarified this point in the paragraph that the reviewer refers to, and moved the paragraph to Section II so that the parametrization is introduced along with other model components in the beginning.

///// Presentation comments

The writing is polished, but it is not easy to see where the contribution starts. I feel it should spend less time on what is hoped for, and more time on what is actually done.

The introduction can be shorter. That would make space for saying by what measures success will be judged.

We have split up the second to last paragraph in the Introduction section of the original manuscript, to put discussions on what we have done in its own paragraph. We also add more details in that paragraph to explain our results. The paragraph is reproduced below:

- **In our model**, we formulate topological methods in the context of stochastic systems and non-equilibrium thermodynamics to quantify the emergence of global cycles that naturally describe biological oscillations. We illustrate these methods in the context of the KaiABC system that regulates the circadian rhythm of cyanobacteria [29]. Following [6], we define coherence \mathcal{R} based on the damped oscillations of the decaying eigenmodes to quantify robustness of the emergent oscillations. With only a few free parameters, our model supports a dynamical regime with global currents localized on the edge, which has a topological origin [2]. Such localization leads to a novel regime with increasing coherence alongside decreasing entropy production as the external driving is increased. This unusual behavior, i.e., “getting more from pushing less”, has been observed in other scenarios such as driven electronic, Brownian, and glassy systems [30–33]. Its occurrence in a stochastic system like ours remains surprising and, as we discuss later, reflects the topological nature of our model. Moreover, the model also exhibits high coherence compared to other models of KaiABC. Further, we introduce spectral gaps as a predictor of coherence, and examine the saturation of global spectral bounds.

We have also added a paragraph in the introduction that discusses predictions generated by the characteristic behavior of our model that can be tested experimentally. These are new predictions we have come up with after more thought and discussions with experimentalists in the field. Detection of these signatures should provide good evidence for our model, since such characteristic behavior like transient spikes in energy consumption or negative correlation between external driving and energy dissipation are not observed in any other models we analyze. The paragraph is reproduced here

- **The characteristic behavior of our topological model leads to several experimentally testable predictions.** For instance, by tuning the ATP concentration that changes the external driving on the system, one can probe the regime where stronger driving leads to increasing oscillation coherence but decreasing total energy dissipation (measurable as total ATP consumption or ADP production). Moreover, perturbation from the edge cycles or initialization in the bulk of the state space leads to transient spikes in energy dissipation due to dissipative phosphorylation cycles that make up the state space. Identifying these signatures by tracking the energy dissipation provides a test for whether topology as illustrated in our model contributes to the robustness of circadian oscillations.

These new experimental predictions have also been added to the list of predictions we had in the second paragraph in Section IV of the manuscript. To avoid repeating everything twice, we have

only put in the introduction those predictions that are more unique to our model and more easily measurable. The second paragraph in Section IV is reproduced here, which contains the whole list of experimental predictions:

- We further predict testable signatures of topological edge states in biological oscillators and the KaiABC system. Examining our steady-state distribution (Fig. 1d), we expect to find mixtures of molecules with specific fractions of differentially phosphorylated subunits, where such mixtures have been experimentally observed [19]. Here, KaiB-bound KaiC should be less abundant than unbound KaiC, because the slow γ_{in} transitions cause a buildup of the latter before they bind to KaiB. Moreover, new innovations in calorimetric techniques for the heat fluxes involved in the cell cycle could estimate the amount of energy dissipation in the production of oscillations, which intriguingly highlight the strong contribution of non-equilibrium cycles [34], similar to our hypothesis. **By changing the ATP concentration or ATP/ADP ratio, we can tune the thermodynamic force μ into the regime where ΔS has a negative slope in μ (Fig. 3c). In this regime, we would measure an increasing energy dissipation as ATP concentration is decreased, a unique feature of our topological model. Lastly, by perturbing the system or initializing it in the bulk of the state space with partially phosphorylated T and S sites, we should see a transient increase in ATP consumption or energy dissipation as the system goes through rapid cycles of external transitions within the system bulk before converging to the edge. Similar spikes in energy dissipation are also expected when the system occasionally wanders into the bulk by chance from its typical trajectories along the edge.** Monomer phosphorylation cycles in the reverse direction from the global cycle may also be observed, as the external cycles in the system bulk have the opposite chirality from our global cycle on the system edge (see Fig. 1c).

Moreover, we modify the abstract to better reflect the contribution of our paper, i.e., we propose a topological model with parsimonious parameters that reduce the system dynamics to lower dimensions and generate long and stable timescales (oscillations) from a large phase space of faster reactions. The abstract is reproduced below:

- **Long and stable timescales** are often observed in complex biochemical networks, such as in emergent oscillations. How these robust dynamics persist remains unclear, given the many stochastic reactions and shorter time scales demonstrated by underlying components. We propose a topological model **with parsimonious parameters that produces long oscillations** around the network boundary, effectively reducing the system dynamics to a lower-dimensional **current**. Using this to model KaiC, which regulates the circadian rhythm in cyanobacteria, we compare the coherence of oscillations to that in other KaiC models. Our topological model localizes currents on the system edge for an efficient regime with simultaneously increased precision and decreased cost. Further, we introduce a new predictor of coherence from the analysis of spectral gaps, and show that our model saturates a global thermodynamic bound. Our work presents a new mechanism for emergent oscillations in complex biological networks **utilizing** dissipative cycles to achieve robustness and efficient performance.

This point has also been emphasized more in the first paragraph of the Discussion section, reproduced below:

- We have proposed a **parsimonious** topological model that generates coherent oscillations, which supports an unusual regime with increased coherence and simultaneously decreased energetic cost. We find that the imaginary spectral gap can be used to predict the oscillation coherence. Applied to the KaiABC system, our model has high coherence compared to other KaiABC models and more closely saturates a global spectral bound, similar to the most coherent unicyclic models. Further, the kinetic ordering of the KaiABC phosphorylation cycle arises naturally as an edge current in our model. In contrast to typical MWC-type models,

it does not require restriction of the configuration space to all-or-none conformational change or fine-tuning of the reaction rates. With only a few parameters, our model can generate the experimentally observed phosphorylation sequence.

Is the metric Q -value per energy consumption, or what?

The coherence (Q -value) is proportional to the number of coherent oscillations in the correlation function before it decays to steady state (see Fig. S3, originally Fig. S2). Its definition does not involve energy consumption explicitly, although its magnitude is intimately related to the energy consumption involved, as more energy consumption generally leads to a higher coherence. To compare performance of different oscillators, we therefore need to take into account energy consumption, quantified by the entropy production per period ΔS in our paper. However, it is difficult to directly compare the performance of existing models and their variations one-on-one. In this paper we attempt to evaluate and compare the performance for various simplified models, by looking at the cost-precision tradeoff $\Delta S/\mathcal{R}$ [35] and how closely the models saturate their respective thermodynamic bounds. The inverse of the tradeoff $\Delta S/\mathcal{R}$ is the coherence per energy consumption suggested by the reviewer.

Is “thermodynamic force” just some parameter combination or something real?

The “thermodynamic force” has a physical interpretation. Going around an arbitrary cycle in the network, the sum of μ along each transition in the cycle is the energy input into that cycle from external driving such as ATP hydrolysis [27]. The following sentence has been added to the paragraph introducing μ , which is now in Section II of the manuscript: “For an arbitrary cycle in the network, the sum of \$\mu\$ along each transition in the cycle is the energy input into that cycle from external driving such as ATP hydrolysis [27].”

I think it is confusing to first say that (x,y) could be anything (line 133, “They could also represent the state of a biopolymer assembled...”) before describing this paper’s actual model. The model is confusing enough on its own, without hypothetical variants. This can be reversed — point out in the conclusion that much the same mathematics could perhaps be mapped to other systems.

We thank the reviewer for the feedback on the clarity of the manuscript. In Section II, we have now introduced (x,y) in the context of KaiABC only, without the hypothetical variants. Possible applications to other models are now briefly discussed Section IV at the end of the article, where we have added in the last paragraph that “Even though we have mostly discussed our model in the context of the KaiABC system, the model is more general. It can be mapped to various other biological systems where \$x\$ and \$y\$ represent other possible types of molecular modifications such as polymerization [2].”

///// Fig 1 / section 2

Figure 1 introduces the model. It is, I think, impossible to parse without reading the 2021 paper. By my count there are 10 different meaningful colors, 5 distinct kinds of arrows (with many rate constants), and 5 differently shaped objects, in panels (a) and (b) alone. It is fantastically hard to puzzle out the correspondence between these pictures and the caption text below.

We agree with the reviewer that the large number of colors and objects may make it difficult to understand the picture with the given caption. We have simplified the picture and also provided more detailed explanations on the meanings of these colors and objects in the text and figure caption. For simplicity, We have adopted the same color for the four different internal states in Fig. 1. The modified figure and its caption are reproduced here as Fig. 5 in this document. To make the introduction of the model more coherent, we have also moved some specifications of

FIG. 5. (Fig. 1 in the manuscript.) **Topological model for emergent oscillations, illustrated with KaiABC that regulates the circadian rhythm.** **a**, Based on the prevalence of non-equilibrium cycles observed in various biological systems [23, 24], we hypothesize that KaiC monomers undergo phosphorylation and dephosphorylation cycles (black arrows γ_{ex} , and slower reverse transitions γ'_{ex} , where two types of phosphorylation T and S are shown with the addition of orange and pink circles and the numbers of each are given in brackets). We only keep track of the total phosphorylation levels in T and S, while the location of each phosphorylated monomer can be arbitrary. **b**, Within a given phosphorylation level, internal transitions (grey arrows, γ_{in} and γ'_{in}) can take place due to conformational changes (illustrated by circle and square shapes) or ligand binding (e.g., of KaiA or KaiB). The internal state labels the direction of the next external transition that it primes. We represent the interaction with KaiA with a curly arrow because KaiA promotes phosphorylation by rapid association and dissociation rather than binding to fixed KaiC hexamers [18]. In both **a** and **b**, the curly line from the internal label attaches to the monomer that gets modified in the next forward transition. **c**, These cycles can be laid out in a lattice, with T phosphorylation along the horizontal axis and S phosphorylation along the vertical axis, while each set of four internal transitions (e.g., in the blue box corresponding to **b**) repeats along these axes. The external cycle in **a** is also highlighted in the green box. **d**, This lattice allows probing of its topological properties. In the ordinary case with similar rates throughout, i.e. $\gamma_{ex} \sim \gamma_{in}$, the system will perform a random walk ergodically through the phase space. In the topological regime when $\gamma_{ex} \gg \gamma_{in}$, once the system hits an edge it will continue around the edge, as can be verified by inspection. We plot the steady state in the topological regime which lies on the system edge, taking $\mu = 1, \rho = 2, \gamma_{tot} = 1$. There are 28 states along the edge with high probability, labeled by the order in which they are traversed in a typical trajectory. These trajectories form a global current along the edge of the state space.

model details, which were scattered around the text, to the beginning to Section II so that all the components of the model are introduced all at once. Such changes include:

- Moving the specification on the number of different conformations (squares and circles) for a generic state $(x, y)_s$ (line 257-267 in the manuscript) to the second paragraph of Section II where the internal cycles are introduced.
- Moving the explanation of KaiA interaction (line 239-243 in the manuscript) to the second paragraph of Section II as well, and explaining that the special nature of this interaction is the reason why we use a curly arrow to represent it: “We represent the internal transition associated with KaiA with a curly arrow, since KaiA promotes phosphorylation through rapid association and dissociation, activating a larger stoichiometric amount of KaiC as compared to the typical one-on-one binding like KaiB [18].” An explanation is also included in the Fig. 1 caption: “We represent the interaction with KaiA with a curly arrow because KaiA promotes phosphorylation by rapid association and dissociation rather than binding to fixed KaiC hexamers [18].”

*The model’s pieces need to be clearly, *individually*, described before putting them together. Starting with the cyclic part of the state (my s above) in isolation, and the important fact that it’s constrained to move $A \leftrightarrow B \leftrightarrow C \leftrightarrow D \leftrightarrow A$. (New sub-figure, showing a compass, N-E-S-W?) This cycle is an important ingredient. Then the conformation and the phosphorelation of the monomers, x, y, z . (Another new sub-figure? Not sure.) Then their transitions. (Existing a, b with simpler arrow labels?)*

We agree with the reviewer that the internal cycles are an important ingredient that contributes to the emergent oscillations. However, we introduce the external cycles first because these are the cycles that are actually observable in the topological regime, which takes inspiration from futile cycles in biology. As discussed earlier, the internal cycles do not correspond to real cycles in biology and internal states are thought to signal what external transition to occur next. Therefore, we think the definition of internal states and transitions would make more sense after the external transitions are introduced. The conformational changes in our current interpretation correspond to internal transitions, and are therefore introduced at the same time as the internal cycle. Because the same mathematical model has been introduced in the 2021 PRX paper, we would like to avoid repetition and therefore introduce the monomer reaction cycles in the context of KaiABC, where some arrows are labeled by possible biochemical reactions in addition to transition rates.

We have adopted the reviewer’s suggestion to add a simple “compass” between Fig. 1a and Fig. 1b in the main text, to clarify the directions of external reactions that the internal labels prime. Also see Fig. 5 in this document.

It must be abundantly clear exactly what numbers specify the complete state of the model. It must be abundantly clear exactly how many different rate constants are present, and what is assumed about their magnitudes. It must be abundantly clear whether which monomer is phosphorelated matters or not, whether the positions of the monomers matters or not. Whether the attachment point of the curly line connecting the A to the monomer matters or not. The reader should not have to guess these things by playing sudoku.

We thank the reviewer for pointing out the key details in our model that require clarification. We have updated Section II to introduce all the specifications of the model, including those that the reviewer suggest, all in the first few paragraphs of the section. Clarifications we have added in Section II include:

- Stating what numbers specify the complete state of the model, “The state of the system is completely specified by three variables $(x, y)_s$ ” in the second sentence of the first paragraph.

- Specifying how many parameters are present, “Before analyzing the model, we simplify our parametrization for the four transition rates γ_{ex} , γ'_{ex} , γ_{in} , γ'_{in} down to two parameters” in the beginning of the fifth paragraph.
- Clarifying what is assumed about the magnitude of the transition rates, “Before analyzing the model, we simplify our parametrization for the four transition rates γ_{ex} , γ'_{ex} , γ_{in} , γ'_{in} down to three parameters” at the end of the sixth paragraph.
- Specifying whether the position of the phosphorylated monomer matters or not, “In our model, we only keep track of the number of T and S phosphorylated monomers, while the specific location of each monomer does not matter.” at the end of the first paragraph and also “We only keep track of the total phosphorylation levels in T and S, while the location of each phosphorylated monomer can be arbitrary” in the Fig. 1 caption.
- Clarifying whether the attachment point of the internal tags matter or not: “In both a and b, the curly line from the internal label attaches to the monomer that gets modified in the next forward transition” in the Fig. 1 caption.

Pg 5 (line 324) says it is “simplifying the notation”. But in fact it seems to impose one new constraint on the 4 rate constants, so that only 3 are independent. Were they coupled this way before, does it matter? Why not just tell us how many parameters there are at the beginning?

The reviewer is correct that our new parametrization also imposes a new constraint. This constraint has been assumed throughout the paper, e.g., in choosing the transition rates for Fig. 1d. In principle, we should be able to vary all 4 transition rates, but we impose this constraint for simplification so that only 3 free independent parameters remain. In the revised manuscript we have clarified this point and moved this paragraph to Section II, where the model components are introduced all at once.

It’s very confusing, deceptive even, to call the internal cyclic state ABCD, and I think this ought to be changed. They are drawn as if they are somehow subsidiary components, but their constraint to change only in a cyclic order $A \leftrightarrow B \leftrightarrow C \leftrightarrow D \leftrightarrow A$ is really key to the whole model. It’s not clear to me whether they map to any known components of the real system. While KaiA and KaiB are real things, I don’t see the analogy, as the rest of the chain seems to be missing. Clearly C cannot be KaiC as that’s already something else.

We thank the reviewer for pointing out that the internal labels ABCD may be confused with the KaiA, KaiB, and KaiC molecules. We have changed the labels for internal states to avoid this possible confusion. The labels ABCD are arbitrary and have nothing to do with the KaiA, KaiB, and KaiC proteins. They are replaced with the labels NESW.

These internal transitions map to known reactions of the real KaiC system, which involve, for example, interaction with KaiA and KaiB and A-loop conformational changes. Possible reactions that correspond to each internal transition are listed below

- $S \rightarrow E$ could be the interaction with KaiA that promotes T phosphorylation [19].
- $E \rightarrow N$ could be the conformational change of the A-loop from the exposed to the buried state, stabilized by S phosphorylation [10].
- $N \rightarrow W$ could be the interaction with KaiB that primes T dephosphorylation [19].
- $W \rightarrow S$ could be the conformational change of the A-loop back from the buried to the exposed state. These reactions are less studied but should occur before KaiC starts to interact with KaiA again.

These references are also mentioned in the second paragraph of Section II when we introduce the internal transitions.

The assumption that the internal states change cyclically in this order has experimental support as well:

- The exposed A-loop interacts with KaiA, allowing it to promote the phosphorylation of KaiC. Therefore, conformational change to the exposed state ($W \rightarrow S$) should precede KaiA interaction ($S \rightarrow E$) [9].
- Buried A-loops prohibit binding of KaiA with KaiC [9]. Thus, conformational change of the A-loop from the exposed to buried state ($E \rightarrow N$) happens after KaiA interaction with KaiC ($S \rightarrow E$).
- A-loop burial ($E \rightarrow N$) makes the KaiC hexamer more rigid and promotes KaiB binding ($N \rightarrow W$). KaiB binding cannot occur with a more flexible KaiC variant with T and S phosphorylation but an exposed A-loop [11].

We add the following brief discussion of these evidence to the second to last paragraph of Section II, where experimental support for the interpretation of our model is discussed: “We also propose directed cycles of internal transitions as in Fig. 1b based on experimental results relating A-loop conformations with KaiA and KaiB. For example, A-loop exposure promotes KaiA binding [9] and prohibits KaiB binding [11] while A-loop burial prohibits KaiA binding and promotes KaiB binding [11].”

Figure 1(c) highlights the block $(x,y) = (1,1)$, why not the exact same block $(2,1)$ as Fig. 1(b)? (And perhaps, why not highlight where 1(a) fits in, too?)

We have highlighted in Fig. 1c both of the cycles depicted in Fig. 1a and Fig. 1b, with green and blue boxes respectively. We have also added the same green and blue boxes to the background of Fig. 1a and 1b to make the correspondence more transparent. See Fig. 5 in this document.

Figure 1(d) changes the arrangement of these blocks to group them more tightly. Can the two be drawn more similarly? At least the thing highlighting where the $(2,1)$ block from 1(b) lives, could be drawn in both.

Fig. 1c is redrawn to be more similar to Fig. 1d, where the four internal states for the same (x,y) are grouped together more tightly. The green and blue boxes in Fig. 1c are also reproduced in Fig. 1d. Because the green and blue colors appear both on these boxes and on lattice sites representing probabilities, to avoid confusion we have changed the color scheme for the steady-state probability in Fig. 1d. See Fig. 5 in this document.

The grouping here puts the slow transitions tightly together, and the fast transitions further apart. So it has 24 blocks on the edge, but 28 yellow sites. Instead of introducing one more kind of arrow around the cube for the cyclic motion, perhaps numbering the high-probability sites 1,2,3... 27,28 around the perimeter would help explain where this number comes from.

As suggested, we have labeled the high-probability sites along the edge of the state space in Fig. 1d, where the numbers increase in the direction of the edge currents. See Fig. 5 in this document. We make the connection between these high-probability states and the 28 unicyclic model we investigate in Fig. 3e of the manuscript, in the fifth paragraph of Section III: “This is because there are 28 slow internal transitions on the edge that form the effective bottleneck and dominate over the other fast external transitions, resulting in 28 high-probability states as noted in Fig. 1d.” We have also added the following explanation to the caption of Fig. 1d to explain

the numbering: “There are 28 states along the edge with high probability, labeled by the order in which they are traversed in a typical trajectory. These trajectories form a global current along the edge of the state space.”

If Fig 1(d) is data not a cartoon, it should show the what γ s are used. And should probably pick less extreme values, so that some intermediate colors are shown. (“Occupancy” might be clearer than “Prob.” too? Or “steady-state probability”?)

We have chosen less extreme values of transition rates that are closer to the transition between the topological and the trivial regime, and included the chosen parameters in the caption. The label for the color bar has also been updated, where “Prob.” has been replaced with “Steady-state probability.” However, the steady-state probability in the bulk turns out to be highly uniform in general, such that it is only possible to see a few colors in the visualization for any parameters chosen.

Fig 1(f) shows 24 edge states, not 28. This is wrong, surely.

These two numbers correspond to two different ways of representing the trajectory along the edge. Fig. 1f (Fig. 2b in the new manuscript) chooses a representative state from each of the 24 blocks on the edge. On the other hand, the 28 states obtain from ignoring the fast external transitions and can correspond to the 28 high probability states in Fig. 1d. We clarify this in the caption for Fig. 1f (now Fig. 2b). We reproduce Fig. 2 in the new manuscript and its caption in this document as Fig. 4, which is Fig. 1e and 1f from the original manuscript.

We chose the representation in Fig. 1f (now Fig. 2b) to best reflect how KaiC changes along a trajectory on the edge of the state space in our pictorial representation of the hexamer. For the extra four states from the 28 high-probability states, one in each corner, transitions from these states to the corner states already drawn do not lead to visible changes in the picture, since we only explicitly show changes in monomer conformations and phosphorylation/dephosphorylation in T and S.

In Figure 1(e), I think that a 45 degree line dividing day from night is confusing. The text claims that the T and S phosphorelation stages both occur during the day. Perhaps we are meant only to notice what color the small arrows are on, not what color the large diagrams are on. Surely this can be done in a less confusing way. It would help if this wasn't muddled up with the model – probably it should be a new figure, with its own caption.

The 45 degree line dividing day from night might be unconventional, but we draw the diagram specifically in this way such that the phosphorylation cycle in Fig. 1e (Fig. 2a in the revised manuscript) looks similar to the global cycle along the edge of the state space in Fig. 1f (Fig. 2b). For example, in both figures the fully T phosphorylated state with no S phosphorylation is in the lower right corner and the fully T and S phosphorylated state is in the top right corner.

This 45 degree separation also agrees with what is known about the timing in which the phosphorylation cycle proceeds. The phosphorylation phase happens during the day, which are the arrows in Fig. 1e (Fig. 2a) in the yellow region to the right of the 45 degree line representing daytime. The fully unphosphorylated and fully phosphorylated states sitting on the dividing line would be observed right around the time when night turns into day or vice versa (also see [36]).

As suggested, we have taken Fig. 1e and 1f to be their own figure, which is Fig. 2 in the revised manuscript and reproduced in this document as Fig. 4.

I do not know what we are to take from KaiA and KaiB on this figure. They are not mentioned in the caption.

The caption for Fig. 2 in the revised manuscript have been updated to include the role of KaiA and KaiB in the cycle. In particular, we add: “During the day, all six KaiC monomers get phosphorylated at the T-sites, and then at the S-sites. This phosphorylation phase is promoted by interaction with KaiA molecules [28]. By night, fully phosphorylated KaiC binds to KaiB, which sequesters KaiA from the solution. In the absence of KaiA, all the T-sites get dephosphorylated, followed by the S-sites [8].”

Note that KaiC is labelled only once on this figure, as if to hint at an ABCX cycle... but in fact KaiC applies equally to every part of the figure. This will be less confusing once the internal cycle is not called ABCD anymore.

The internal states are relabeled by NESW as suggested to avoid confusion.

///// Fig 2 / section 3

Below (1), “first nonzero eigenvalue”. I think this mean the nonzero eigenvalue of smallest absolute value? “first” doesn’t make much sense unless you have told us how you order them.

We thank the reviewer for pointing out that the “first nonzero eigenvalue” is not properly defined. The “first nonzero eigenvalue” refers to the nonzero eigenvalue with the smallest modulus in the real part, as the corresponding eigenstate is the slowest decaying mode [6]. We have clarified this point in the manuscript. We also adopt the same notation in [6] or [2] to denote the eigenvalue by $-\lambda_R \pm i\lambda_I$ with $\lambda_R, \lambda_I \geq 0$, to remove the repeated use of the absolute value symbol $|\cdot|$. Because such eigenvalues are either real or come in conjugate pairs, we take $\lambda_1 = -\lambda_R + i\lambda_I$ to denote the eigenvalue with a non-negative imaginary part. Changes made to the text include:

- Clarify the definition of the “first nonzero eigenvalue” in the first paragraph of Section III: “The dynamics of oscillations is typically dominated by the first non-zero eigenvalue of \mathcal{W} , which is the eigenvalue with the smallest modulus in the real part [6], denoted as $-\lambda_R \pm i\lambda_I$ for the real and imaginary parts λ_R and λ_I , respectively. Here $\lambda_R, \lambda_I \geq 0$ and we take $\lambda_1 = -\lambda_R + i\lambda_I$.”
- Similarly, clarify the definition in the second paragraph in Section II of the Supplementary Information: “The dynamics of $\mathbf{p}(t)$, as we will see soon, is in general dominated by the first non-zero eigenvalue of \mathcal{W} , which is the eigenvalue with the smallest modulus in the real part [6]. Such eigenvalues generally come in conjugate pairs, denoted by $-\lambda_R \pm i\lambda_I$ for the real and imaginary parts $\lambda_R, \lambda_I \geq 0$. In particular, we take $\lambda_R, \lambda_I \geq 0$ and $\lambda_1 = -\lambda_R + i\lambda_I$.”
- Get rid of the absolute value sign (since both λ_R and λ_I are non-negative quantities) in:
 - Defining decay time in the first paragraph of Section III: “In general, $\mathbf{p}(t)$ relaxes to the steady-state distribution through damped oscillations, with a decay time λ_R^{-1} and oscillation period $\mathcal{T} = 2\pi/\lambda_I$ [6] (also see Supplementary Fig. S3).” and in the third paragraph in Section II of the Supplementary Information: “After some transient behavior in the first oscillation cycle, the dynamics is dominated by λ_1 , where the oscillation period is given by $T = 2\pi/\lambda_I$ and the exponentially decaying envelope has a decay time λ_R^{-1} [6].”
 - Eq. 2 in the main text where coherence is defined: “ $\mathcal{R} \equiv \frac{\lambda_I}{\lambda_R}$,” and also Eq. (S4) in the Supplementary Information
 - Second to last paragraph in Section III which discusses how λ_I and λ_R changes individually: “Moreover, λ_R decreases more quickly than λ_I with increasing ρ , so that their ratio \mathcal{R} increases (see Supplementary Fig. S8).”

- The label indicating the exponential decay of the correlation function in Fig. S3, where we replace $e^{-|\lambda_R|t}$ with $e^{-\lambda_R t}$
- In the caption of Fig. S3 where the decay time is mentioned: “The oscillation has a decay time λ_R^{-1} and period $2\pi/\lambda_I$.”
- Various places in the last two paragraphs in Section V of the Supplementary information, where λ_I and λ_R are discussed with the spectral gaps: “Increasing μ leads to an increase in λ_I but very little change in λ_R . Since \mathcal{R} is proportional to λ_I while λ_R is roughly constant, coherence follows the same functional relationship as λ_I when μ increases (Fig. S8b). Therefore, since Δ_I closely tracks λ_I (see Fig. 4c in the main text), it also closely tracks \mathcal{R} monotonically.” and “As shown in Fig. S8c, λ_R still decreases monotonically as expected, as the rightmost band is compressed toward the origin. λ_I , on the other hand, develops non-monotonic behavior. Nevertheless, when λ_I begins to decrease, it approaches zero more slowly than λ_R and gives rise to a monotonically increasing \mathcal{R} that tracks Δ_I^o (and therefore Δ_I) in ρ (Fig. S8d). ”
- In Fig. S7a for the formula of coherence \mathcal{R}
- In Fig. S8a and S8c where the points are labeled with $|\lambda_I|$ and $|\lambda_R|$

“ ΔS_e (blue) increases while ΔS_b (orange) decreases with μ ”. This seems to describe only $1 < \mu < 5$ or so, which is confusing.

The reviewer is right that the decrease in ΔS with increasing μ only applies to intermediate values of μ ranging from $1.5 < \mu < 7$. This has been clarified in the following places:

- In the third paragraph of Section III in the revised manuscript: “This causes the entropy production on the edge (blue arrows) to increase, while the entropy production in the bulk (orange arrows) decreases, in the region $1.5 < \mu < 7$.”
- In the caption for Fig. 3c (originally Fig. 2c): “Due to the localization effects in Fig. 3b, the entropy production on the edge ΔS_e (blue) increases while entropy production in the bulk ΔS_b (orange) decreases with μ around $1.5 < \mu < 7$.”
- In the caption for Fig. S6 (originally Fig. S5) in the Supplementary Information: “On the other hand, the topological model supports a regime with increasing coherence but decreasing entropy production around $1.5 < \mu < 7$.”

Figure 2(b) has at least 4 kinds of arrows. I think one black arrow is the word “increases”. I do not understand what the blue and orange arrows are. The axes are not labelled but I assume they are x, y .

The reviewer’s understanding is correct that the black arrow to the right of μ represents “increase” in μ . We have changed the solid black arrow below μ to a dashed black arrow, to distinguish it from the previous arrow that represents “increase”. This dashed arrow signifies changes in the distribution of entropy production for the lattice after μ has increased. The updated figure is included in this document as Fig. 6.

The thickness of the blue and orange arrows represents the magnitude of the entropy production for the corresponding transitions. The blue arrows represent the contributions from transitions on the edge of the state space, while the orange arrows represent contributions from transitions in the bulk. Arrows point in the direction of the probability flux. We have updated Fig. 3b to add more detailed legend to the yellow and blue arrows, and also labeled the axes of the lattice in (x, y) (see Fig. 6 in this document). We have also tried to make the meanings of the arrows clearer in the main text and caption:

- In the third paragraph of Section III in the revised manuscript: “This causes the entropy production on the edge (blue arrows) to increase, while the entropy production in the bulk (orange arrows) decreases, in the region $1.5 < \mu < 7$.”
- In the caption for Fig. 3b (originally Fig. 2b): “Arrow thickness corresponds to the magnitude of the entropy production for the corresponding transitions and arrows point in the direction of the probability flux.” Here we have also added what the arrow directions mean, which is not mentioned in the original manuscript.

I am not sure that I have puzzled out why the sign of the slope of ΔS with μ is interesting. Can the goals here be made obvious?

The negative slope indicates that the system dissipates less free energy overall even when the external driving μ supplied to the system is stronger, a phenomenon aptly summarized in the literature as “getting more from pushing less” [30–33]. In general, energy dissipation should increase when the driving force increases, as is the case for the MWC model in the SI (Fig. S6). Because coherence monotonically increases with stronger driving, this negative slope leads to a system with more coherent oscillations but less energy cost as μ increases. We have edited the corresponding paragraph (third paragraph in Section III) to further highlight the significance of this negative slope. The paragraph is reproduced here.

- On the contrary, our model in the topological regime displays an unusual regime where coherence increases while entropy production per period becomes lower. Fig. 3a shows the coherence of our model as a function of the two parameters μ and ρ (also see Supplementary Fig. S4). Both increasing the thermodynamic force μ and going deeper into the topological regime by increasing ρ increase oscillation coherence as expected. However, the entropy production per period ΔS does not change monotonically with μ . As illustrated in Fig. 3b in a smaller lattice, the system response becomes localized to the edge as μ increases. This causes the entropy production on the edge (blue arrows) to increase, while the entropy production in the bulk (orange arrows) decreases, in the region $1.5 < \mu < 7$. Since the bulk contribution typically dominates the edge contribution, i.e., $\mathcal{O}(N^2) \gg \mathcal{O}(N)$ where N is the typical system size, the sum of their contributions also decreases (green curve in Fig. 3c). This negative slope of ΔS with respect to μ implies that the system dissipates less free energy overall even when the external driving μ supplied to each reaction is stronger. In other words, we “get more from pushing less” [30–33]. Meanwhile, since coherence increases monotonically with μ , this leads to an efficient regime with simultaneously increasing coherence and decreasing cost – see Fig. 3d for the region in parameter space that optimizes the cost-precision tradeoff, characterized by the ratio $\Delta S/\mathcal{R}$ [35].

This point is also discussed in more detail in the introduction, with the relevant paragraph reproduced below:

- In our model, we formulate topological methods in the context of stochastic systems and non-equilibrium thermodynamics to quantify the emergence of global cycles that naturally describe biological oscillations. We illustrate these methods in the context of the KaiABC system that regulates the circadian rhythm of cyanobacteria [29]. Following [6], we define coherence \mathcal{R} based on the damped oscillations of the decaying eigenmodes to quantify robustness of the emergent oscillations. With only a few free parameters, our model supports a dynamical regime with global currents localized on the edge, which has a topological origin [2]. Such localization leads to a novel regime with increasing coherence alongside decreasing entropy production as the external driving is increased. This unusual behavior, i.e., “getting more from pushing less”, has been observed in other scenarios such as driven electronic, Brownian,

FIG. 6. (Fig. 3 in the manuscript.) Coherence and an efficient regime with simultaneously increased precision and decreased cost. **a**, Phase diagram of coherence \mathcal{R} for the topological model, which increases with respect to thermodynamic force μ and ratio of timescales ρ as expected. **b**, Entropy production ΔS moves from the bulk to the boundary of the system with increasing μ , illustrated on a smaller lattice. The system is in the topological regime; $\rho = 5$. Blue arrows represent entropy production on the edge ΔS_e and orange arrows represent entropy production in the bulk ΔS_b . Arrow thickness corresponds to the magnitude of the entropy production for the corresponding transitions and arrows point in the direction of the probability flux. **c**, Entropy production per period ΔS for the topological model ($N_x = N_y = 6$) is decomposed into a bulk and an edge contribution. Due to the localization effects in Fig. 3b, the entropy production on the edge ΔS_e (blue) increases while entropy production in the bulk ΔS_b (orange) decreases with μ around $1.5 < \mu < 7$. Typically, the bulk contribution dominates the edge contribution, hence their sum ΔS (green) also decreases with increasing μ . **d**, Phase diagram of the cost-precision tradeoff $\log(\Delta S/\mathcal{R})$ [35]. There is a global optimum at $\mu = 6.5$, $\rho = 4.1$ due to the non-monotonic behavior of ΔS that suggests an unusual efficient regime. **e**, Comparison of coherence for different KaiC models with randomly sampled parameters (individual points): there is a strongly-driven regime where the topological model has the highest coherence. The dashed lines represent the upper bounds on coherence for the corresponding models in terms of the thermodynamic force μ [6] (see Methods). The purple dashed line is the upper bound for the 28-state unicyclic model, which is approached by the edge state in the strongly topological regime. Values of ΔS in all panels are given in units of $\gamma_{tot} k_B$.

and glassy systems [30–33]. Its occurrence in a stochastic system like ours remains surprising and, as we discuss later, reflects the topological nature of our model. Moreover, the model also exhibits high coherence compared to other models of KaiABC. Further, we introduce spectral gaps as a predictor of coherence, and examine the saturation of global spectral bounds.

The MWC model presented in (S4) (S5) is a pretty complicated parameterization with lots and lots of researcher degrees of freedom. Apparently this is taken from a reference, so not changing it is probably the right thing to do.

Presentation: if $k=1$, then why introduce it at all? Since χ depends on i , why not write χ_i with

a subscript? Or just immediately write the rates as being either 1 or $\exp(E^*(i-3)/3)$?

The reviewer is right that we took the MWC model and its notations from a reference, i.e., Barato & Seifert 2017 [6]. We have simplified the notation based on the reviewer's suggestions. We have removed any mention of k from the original parametrization, and have rewritten the transition rates from C_i (\tilde{C}_i) to \tilde{C}_i (C_i) as being either 1 or $e^{E(i-3)/3}$ ($e^{E(3-i)/3}$). These changes are made to the third paragraph of the Supplementary Information, reproduced below:

- In this paper we follow one of the simplest versions of MWC-type models described in [6] and use the same parametrization of the transition rates. When KaiC is in the active state, it is more likely to be phosphorylated. On the other hand, when KaiC is in the inactive state, it is more likely to be dephosphorylated. These dominant reactions are represented by the red vertical arrows in Fig. S5a with transition rates $\gamma e^{\eta/2}$. The reverse transitions (smaller black vertical arrows) have rates $\gamma e^{E/6}$. γ , η , and E are model parameters that can be varied. For the horizontal transitions in Fig. S5a, the rates from C_i to \tilde{C}_i are denoted as $k_{i\tilde{i}}$ and the rates from \tilde{C}_i to C_i are denoted as $k_{\tilde{i}i}$, where

$$k_{i\tilde{i}} = \begin{cases} 1, & i = 0, 1, 2, 3 \\ e^{E(i-3)/3}, & i = 4, 5, 6 \end{cases}$$

and

$$k_{\tilde{i}i} = \begin{cases} e^{E(3-i)/3}, & i = 0, 1, 2, 3 \\ 1, & i = 4, 5, 6 \end{cases}.$$

The value of γ sets the relative timescales between the phosphorylation/dephosphorylation reactions and conformational changes.

Also, why call the phosphorelation level i here, not x or y as elsewhere?

In Barato & Seifert 2017 [6] and the earlier model from van Zon et al. [37], only one kind of phosphorylation was assumed. This is neither the T nor S phosphorylation as described in our model, so we have not adopted the same notation x or y as is used in our model.

Fig S4: This has 4 colors & 3 styles of arrows for rate constants. I'm not sure why some are red. I think I eventually puzzled out why some are green, but this does not have to be a sudoku puzzle. Panel (a) could have axes arrows phosphorelation & activity like (b1).

We have added more clarifications in the text and figure captions to explain what the various arrows and colors mean in the figure. Specifically, the red arrows are the dominant directions of phosphorylation/dephosphorylation, used to highlight the direction in which the cycle will proceed. This was discussed in the SI text but not in the caption, and we have added the explanation to the caption. As suggested, we have also added axes arrows (in red) showing the direction of phosphorylation and dephosphorylation in panel (a).

More detailed discussions have also been added to explain Fig. S5 that illustrates the bilayer lattice model, in particular how the green and black states in the lattice relate to the dashed arrows between layers representing KaiB binding/unbinding. To refer to the different subfigures more easily, we have also split the left and right panels of Fig. S5b into two subfigures, Fig. S5b and S5c. The modified figure is reproduced here as Fig. 7. The last paragraph in Section III of the SI, which introduces the bilayer lattice model, is reproduced below:

- In Fig. S5b and S5c we show the network structure for the bilayer lattice model, adapted from Li *et al.* [38]. The state space consists of two layers of 7×7 lattices that are connected to each other. Fig. S5b shows the network structure of bottom layer from Fig. S5c. Similar to our

FIG. 7. (Supplementary Fig. S5 in the manuscript.) **Single-molecule models for KaiC.** **a**, The MWC model. The KaiC hexamer can be in one of two conformational states, the active state C_i or the inactive state \tilde{C}_i . KaiC tends to phosphorylate in the active state and dephosphorylate in the inactive state, highlighted by the vertical red arrows. The slower reverse reactions are represented by smaller black vertical arrows. Horizontal transitions correspond to conformational changes of the KaiC hexamer between active and inactive states. **b**, The bottom layer of the bilayer lattice model. Subscripts x and superscripts y represent T and S phosphorylation level, respectively, just as for the topological model. Solid black arrows represent phosphorylation/dephosphorylation transitions with rates γ_{ex} and slower reverse rates γ'_{ex} . **c**, A zoomed-out view of the bilayer lattice model. Individual states in each layer are not shown. The green dashed arrows from the bottom to the top layer represent KaiB binding, while the gray dashed arrows from the top to the bottom layer represent KaiB unbinding. For the states colored green in **b**, transition rates are γ_{in} for KaiB binding and γ'_{in} for KaiB unbinding. For the states colored black, transition rates are γ'_{in} for KaiB binding and γ_{in} for KaiB unbinding. Larger dashed arrows (for any color) correspond to faster rates γ_{in} while smaller dashed arrows correspond to slower rates γ'_{in} . During KaiB binding and unbinding, the phosphorylation levels remain the same.

topological model, the x and y coordinates (labeled by superscripts and subscripts C_y^x in Fig. S5b) represent T and S phosphorylation levels, respectively. The top layer has an identical structure to the bottom layer, with states labeled by \tilde{C}_y^x . C_y^x in the bottom layer corresponds to unbound KaiC while \tilde{C}_y^x in the top layer represents KaiB-bound KaiC. The system can make transitions $C_y^x \rightleftharpoons \tilde{C}_y^x$ between layers while keeping x and y coordinates fixed, which corresponds to KaiB binding and unbinding. To aid comparison, we simplify the model by parametrizing the transition rates with μ and ρ , the same parameters for our topological model. Within both layers, the orientations for faster reactions on each edge are chosen such that they form an overall counterclockwise cycle, to capture the order of the phosphorylation cycle (see Fig. S5b). These phosphorylation/dephosphorylation reactions (solid black arrows) are assumed to have uniform rates γ_{ex} while the slower reverse reactions have rates γ'_{ex} . On the other hand, the KaiB-binding transitions between layers have rates γ_{in} and γ'_{in} . In the upper right half of the lattice where the states are labeled green in Fig. S5b, the dominant reaction between layers is the upward KaiB binding transitions (green dashed arrows in Fig. S5c) with rates γ_{in} , while the downward KaiB unbinding transitions (gray dashed arrows) have slower rates γ'_{in} . This is consistent with the fact that S phosphorylation promotes KaiB binding [39]. In the rest of the lattice where the states are labeled black in Fig. S5b, the dominant reaction between layers is the downward KaiB unbinding transition with rates γ_{in} , while the upward KaiB binding transitions take the slower rate γ'_{in} . This parametrization scheme gives rise to cycles where a KaiC molecule phosphorylates in the bottom layer, binds to KaiB, gets dephosphorylated in the top layer, and unbinds with KaiB to restart the cycle.

FIG. 8. (Supplementary Fig. S6 in the manuscript.) **Entropy production per period and coherence for the MWC and the topological model.** The MWC model requires more entropy production to maintain a higher coherence, and its coherence decreases as still more entropy is produced under a stronger driving μ . On the other hand, the topological model supports a regime with increasing coherence but decreasing entropy production around $1.5 < \mu < 7$. For the MWC model, the parameters used are $\gamma = e^5$, $E = 10$. We plot μ in a smaller range for this model because outside the plotted range, we have $\lambda_I = 0$, which makes the oscillation period \mathcal{T} and hence entropy production per period ΔS ill-defined. Entropy production per period is given in units of $k_B \gamma_{tot}$

Fig S5: Is the green y-axis $\Delta S/T$? Or maybe we are to puzzle out that it is $\Delta S/(T k_B \gamma_{tot})$? I think the symbols should appear in the label. (Likewise Coherence R , as before.) All x-axes are μ , I think, although their ranges differ and only the right hand ones are labelled.

The green y-axis is entropy production per period ΔS , in units of $k_B \gamma_{tot}$. We have updated Fig. S5 (Fig. S6 in the revised manuscript) to include math symbols in the label. We have also labeled all the x-axes in the figure, all of which are in μ . This figure is reproduced here as Fig. 8. The range in μ is smaller for the MWC model because outside the range we have plotted, there is no oscillation and thus coherence and entropy production per period becomes ill-defined. We have included a clarification in the Fig. S6 caption: “We plot μ in a smaller range for this model because outside the plotted range, we have $\lambda_I = 0$, which makes the oscillation period \mathcal{T} and hence entropy production per period ΔS ill-defined.”

I’m unsure that this idea of deleting some states from the grid makes sense for a state-space model. It seems to be borrowed from solid state physics, where obviously one may chip a bit off a crystal. Here, why would the allowed S-phosphorelation states depend on how many T-phosphorelations are present?

In biological systems, changes in environmental conditions like a limited amount of substrates can block off some states from being accessed, deleting them from the state space. For example, having a limited number of KaiA molecules that promote phosphorylation could block off some highly phosphorylated states from being accessible. We have rewritten the paragraph in Section I in the Supplementary Information, to focus more on explaining our model in terms of biology and stochastic systems; reproduced below.

- In the topological regime ($\rho \gg 0$), our model supports global currents that propagate along the edge of the state space. The edge currents turn out to be robust against changes in the environment that render certain states inaccessible. For example, a limited number of KaiA molecules in solution could prevent the system from accessing the highly phosphorylated states, while a limited number of KaiB could block off hypophosphorylated states where x and y are small. Despite missing certain states in the lattice, the probability currents in our model can continue to propagate along the new edge of the state space, as illustrated by Fig. S1. The robustness in edge currents could explain how biological systems maintain stable dynamics in the face of changing external conditions.

We have also added the following clarification to the caption of Fig. S1: “For illustration purposes, we use a smaller system with $N_x = 4, N_y = 3$ and only show the forward transitions γ_{ex} and γ_{in} .”

On a broader note, our model is the only one that gives rise to a well-defined order in T and S phosphorylations (other models put this in by hand), even while allowing the two phosphorylations to occur independently. Here, the observation that the S and T phosphorylation are indeed not independent of each other (which is also true and necessary in the real biological system where S phosphorylation only happens after complete T phosphorylation) emerges as a result of the topology, even if some components or states are blocked, which gives rise to robustness.

III. REVIEWER 3

In this manuscript the authors propose a mechanism for coherent biochemical oscillations that relies on the emergence of a “topological phase” where the system gets trapped into a region of the space states. In particular, their model is 2D model and this region corresponds to the edges of a 2D lattice. The authors also discuss the applicability of their model to the KaiC system, one the best understood circadian clocks. They also discuss the relation between energy dissipation and coherent oscillations in the light of known thermodynamic bounds, showing that it is possible for the model to saturate such bound in a limit of timescale separation.

I think the idea proposed by the authors connecting topology concepts from “traditional” condensed matter physics with biochemical oscillations is interesting, independent of the speculation that their model might be a good model for the specific KaiC system. I am inclined to recommend acceptance of the manuscript.

We thank the reviewer for their positive evaluation of the article.

1) In the intro: “It remains unclear how such oscillations ... emerge”. Not the best sentence. There are many mechanism, such as certain feedback loops in chemical reactions, that demonstrate that such oscillations can occur. While it is valuable to propose a new mechanism it is not correct that it is unclear how such oscillations emerge. Off course there is much to understand in this field.

The reviewer is right to point out that there has been many proposed mechanisms that demonstrate the emergence of such circadian oscillations. We have made this clear in the revised manuscript and included relevant references. However, we note that models such as feedback loops generally involve relatively small reaction networks, where the underlying reactions occur on a similar timescale as the oscillation itself. On the other hand, more detailed models generally requires a large collection of parameters that depend on system details. The contribution of our paper is to provide a parsimonious mechanism with just a few parameters that clearly demonstrate the emergence of long oscillations from a large phase space of faster reactions. This is what we hope to convey in this part of the introduction. The sentence pointed out by the reviewer is thus corrected as below: “Previously proposed models such as feedback loops of chemical reactions typically involve either small reaction networks consisting of reactions on a similar timescale as the oscillation itself, or a large number of system-dependent parameters [22, 37, 40]. A parsimonious model is still lacking for explaining how such oscillations with their long timescales can emerge from a large phase space of faster chemical reactions.” We have also modified the abstract to clarify the contribution of our model compared to others. The abstract is reproduced below:

- **Long and stable timescales** are often observed in complex biochemical networks, such as in emergent oscillations. How these robust dynamics persist remains unclear, given the many stochastic reactions and shorter time scales demonstrated by underlying components. We propose a topological model **with parsimonious parameters that produces long oscillations** around the network boundary, effectively reducing the system dynamics to a lower-dimensional **current**. Using this to model KaiC, which regulates the circadian rhythm in cyanobacteria, we compare the coherence of oscillations to that in other KaiC models. Our topological model localizes currents on the system edge for an efficient regime with simultaneously increased precision and decreased cost. Further, we introduce a new predictor of coherence from the analysis of spectral gaps, and show that our model saturates a global thermodynamic bound. Our work presents a new mechanism for emergent oscillations in complex biological networks **utilizing** dissipative cycles to achieve robustness and efficient performance.

The fact that we propose a parsimonious model that generates coherent oscillations with few parameters is also mentioned in the first paragraph of the Discussion section, reproduced below:

- We have proposed a parsimonious topological model that generates coherent oscillations, which supports an unusual regime with increased coherence and simultaneously decreased energetic cost. We find that the imaginary spectral gap can be used to predict the oscillation coherence. Applied to the KaiABC system, our model has high coherence compared to other KaiABC models and more closely saturates a global spectral bound, similar to the most coherent unicyclic models. Further, the kinetic ordering of the KaiABC phosphorylation cycle arises naturally as an edge current in our model. In contrast to typical MWC-type models, it does not require restriction of the configuration space to all-or-none conformational change or fine-tuning of the reaction rates. With only a few parameters, our model can generate the experimentally observed phosphorylation sequence.

2) *In Fig. 1f the authors connect their 2D model with a single KaiC molecule. How does the connection work with more KaiC molecules? Is the network still 2D if we have two KaiC molecules?*

The connection depends on the interaction between the KaiC molecules – for non-interacting molecules, the networks are still 2D. More interesting, when KaiC molecules interact (as we expect), the network will be higher-dimensional. For N molecules, the network will be a maximum of $2N$ -dimensional. Meanwhile, specific molecular interactions can introduce constraints that change the geometry of the state space. For instance, we consider the effects of competition over a limited amount of KaiA molecules as the reviewer suggests in the next point, which turns the state space from a hypercube to a hyperprism. We add a detailed discussion in a new section in the Supplementary Information titled “Generalizations to many molecules” along with a new figure (Fig. S9, reproduced in this document as Fig. 9), where we study the geometry of the state space for two or more molecules when KaiA is considered a scarce resource that can be depleted. We have included the following sentence in the second to last paragraph in the Discussion section to refer to these new results: “See Section VII in Supplementary Information for preliminary work on generalizing our single-molecule model to many molecules and incorporating population-level effects such as competition for KaiA molecules.” The relevant paragraphs in the Supplementary Information are reproduced here:

- In this section, we explore generalizations of our single-molecule model to many molecules. In particular, we investigate the higher-dimensional state space formed by two or more KaiC molecules. We further assume that KaiA is a scarce resource and study the resulting state space geometry modified by the constraints introduced by the competition over KaiA. This competition is intrinsically a population-level effect, as all KaiC molecules compete for the same scarce resource to complete the phosphorylation cycle. Finally, we look at the effects of limited or excess amounts of KaiA on the phosphorylation cycles for our single-molecule model or the many-molecule generalization.

We first look at the state space of our model when there are two KaiC molecules in consideration. In this case, the indices \$(x_1, x_2, y_1, y_2)_{s_1, s_2}\$ completely specify the system of two molecules, where the subscripts on the variables \$x, y, s\$ denote molecule number. For each phosphorylation level \$(x_1, x_2, y_1, y_2)\$, there are \$4 \times 4 = 16\$ internal states that form a tesseract, the 4D generalization of a cube. These internal states repeat themselves in the four directions spanned by the external variables \$x_1, x_2, y_1, y_2\$ to form a state space in the shape of a 4D hypercube.

To study population-level effects, we assume that KaiA molecules, which promotes KaiC phosphorylation [28], are a scarce resource that can be depleted. This agrees with experimental observations that when KaiA concentrations drop below a certain point, oscillations are prohibited and KaiC phosphorylation levels remain low [18]. We further assume that each KaiA molecule promotes the T phosphorylation of one KaiC monomer. Since there are six T phosphorylation sites for each KaiC hexamer, we consider the regime \$N_A < 12\$, where the number of KaiA molecules \$N_A\$ is not enough to fully phosphorylate both molecules. This competition

FIG. 9. (Fig. S9 in the manuscript.) Cross sections of the 4D state space for two KaiC molecules when competition for KaiA is taken into account. **a**, For $N_A = 4$, the constraint $x_1 + x_2 \leq N_A$ from competition over KaiA reduces the state space in x_1 - x_2 from a 7×7 square to an isosceles right triangle with side length 5. All side lengths refer to the number of phosphorylation levels contained in that direction. The 3D cross section in y_2 for the complete 4D state space is a right triangular prism, obtained by repeating the isosceles right triangle in y_1 . Such geometries come up in the regime $N_A \leq 6$. **b**, For $N_A = 8$, the constraint reduces the state space to an irregular pentagon in x_1 - x_2 , which results from cutting off a corner from a 7×7 square. The 3D cross section in y_2 is a pentagonal prism that derives from repeating the pentagon in y_1 . Such geometries come up in the regime $6 < N_A < 12$.

introduces the constraint $x_1 + x_2 \leq N_A$. In the following, we consider the geometry of this state space in two cases. In either case, the hypercubic state space of the system is reduced to a hyperprism from the constraint introduced by competition over KaiA. Depending on the amount of available KaiA, the cross-sections of the hyperprism can be different, as illustrated in Fig. S9. Such changes to the geometry of the state space modifies its edges and introduces dependence on phosphorylation levels of different molecules along these new edges, e.g., $x_1 + x_2 = 4$ on the hypotenuse of the triangle in Fig. S9a.

One regime is when $N_A \leq 6$. This is when KaiA can fully phosphorylate one KaiC hexamer at most. In this case, the x_1 - x_2 subspace will be reduced from a 2D square to an isosceles right triangle, as illustrated in the lower part of Fig. S9a, taking $N_A = 4$ as an example. To study the shape of the four-dimensional state space, we can look at its 3D cross-sections by holding one coordinate fixed. As shown in Fig. S9a, with a fixed y_2 , the state space in x_1 - x_2 - y_1 space is a right triangular prism, obtained by repeating the right triangle in x_1 - x_2 along the y_1 direction. This 3D cross-section is the same for any y_2 , which means that this prism repeats itself in the fourth dimension y_2 , forming a 4D hyperprism.

In another regime, $6 < N_A < 12$. This is when KaiA can fully phosphorylate one molecule but not enough to phosphorylate both. In Fig. S9b, we illustrate the geometry for the case

$N_A = 8$. The x_1 - x_2 subspace is an irregular pentagon, formed by removing the upper right corner (an isosceles right triangle) from the 2D square. The 3D cross-section for a fixed y_2 is a pentagonal prism that repeats in y_2 to form another 4D hyperprism, which differs from the above by its different 3D cross-section.

In general, if there are N KaiC molecules, the constraint on the T phosphorylation levels becomes $x_1 + x_2 + \dots + x_N \leq N_A$. The state space will be a $2N$ -dimensional hyperprism, where high-dimensional ‘‘corners’’ are removed from a $2N$ -dimensional hypercube due to the constraint.

3) *One feature that is found in many biochemical oscillators is the ‘‘competition for a scarce resource’’. In the case of the KaiC system this scarce resource would be the KaiA molecules. Roughly the number of KaiC molecules that ‘‘synchronizes’’ is equal to the number of KaiA molecules, which must be smaller than the number of KaiC molecules. The typical models for the KaiC system mentioned by the authors have this feature. Does the present 2D model have this feature? If we were to remove the KaiA molecules from the system what would happen to the present model?*

We thank the reviewer for the questions that help clarify the assumptions of our model and expand our analysis. The present model does not have this feature of ‘‘competition for a scarce resource’’. It assumes that resources (e.g., KaiA and KaiB) are not scarce. KaiA and KaiB have fixed concentrations and act as reservoirs that power the KaiC reaction cycles. We have included a discussion on incorporating competition over KaiA when we generalize our model to many molecules in the new Section VII in the Supplementary Information. The competition introduces constraints to the state space and modify its geometry from a high-dimensional hypercube to a hyperprism.

In addition, we discuss the consequences when KaiA is removed, and when KaiA is in limited or excess amounts, in Section VII of the Supplementary Information. We also add a new figure (Fig. S10, reproduced in this document as Fig. 10). The effects of KaiA are discussed for both our single-molecule model and the generalization to many molecules introduced above. In both models, removing KaiA leads to sustained low levels of phosphorylation while excess KaiA leads to sustained high levels of phosphorylation, consistent with experiments [18, 41]. The relevant paragraphs are reproduced here:

- When the amount of KaiA is varied, this simple model behaves the same as what is observed in experiments, i.e., limited KaiA leads to sustained low levels of phosphorylation while excess KaiA leads to sustained high levels of phosphorylation [18, 41]. When there is little to no KaiA, states with high T phosphorylation levels are blocked. When there is too much KaiA, there is no competition and no constraint whatsoever on the state space, which means that each KaiC molecule oscillates independently without synchronizing. Moreover, the strong driving from KaiA would promote phosphorylation of each molecule (see Fig. S10b), leading to a high overall phosphorylation level. We note that the above discussions have relied on simplifying assumptions on the nature of KaiA-KaiC interactions. A more thorough investigation of a many-molecule model would take into account more realistic properties of KaiA.

Such effects of KaiA can also be captured just by our single-molecule model in the main text. Because the thermodynamic force for the $S \rightarrow E$ internal transition, which we denote as μ_A , characterizes the driving from interaction with KaiA, changes in μ_A correspond to changes in concentrations of KaiA [27]. We keep the slower $E \rightarrow S$ transition rate fixed at γ'_{in} and let the $S \rightarrow E$ transition rate depend on μ_A by $\gamma'_{in} e^{\mu_A/k_B T}$. Fig. S10a shows the steady-state probability distribution for $\mu_A = 0$ while μ and ρ are kept the same, which corresponds to complete removal of KaiA. As we can see, the probability is mostly localized in states near the lower left corner, corresponding to low levels of phosphorylation. In contrast, in Fig. S10b we set $\mu_A = 5$, corresponding to excess amounts of KaiA. The probability, in turn, becomes localized in highly phosphorylated states. The average total phosphorylation level (T and S combined) as a function of μ_A is shown in Fig. S10c, which shows increasing levels

FIG. 10. (Fig. S10 in the manuscript.) **The effects of limited or excess amounts of KaiA on the KaiC phosphorylation cycle.** **a**, Steady-state probability distribution in the absence of KaiA. The thermodynamic force for the $S \rightarrow E$ internal transition, μ_A , which characterizes the external driving from interaction with KaiA, is set to be 0. The probability is localized on the lower left corner, corresponding to a hypophosphorylated state for KaiC in the absence of KaiA, consistent with experimental results [18]. In all panels, we take $\mu = 3, \rho = 5$. **b**, Steady-state probability distribution for $\mu_A = 5$. The probability is localized in highly phosphorylated states of KaiC, also consistent with experiments [41]. **c**, Total phosphorylation level of T and S as a function of μ_A , averaged over the steady-state probability distribution. KaiC autodephosphorylates as μ_A decreases with the removal of KaiA, and phosphorylates as μ_A increases with the addition of KaiA. $\mu_A = \mu = 3$ corresponds to the unmodified model as introduced in the main text, whose phosphorylation level is illustrated with a horizontal dashed line. **d**, Coherence as a function of μ_A . Coherent oscillations are inhibited with the removal of KaiA or an excess of KaiA. Coherence for $\mu_A = 3$ is illustrated with a horizontal dashed line.

of phosphorylation as μ_A increases and KaiA is added. For either limited ($\mu_A < 3$) or excess ($\mu_A > 3$) amounts of KaiA, oscillation tends to be attenuated as illustrated by the decrease in coherence in Fig. S10d.

4) *In models for biochemical oscillators with “competition for a scarce resource” the network of states is typically very complicated. But features such as this competition and feedback loops will make the system get trapped in some sort of “limit cycle”. Well there might be more than one cycle. Would the authors say that this is a similar thing to what is happening in their model? In other words, is their model a nice and simple way to express this idea of dimension reduction that probably also happens in other models for biochemical oscillators or do the authors think that this emergence of a topological cycle is new feature in their model different from other models for biochemical oscillators?*

We thank the reviewer for considering our work in a broader context. While our model does not involve competition, the typical mechanism of feedback loops mentioned by the reviewer has similarities with our model, i.e., the existence of loops in our network leads to a regime where the system dynamics gets trapped in a cycle. In this sense, our model is a simple way to express this idea

of dimensionality reduction. However, our model also has some unique properties that differ from traditional feedback loops, which usually requires multiple species and time varying transition rates. In our model, dimensionality reduction and limit cycle dynamics are observed in a single-molecule description and arise due to the topological properties of the single-molecule state space. Our model therefore offers a new general mechanism for dimensionality reduction which is different from previous proposals. How general these topological ideas can be applied to biological systems remain open.

5) Ref 43. I think the authors mean to cite *Phys. Rev. E* 106, 014106 2022, which contains a bound in terms of the thermodynamic cost ΔS and not the thermodynamic affinity.

We thank the reviewer for pointing this out, indeed the thermodynamic cost ΔS does shed new light in addition to the thermodynamic affinity. We have included the reference in the following places where we discuss this important quantity ΔS :

- In the third paragraph of Section III, last sentence: “Meanwhile, since coherence increases monotonically with μ , this leads to an efficient regime with simultaneously increasing coherence and decreasing cost – see Fig. 3d for the region in parameter space that optimizes the cost-precision tradeoff, characterized by the ratio \$\Delta S/\mathcal{R}\$ [35].”
- In the caption for Fig. 3d: “**d**, Phase diagram of the cost-precision tradeoff $\Delta S/\mathcal{R}$ [35].”

-
- [1] Feng Liu and Katsunori Wakabayashi, “Novel topological phase with a zero berry curvature,” *Physical review letters* **118**, 076803 (2017).
- [2] Evelyn Tang, Jaime Agudo-Canalejo, and Ramin Golestanian, “Topology protects chiral edge currents in stochastic systems,” *Physical Review X* **11**, 031015 (2021).
- [3] Michel Fruchart, Ryo Hanai, Peter B Littlewood, and Vincenzo Vitelli, “Non-reciprocal phase transitions,” *Nature* **592**, 363–369 (2021).
- [4] Jürgen Schnakenberg, “Network theory of microscopic and macroscopic behavior of master equation systems,” *Reviews of Modern physics* **48**, 571 (1976).
- [5] Hao Ge and Hong Qian, “Physical origins of entropy production, free energy dissipation, and their mathematical representations,” *Physical Review E* **81**, 051133 (2010).
- [6] Andre C Barato and Udo Seifert, “Coherence of biochemical oscillations is bounded by driving force and network topology,” *Physical Review E* **95**, 062409 (2017).
- [7] Matthias Uhl and Udo Seifert, “Affinity-dependent bound on the spectrum of stochastic matrices,” *Journal of Physics A: Mathematical and Theoretical* **52**, 405002 (2019).
- [8] Michael J Rust, Joseph S Markson, William S Lane, Daniel S Fisher, and Erin K O’Shea, “Ordered phosphorylation governs oscillation of a three-protein circadian clock,” *Science* **318**, 809–812 (2007).
- [9] Yong-Ick Kim, Guogang Dong, Carl W Carruthers Jr, Susan S Golden, and Andy LiWang, “The day/night switch in KaiC, a central oscillator component of the circadian clock of cyanobacteria,” *Proceedings of the National Academy of Sciences* **105**, 12825–12830 (2008).
- [10] Roger Tseng, Yong-Gang Chang, Ian Bravo, Robert Latham, Abdullah Chaudhary, Nai-Wei Kuo, and Andy LiWang, “Cooperative KaiA–KaiB–KaiC interactions affect KaiB/SasA competition in the circadian clock of cyanobacteria,” *Journal of molecular biology* **426**, 389–402 (2014).
- [11] Yong-Gang Chang, Nai-Wei Kuo, Roger Tseng, and Andy LiWang, “Flexibility of the C-terminal, or CII, ring of KaiC governs the rhythm of the circadian clock of cyanobacteria,” *Proceedings of the National Academy of Sciences* **108**, 14431–14436 (2011).
- [12] Taeko Nishiwaki and Takao Kondo, “Circadian autodephosphorylation of cyanobacterial clock protein KaiC occurs via formation of ATP as intermediate,” *Journal of Biological Chemistry* **287**, 18030–18035 (2012).

- [13] Mattia Rigotti, Omri Barak, Melissa R Warden, Xiao-Jing Wang, Nathaniel D Daw, Earl K Miller, and Stefano Fusi, “The importance of mixed selectivity in complex cognitive tasks,” *Nature* **497**, 585–590 (2013).
- [14] Evelyn Tang, Marcelo G Mattar, Chad Giusti, David M Lydon-Staley, Sharon L Thompson-Schill, and Danielle S Bassett, “Effective learning is accompanied by high-dimensional and efficient representations of neural activity,” *Nature neuroscience* **22**, 1000–1009 (2019).
- [15] Greg J Stephens, Bethany Johnson-Kerner, William Bialek, and William S Ryu, “Dimensionality and dynamics in the behavior of *C. elegans*,” *PLoS computational biology* **4**, e1000028 (2008).
- [16] Masato Nakajima, Keiko Imai, Hiroshi Ito, Taeko Nishiwaki, Yoriko Murayama, Hideo Iwasaki, Tokitaka Oyama, and Takao Kondo, “Reconstitution of circadian oscillation of cyanobacterial KaiC phosphorylation in vitro,” *science* **308**, 414–415 (2005).
- [17] Christian Brettschneider, Rebecca J Rose, Stefanie Hertel, Ilka M Axmann, Albert JR Heck, and Markus Kollmann, “A sequestration feedback determines dynamics and temperature entrainment of the KaiABC circadian clock,” *Molecular Systems Biology* **6**, 389 (2010).
- [18] Hakuto Kageyama, Taeko Nishiwaki, Masato Nakajima, Hideo Iwasaki, Tokitaka Oyama, and Takao Kondo, “Cyanobacterial circadian pacemaker: Kai protein complex dynamics in the KaiC phosphorylation cycle in vitro,” *Molecular cell* **23**, 161–171 (2006).
- [19] Jenny Lin, Justin Chew, Udaysankar Chockanathan, and Michael J Rust, “Mixtures of opposing phosphorylations within hexamers precisely time feedback in the cyanobacterial circadian clock,” *Proceedings of the National Academy of Sciences* **111**, E3937–E3945 (2014).
- [20] Jun Abe, Takuya B Hiyama, Atsushi Mukaiyama, Seyoung Son, Toshifumi Mori, Shinji Saito, Masato Osako, Julie Wolanin, Eiki Yamashita, Takao Kondo, *et al.*, “Atomic-scale origins of slowness in the cyanobacterial circadian clock,” *Science* **349**, 312–316 (2015).
- [21] Damien Simon, Atsushi Mukaiyama, Yoshihiko Furuike, and Shuji Akiyama, “Slow and temperature-compensated autonomous disassembly of kaiB–kaiC complex,” *Biophysics and Physicobiology* **19**, e190008 (2022).
- [22] Joris Paijmans, David K Lubensky, and Pieter Rein Ten Wolde, “A thermodynamically consistent model of the post-translational Kai circadian clock,” *PLoS computational biology* **13**, e1005415 (2017).
- [23] John J Hopfield, “Kinetic proofreading: a new mechanism for reducing errors in biosynthetic processes requiring high specificity,” *Proceedings of the National Academy of Sciences* **71**, 4135–4139 (1974).
- [24] Michael Samoilov, Sergey Plyasunov, and Adam P Arkin, “Stochastic amplification and signaling in enzymatic futile cycles through noise-induced bistability with oscillations,” *Proceedings of the National Academy of Sciences* **102**, 2310–2315 (2005).
- [25] Taeko Nishiwaki, Yoshinori Satomi, Masato Nakajima, Cheolju Lee, Reiko Kiyohara, Hakuto Kageyama, Yohko Kitayama, Mioko Temamoto, Akihiro Yamaguchi, Atsushi Hijikata, *et al.*, “Role of KaiC phosphorylation in the circadian clock system of *Synechococcus elongatus* PCC 7942,” *Proceedings of the National Academy of Sciences* **101**, 13927–13932 (2004).
- [26] Rekha Pattanayek, Jimin Wang, Tetsuya Mori, Yao Xu, Carl Hirschie Johnson, and Martin Egli, “Visualizing a circadian clock protein: crystal structure of KaiC and functional insights,” *Molecular cell* **15**, 375–388 (2004).
- [27] Terrell L Hill, *Free energy transduction and biochemical cycle kinetics* (Springer-Verlag New York Inc., 1989).
- [28] Yao Xu, Tetsuya Mori, and Carl Hirschie Johnson, “Cyanobacterial circadian clockwork: roles of KaiA, KaiB and the kaiBC promoter in regulating KaiC,” *The EMBO Journal* **22**, 2117–2126 (2003).
- [29] Masahiro Ishiura, Shinsuke Kutsuna, Setsuyuki Aoki, Hideo Iwasaki, Carol R Andersson, Akio Tanabe, Susan S Golden, Carl H Johnson, and Takao Kondo, “Expression of a gene cluster kaiABC as a circadian feedback process in cyanobacteria,” *Science* **281**, 1519–1523 (1998).
- [30] Esther M Conwell, “Negative differential conductivity,” *Physics Today* **23**, 35–41 (1970).
- [31] Marcin Kostur, Lukasz Machura, Peter Hänggi, Jerzy Luczka, and Peter Talkner, “Forcing inertial brownian motors: Efficiency and negative differential mobility,” *Physica A: Statistical Mechanics and*

- its Applications **371**, 20–24 (2006).
- [32] Robert L Jack, David Kelsey, Juan P Garrahan, and David Chandler, “Negative differential mobility of weakly driven particles in models of glass formers,” *Physical Review E* **78**, 011506 (2008).
 - [33] RKP Zia, Eigil Luxhøj Praestgaard, and OG Mouritsen, “Getting more from pushing less: Negative specific heat and conductivity in nonequilibrium steady states,” *American Journal of Physics* **70**, 384–392 (2002).
 - [34] Xingbo Yang, Matthias Heinemann, Jonathon Howard, Greg Huber, Srividya Iyer-Biswas, Guillaume Le Treut, Michael Lynch, Kristi L Montooth, Daniel J Needleman, Simone Pigolotti, *et al.*, “Physical bioenergetics: Energy fluxes, budgets, and constraints in cells,” *Proceedings of the National Academy of Sciences* **118**, e2026786118 (2021).
 - [35] Lukas Oberreiter, Udo Seifert, and Andre C Barato, “Universal minimal cost of coherent biochemical oscillations,” *Physical Review E* **106**, 014106 (2022).
 - [36] Susan E Cohen and Susan S Golden, “Circadian rhythms in cyanobacteria,” *Microbiology and Molecular Biology Reviews* **79**, 373–385 (2015).
 - [37] Jeroen S van Zon, David K Lubensky, Pim RH Altena, and Pieter Rein ten Wolde, “An allosteric model of circadian KaiC phosphorylation,” *Proceedings of the National Academy of Sciences* **104**, 7420–7425 (2007).
 - [38] Congxin Li, Xiaofang Chen, Pengye Wang, and Weichi Wang, “Circadian KaiC phosphorylation: a multi-layer network,” *PLoS computational biology* **5**, e1000568 (2009).
 - [39] Gary K Chow, Archana G Chavan, Joel Heisler, Yong-Gang Chang, Ning Zhang, Andy LiWang, and R David Britt, “A night-time edge site intermediate in the cyanobacterial circadian clock identified by EPR spectroscopy,” *Journal of the American Chemical Society* **144**, 184–194 (2022).
 - [40] Paul Smolen, Douglas A Baxter, and John H Byrne, “Modeling circadian oscillations with interlocking positive and negative feedback loops,” *Journal of Neuroscience* **21**, 6644–6656 (2001).
 - [41] Archana G Chavan, Jeffrey A Swan, Joel Heisler, Cigdem Sancar, Dustin C Ernst, Mingxu Fang, Joseph G Palacios, Rebecca K Spangler, Clive R Bagshaw, Sarvind Tripathi, *et al.*, “Reconstitution of an intact clock reveals mechanisms of circadian timekeeping,” *Science* **374**, eabd4453 (2021).

REVIEWER COMMENTS

Reviewer #1 (Remarks to the Author):

The authors have presented a lengthy response to the referees' comments and I think the discussion has helped clarify the paper a great deal. I like the introduction of the Zak phase to characterise the topology. Overall I think this is a stimulating piece of work and I recommend publication.

Reviewer #2 (Remarks to the Author):

Second review of "A topological mechanism for robust and efficient global oscillations in biological networks" Chongbin Zheng & Evelyn Tang, by reviewer 2.

Improvements:

I see that the figures describing the model have been re-drawn, and are strictly better than before.

The dynamical variables & number of independent couplings are now clearly stated.

The fact that the model is mathematically equivalent to the earlier PRX one is clearly stated, although on page 4. The introduction still makes no mention of this close connection to earlier work (ref [33] is given without explanation).

Concerns:

What's new compared to the PRX article, besides extra complexity? I think the short answer is the comparison to MWC, of which the clearest piece seems to be Figure S6. I have not understood why those curves look different to Fig 3e, nor why the rest of Fig 3 is interesting. Nor really why the curves are good — the enlarged explanation about ΔS going down as μ is increased seems to say that it's good because ΔS goes down as μ is increased. Why does it matter if the low-driving regime has higher or lower entropy production, if the real system won't be operating in this regime? The one clear point is

that the sign of the slope might be a nice experimental prediction, if you could perturb a wild system away from its design point, and test.

I share reviewer 1's concern about whether this ought to be called topological. I see the note added about the Zak phase, but am not sure this answers the question. Is the claim that the model's dynamics preserve this phase, for all ρ ? Or only in the "topological regime ($\rho > 0$)", or only at large ρ ? Or is this just an elaborate way of detecting whether you are in an edge state? But I am not an expert, and will defer to whether reviewer 1 considers this answered (perhaps by reading this paper and the PRX one together).

I remain perturbed that the description of the model seems intent on muddling biological motivations in with telling us what the model actually is. For example (but there are many!) new text in the Fig 1 caption tells us about why the curly line from the internal label is attached to a particular monomer... this is a confusing graphical choice, but is in no way a feature of the mathematical model being defined! However, such mixing of concerns appears to be an explicit goal which the authors do not intend to change.

I find the repeated insertion of "parsimonious" surprising, as the model is larded with all sorts of complexity, perhaps to please biologically minded readers? Certainly there are simpler models which work equally well — their own PRX paper published one! The details obscure the core of how it works: The core N-E-S-W-N cycle which drives "external" transition is *designed* to produce edge states, and it does.

I still find this quite hard to read. The entire first page seems like it could describe almost anything.

Recommendation:

I feel this should be rejected. It strikes me as an incremental addition to an earlier model. Perhaps there is room for a paper selling these ideas to a new audience, but this paper doesn't do that well either.

It would be interesting if edge states like these proved to have biological relevance, to circadian clocks or elsewhere.

Response to the Referees for “A topological mechanism for robust and efficient global oscillations in biological networks”

Chongbin Zheng^{1,2}, Evelyn Tang^{1,2,*}

1 Center for Theoretical Biological Physics
Rice University, Houston, Texas 77005, USA

2 Center for Theoretical Biological Physics
Rice University, Houston, Texas 77005, USA

* E-mail: Corresponding e.tang@rice.edu

I. REVIEWER 1

The authors have presented a lengthy response to the referees' comments and I think the discussion has helped clarify the paper a great deal. I like the introduction of the Zak phase to characterise the topology. Overall I think this is a stimulating piece of work and I recommend publication.

Thank you! We appreciated your comments and recommendation for publication.

II. REVIEWER 2

Second review of "A topological mechanism for robust and efficient global oscillations in biological networks" Chongbin Zheng & Evelyn Tang, by reviewer 2.

Improvements:

I see that the figures describing the model have been re-drawn, and are strictly better than before. The dynamical variables & number of independent couplings are now clearly stated.

Thank you for your positive comments and useful suggestions that helped improve the clarity of the manuscript.

The fact that the model is mathematically equivalent to the earlier PRX one is clearly stated, although on page 4. The introduction still makes no mention of this close connection to earlier work (ref [33] is given without explanation).

We have added a clarification in the third paragraph in the introduction to mention the close connection of our model to the PRX one: "We expand on the topological model first proposed in [1] and apply it to the KaiABC system, which regulates the circadian rhythm of cyanobacteria [2]."

Concerns:

What's new compared to the PRX article, besides extra complexity? I think the short answer is the comparison to MWC, of which the clearest piece seems to be Figure S6. I have not understood why those curves look different to Fig 3e, nor why the rest of Fig 3 is interesting. Nor really why the curves are good — the enlarged explanation about ΔS going down as μ is increased seems to say that it's good because ΔS goes down as μ is increased. Why does it matter if the low-driving regime has higher or lower entropy production, if the real system won't be operating in this regime? The one clear point is that the sign of the slope might be a nice experimental prediction, if you could perturb a wild system away from its design point, and test.

- We thank the reviewer for expressing these concerns and take the opportunity to state the various new contributions from our paper more clearly and concisely in the Introduction section, listed below:
 - In the third paragraph: "Towards addressing these questions, we propose a mechanism for the dimensional reduction of phase space to robust attractors and long oscillations. In particular, we examine the strongly non-equilibrium feature of a topological model that consists of interlinked "futile cycles", common non-equilibrium motifs that are so named due to their consumption of ATP only to repeat the same states in the cycle [3, 4]. Yet, these motifs are pervasive in many different biological systems from metabolism to sensory systems, muscular contraction, and protein synthesis. Here, we show that such non-equilibrium motifs can link in a topological manner to support emergent global dynamics that move in a directed fashion around the network boundary. We expand on the topological model first proposed in [1] and apply it to the KaiABC system, which

regulates the circadian rhythm of cyanobacteria [2]. The localized edge currents naturally reproduce the kinetic ordering of KaiC phosphorylation cycles [5, 6]. Our work is the first in the literature that connects topological models to a concrete biological system. It also features few parameters compared to alternative models of KaiABC [7, 8] while displaying higher oscillation coherence.”

- At the end of the fourth paragraph: “In this paper, we provide an explicit topological invariant for our model, and demonstrate the robustness of the emerging oscillations with changing transition rates. We also introduce new methods inspired by topological band theory to characterize oscillation coherence.”
 - In the fifth paragraph of the new manuscript: “The characteristic behavior of our topological model leads to several experimentally testable predictions. For instance, the localization of system dynamics on the edge leads to a novel regime with increasing coherence alongside decreasing energy dissipation as the external driving is increased. This unusual behavior, i.e., “getting more from pushing less”, has been observed in other scenarios such as driven electronic, Brownian, and glassy systems [9–12], while its occurrence in a stochastic system like ours remains surprising. In this regime, we expect a decrease in total ATP consumption (energy dissipation) in response to an increasing ATP concentration (external driving), a unique signature of our model. Additional experimental predictions are discussed later in the paper, which provide further tests for whether topology as illustrated in our model contributes to the robustness of circadian oscillations.”
 - We have also removed the fifth paragraph in the original manuscript, and simplified the end of the first paragraph: “Understanding the principles that govern emergent oscillations is crucial as disruptions in biological clocks lead to decreased health and reproductive fitness in multiple organisms [13–17].”
- We agree with the reviewer that the dependence of ΔS on μ offers a nice experimental prediction. We have now emphasized its significance in the third paragraph of Sec. III reproduced below. We have also added black dash-dotted arrows to Fig. 3c and 3d to highlight this prediction more visually. Fig. 3 and updated captions are reproduced in this document as Fig. 1.
 - On the contrary, our model in the topological regime displays an unusual regime where coherence increases while entropy production per period becomes lower. Fig. 3a shows the coherence of our model as a function of the two parameters μ and ρ (also see Supplementary Fig. S4 featuring both positive and negative ρ). Increasing the thermodynamic force increases coherence monotonically, as expected for typical oscillator models [19]. Going deeper into the topological regime by increasing ρ also leads to higher coherence, as the global currents become more localized on the system edge [20]. However, the entropy production per period ΔS does not change monotonically with μ . As illustrated in Fig. 3b in a smaller lattice, the system response becomes localized to the edge as μ increases. This causes the entropy production on the edge (blue arrows) to increase, while the entropy production in the bulk (orange arrows) decreases, in the region $1.5 < \mu < 7$. Since the bulk contribution typically dominates the edge contribution, i.e., $\mathcal{O}(N^2) \gg \mathcal{O}(N)$ where N is the typical system size, the sum of their contributions also decreases (green curve in Fig. 3c). This negative slope of ΔS with respect to μ implies that the system dissipates less free energy overall even when the external driving μ supplied to each reaction is stronger. In other words, we “get more from pushing less” [9–12]. This unusual regime has a topological origin, since the decrease in the bulk entropy production results from the localization of the steady state onto the system edge. This unusual negative

FIG. 1. (Fig. 3 in the manuscript.) Coherence and an efficient regime with simultaneously increased precision and decreased cost. **a**, Phase diagram of coherence \mathcal{R} for the topological model, which increases with respect to thermodynamic force μ and ratio of timescales ρ as expected. **b**, Entropy production ΔS moves from the bulk to the boundary of the system with increasing μ , illustrated on a smaller lattice. The system is in the topological regime; $\rho = 5$. Blue arrows represent entropy production on the edge ΔS_e and orange arrows represent entropy production in the bulk ΔS_b . Arrow thickness corresponds to the magnitude of the entropy production for the corresponding transitions and arrows point in the direction of the probability flux. **c**, Entropy production per period ΔS for the topological model ($N_x = N_y = 6$) is decomposed into a bulk and an edge contribution. Due to the localization effects in Fig. 3b, the entropy production on the edge ΔS_e (blue) increases while entropy production in the bulk ΔS_b (orange) decreases with μ around $1.5 < \mu < 7$. Typically, the bulk contribution dominates the edge contribution, hence their sum ΔS (green) also decreases with increasing μ . Experimentally, the negative slope in this regime predicts a decreased overall ATP consumption when ATP concentration is increased, as indicated by the black dash-dotted arrow. **d**, Phase diagram of the cost-precision tradeoff $\Delta S/\mathcal{R}$ [18]. The black dash-dotted arrow, similar to that in **c**, indicates increasing μ in the energy-efficient regime. This leads to a lower $\Delta S/\mathcal{R}$, which implies a more cost-effective oscillator. There is a global optimum for $\Delta S/\mathcal{R}$ at $\mu = 6.5, \rho = 4.1$. **e**, Comparison of coherence for different KaiC models with randomly sampled parameters (individual points): there is a strongly-driven regime where the topological model has the highest coherence. The dashed lines represent the upper bounds on coherence for the corresponding models in terms of the thermodynamic force μ [19] (see Methods). The purple dashed line is the upper bound for the 28-state unicyclic model, which is approached by the edge state in the strongly topological regime. Values of ΔS in all panels are given in units of $\gamma_{tot} k_B$.

slope leads to a unique experimental signature. Where ΔS has a negative slope, increasing the ATP concentration (increasing μ) leads to a decrease in total ATP consumption (decreasing ΔS). This is a striking prediction of our topological model, as shown by the dash-dotted arrow in Fig. 3c. In addition, since coherence increases monotonically with μ , this leads to an efficient regime with simultaneously increasing coherence and

decreasing cost. The cost-effectiveness in terms of expending free energy for coherent oscillations can be measured by the ratio $\Delta S/\mathcal{R}$ [18]. See Fig. 3d for the region in parameter space with low $\Delta S/\mathcal{R}$, indicating a highly efficient oscillator.

- We have also reorganized our experimental predictions more thoroughly in the second paragraph of Sec. IV, reproduced below.
 - We further predict testable signatures of topological edge states in biological oscillators and the KaiABC system. By changing the ATP concentration or ATP/ADP ratio, we can tune the thermodynamic force μ into the regime where ΔS has a negative slope in μ (dash-dotted arrow in Fig. 3c). In this regime, we expect to measure decreased ATP consumption as ATP concentration is increased, a unique feature of our topological model. Moreover, examining our steady-state distribution (Fig. 1d), we expect to find mixtures of molecules with specific fractions of differentially phosphorylated subunits, which may be measured by mass spectrometry [21]. Further, KaiB-bound KaiC should be less abundant than unbound KaiC, because the slow γ_{in} transitions cause a buildup of the latter before they bind to KaiB. Lastly, by perturbing the system or initializing it in the bulk of the state space with partially phosphorylated T and S sites, we should see a transient increase in ATP consumption or energy dissipation as the system goes through rapid cycles of external transitions within the system bulk before converging to the edge. Monomer phosphorylation cycles in the reverse direction from the global cycle may also be observed, as the external cycles in the system bulk have the opposite chirality from our global cycle on the system edge (see Fig. 1c).
- The curve in Fig. S6 shows coherence as a function of μ with other parameters held fixed, while the dashed curve in Fig. 3e corresponds to the maximum possible coherence for any parameter combinations [19]. We clarify this in Sec. V of the Supplementary Information, which is reproduced below.
 - In this section we discuss how the free energy cost (quantified by ΔS) and the precision (quantified by \mathcal{R}) of various KaiC models change with the external driving μ . For the MWC model, the entropy production rate σ of the network can be obtained by a cycle decomposition method [19, 22]:

$$\sigma = \eta\gamma \sum_{i=0}^5 (e^{\eta/2} p_{C_i}^s - e^{E/6} p_{C_{i+1}}^s),$$

where $p_{C_i}^s$ is the steady-state probability at the state C_i . This expression is used to calculate the entropy production per period $\Delta S = \sigma\mathcal{T}$ for the MWC model. As shown in the left column of Fig. S6, ΔS monotonically increases with \mathcal{R} , as expected. Coherence, however, is not monotonic in μ . When μ is sufficiently large, the MWC model ends up maintaining less coherent oscillations with increased driving and energetic cost. **Note that the \mathcal{R} curve is different from the red dashed line shown in Fig. 3e in the main text. Here we plot \mathcal{R} with (η, E) held fixed, while the dashed line in Fig. 3e shows the maximum coherence for each μ for any parameter combinations (also see Methods).** In contrast to the MWC model, coherence for the topological model increases monotonically in μ for a fixed ρ . ΔS , however, is non-monotonic, supporting a regime with increasing coherence and simultaneously decreasing cost (right column of Fig. S6). This unusual regime has its origin from the topological protection of the probability currents on the edge, which effectively reduces the state space into a one-dimensional cycle along the edge. This edge localization leads to a lower free energy cost but more coherent oscillations.

- We are glad to take the opportunity to explain the significance of Fig. 3 more clearly in the third paragraph of Sec. III. Changes we have made are listed below.

- For Fig. 3a: “Fig. 3a shows the coherence of our model as a function of the two parameters μ and ρ (also see Supplementary Fig. S4 featuring both positive and negative ρ). Increasing the thermodynamic force increases coherence monotonically, as expected for typical oscillator models [19]. Going deeper into the topological regime by increasing ρ also leads to higher coherence, as the global currents become more localized on the system edge [20].”
- For Fig. 3b: “This unusual regime has a topological origin, since the decrease in the bulk entropy production results from the localization of the steady state onto the system edge.”
- As the reviewer mentioned, Fig. 3c is important because of the unique experimental prediction it provides. Hence we have emphasized: “This unusual negative slope leads to a unique experimental signature. Where ΔS has a negative slope, increasing the ATP concentration (increasing μ) leads to a decrease in total ATP consumption (decreasing ΔS). This is a striking prediction of our topological model, as shown by the dash-dotted arrow in Fig. 3c.”
- Fig. 3d: “In addition, since coherence increases monotonically with μ , this leads to an efficient regime with simultaneously increasing coherence and decreasing cost. The cost-effectiveness in terms of expending free energy for coherent oscillations can be measured by the cost-precision tradeoff $\Delta S/\mathcal{R}$ [18]. See Fig. 3d for the region in parameter space with low $\Delta S/\mathcal{R}$, indicating a highly cost-effective oscillator.”

I share reviewer 1’s concern about whether this ought to be called topological. I see the note added about the Zak phase, but am not sure this answers the question. Is the claim that the model’s dynamics preserve this phase, for all ρ ? Or only in the “topological regime ($\rho > 0$)”, or only at large ρ ? Or is this just an elaborate way of detecting whether you are in an edge state? But I am not an expert, and will defer to whether reviewer 1 considers this answered (perhaps by reading this paper and the PRX one together).

- We are glad that Reviewer 1 has considered this concern about the topology answered. To further clarify, our model is topological with the 2D Zak phase serving as a proper topological invariant. We have stated more clearly what values the 2D Zak phase takes in different dynamical regimes, in the sixth paragraph of Sec. II. The second half of the paragraph is reproduced below.
 - The edge state and the associated dynamical regime have a topological origin, as their emergence is governed by the presence of a topological invariant, the 2D Zak phase [1, 23]. For our topological model, the 2D Zak phase takes the trivial value $(0, 0)$ when $\rho < 0$ but becomes (π, π) in the topological regime $\rho > 0$ (see Supplementary Information for more details). As ρ becomes larger in the topological regime, the Zak phase (π, π) is preserved, while the edge state becomes more localized on the system boundary [20]. The edge state further inherits the useful property of topological protection from inaccessible states [1] or perturbations in transition rates (see Supplementary Fig. S1 and S2). Because of these unusual properties, we focus on the topological regime and investigate the properties of our system assuming $\rho > 0$ from now on.

I remain perturbed that the description of the model seems intent on muddling biological motivations in with telling us what the model actually is. For example (but there are many!) new text in the Fig 1 caption tells us about why the curly line from the internal label is attached to a particular monomer... this is a confusing graphical choice, but is in no way a feature of the mathematical model being defined! However, such mixing of concerns appears to be an explicit goal which the authors do not intend to change.

FIG. 2. (Fig. 1 in the manuscript.) Topological model for emergent oscillations, illustrated with KaiABC that regulates the circadian rhythm. **a**, Based on the prevalence of non-equilibrium cycles observed in various biological systems [3, 4], we hypothesize that KaiC monomers undergo phosphorylation and dephosphorylation cycles (black arrows γ_{ex} and slower reverse transitions γ'_{ex} , where two types of phosphorylation T and S are shown with the addition of orange and pink circles and the numbers of each are given in brackets. We only keep track of the total phosphorylation levels in T and S, while the location of each phosphorylated monomer can be arbitrary. **b**, Within a given phosphorylation level, internal transitions (grey arrows, γ_{in} and γ'_{in}) can take place due to conformational changes (illustrated by circle and square shapes) or ligand binding (e.g., of KaiA or KaiB). The internal state labels the direction of the next external transition that it primes. We represent the interaction with KaiA with a curly arrow because KaiA promotes phosphorylation by rapid association and dissociation rather than binding to fixed KaiC hexamers [24]. In both **a** and **b**, the curly line from the internal label attaches to the monomer **most likely to be modified next, indicating that the monomer is tagged by molecular modifications (e.g. KaiA or KaiB binding) and primed for subsequent change.** **c**, These cycles can be laid out in a lattice, with T phosphorylation along the horizontal axis and S phosphorylation along the vertical axis, while each set of four internal transitions (e.g., in the blue box corresponding to **b**) repeats along these axes. The external cycle in **a** is also highlighted in the green box. **d**, This lattice allows probing of its topological properties. In the ordinary case with similar rates throughout, i.e. $\gamma_{ex} \sim \gamma_{in}$, the system will perform a random walk ergodically through the phase space. In the topological regime when $\gamma_{ex} \gg \gamma_{in}$, once the system hits an edge it will continue around the edge, as can be verified by inspection. We plot the steady state in the topological regime which lies on the system edge, taking $\mu = 1, \rho = 2, \gamma_{tot} = 1$. There are 28 states along the edge with high probability, labeled by the order in which they are traversed in a typical trajectory. These trajectories form a global current along the edge of the state space.

- We appreciate the opportunity to explicitly lay out graphical choices. Our labeling of the internal states is based on identification of concrete biological mechanisms, which we clarify in the caption for Fig. 1b in the manuscript: “In both **a** and **b**, the curly line from the internal label attaches to the monomer **most likely to be modified next, indicating that the monomer is tagged by molecular modifications (e.g. KaiA or KaiB binding) and primed for subsequent change.**”
- For consistency, we have moved the internal tags for the $(2, 1)_S$ and $(2, 1)_E$ states to different monomers, which is now updated in this document as Fig. 2.
- As the reviewer mentions, the bare-bones of the model has already been laid out in the PRX paper. Here we would like to point out the biological context as being a central contribution of the paper. This is highlighted in the third paragraph in Sec. I: “**Our work is the first in the literature that connects topological models to a concrete biological system**” and the first paragraph of Sec. IV: “**The mathematical model, which first appeared in [1], is interpreted in a biological context and applied to the KaiABC system.**” This work is intended for an interdisciplinary audience for whom connecting between the model and the biological implications is of great importance. It is also intended to motivate experimental testing and dialogue with experimentalists. Therefore, the combination with biology is a deliberate choice that we hope the reviewer can appreciate, as motivating experimental verification is an important endeavor for useful modeling and fundamental theoretical work.

*I find the repeated insertion of “parsimonious” surprising, as the model is larded with all sorts of complexity, perhaps to please biologically minded readers? Certainly there are simpler models which work equally well — their own PRX paper published one! The details obscure the core of how it works: The core N-E-S-W-N cycle which drives “external” transition is *designed* to produce edge states, and it does.*

- The word “parsimonious” is in relation to other biological models like the MWC-type models [7, 8]. In contrast, our model uses a simple parametrization with just two parameters. The PRX paper features an abstract model that lacks biophysical interpretation, and hence we do not compare our model with the PRX model. We clarify this in the first paragraph of Sec. IV, reproduced below. After all, besides the “core of how it works”, understanding the physical manifestation of fundamental physics principles is an important theoretical contribution.
 - We have proposed a topological model that generates coherent oscillations, which supports an unusual regime with increased coherence and simultaneously decreased energetic cost. **The mathematical model, which first appeared in [1], is interpreted in a biological context and applied to the KaiABC system. Compared to other KaiABC models, our model has high coherence and more closely saturates a global spectral bound, similar to the most coherent unicyclic models. We also find that the imaginary spectral gap can be used to predict oscillation coherence.** Further, the kinetic ordering of the KaiABC phosphorylation cycle arises naturally as an edge current in our model. In contrast to typical MWC-type models **which usually involve a large number of system-dependent parameters [7, 8], our model is parsimonious in that it captures the same phosphorylation dynamics with only a few parameters. Moreover, it also does not require fine-tuning of the reaction rates or restriction of the configuration space to all-or-none conformational change, an assumption that is not supported by emerging experimental evidence [25].**

I still find this quite hard to read. The entire first page seems like it could describe almost anything.

We have revised the manuscript again based on the helpful suggestions of the referee.

- We have revised the introduction by making more specific statements highlighting the new contributions of our paper and explaining the connection between our work and the PRX model (see pages 2-3 above).
- We have added more discussions and visualizations on the unique experimental signature of the unusual regime observed in Fig. 3.
- We have clarified the role of the 2D Zak phase as a topological invariant that only changes value when the system switches to a different dynamical regime.
- We have clarified the significance of different subfigures in Fig. 3.

Recommendation:

I feel this should be rejected. It strikes me as an incremental addition to an earlier model. Perhaps there is room for a paper selling these ideas to a new audience, but this paper doesn't do that well either.

As the reviewer says, there is certainly room to bring these ideas to a new audience, since living matter is an important field with many open questions. We are grateful for the reviewer's suggestions on how to improve the presentation.

It would be interesting if edge states like these proved to have biological relevance, to circadian clocks or elsewhere.

We agree with the reviewer and consider this theoretical paper as a key step towards creating a dialogue with experimentalists who can test and measure its biological relevance. We would like to emphasize that this is the very first paper that connects topological models to a concrete biological system, which has never been done before. Our work proposes experimental tests and novel signatures, and hence is a necessary step towards this goal that the reviewer shares.

-
- [1] Evelyn Tang, Jaime Agudo-Canalejo, and Ramin Golestanian, "Topology protects chiral edge currents in stochastic systems," *Physical Review X* **11**, 031015 (2021).
 - [2] Masahiro Ishiura, Shinsuke Kutsuna, Setsuyuki Aoki, Hideo Iwasaki, Carol R Andersson, Akio Tanabe, Susan S Golden, Carl H Johnson, and Takao Kondo, "Expression of a gene cluster kaiABC as a circadian feedback process in cyanobacteria," *Science* **281**, 1519–1523 (1998).
 - [3] John J Hopfield, "Kinetic proofreading: a new mechanism for reducing errors in biosynthetic processes requiring high specificity," *Proceedings of the National Academy of Sciences* **71**, 4135–4139 (1974).
 - [4] Michael Samoilov, Sergey Plyasunov, and Adam P Arkin, "Stochastic amplification and signaling in enzymatic futile cycles through noise-induced bistability with oscillations," *Proceedings of the National Academy of Sciences* **102**, 2310–2315 (2005).
 - [5] Michael J Rust, Joseph S Markson, William S Lane, Daniel S Fisher, and Erin K O'Shea, "Ordered phosphorylation governs oscillation of a three-protein circadian clock," *Science* **318**, 809–812 (2007).
 - [6] Susan E Cohen and Susan S Golden, "Circadian rhythms in cyanobacteria," *Microbiology and Molecular Biology Reviews* **79**, 373–385 (2015).
 - [7] Jeroen S van Zon, David K Lubensky, Pim RH Altena, and Pieter Rein ten Wolde, "An allosteric model of circadian KaiC phosphorylation," *Proceedings of the National Academy of Sciences* **104**, 7420–7425 (2007).
 - [8] Joris Pajmans, David K Lubensky, and Pieter Rein Ten Wolde, "A thermodynamically consistent model of the post-translational Kai circadian clock," *PLoS computational biology* **13**, e1005415 (2017).
 - [9] Esther M Conwell, "Negative differential conductivity," *Physics Today* **23**, 35–41 (1970).

- [10] Marcin Kostur, Lukasz Machura, Peter Hänggi, Jerzy Luczka, and Peter Talkner, “Forcing inertial brownian motors: Efficiency and negative differential mobility,” *Physica A: Statistical Mechanics and its Applications* **371**, 20–24 (2006).
- [11] Robert L Jack, David Kelsey, Juan P Garrahan, and David Chandler, “Negative differential mobility of weakly driven particles in models of glass formers,” *Physical Review E* **78**, 011506 (2008).
- [12] RKP Zia, Eigil Luxhøj Praestgaard, and OG Mouritsen, “Getting more from pushing less: Negative specific heat and conductivity in nonequilibrium steady states,” *American Journal of Physics* **70**, 384–392 (2002).
- [13] Anna M Puszynska and Erin K O’Shea, “Switching of metabolic programs in response to light availability is an essential function of the cyanobacterial circadian output pathway,” *elife* **6**, e23210 (2017).
- [14] Rachel M Green, Sonia Tingay, Zhi-Yong Wang, and Elaine M Tobin, “Circadian rhythms confer a higher level of fitness to *Arabidopsis* plants,” *Plant physiology* **129**, 576–584 (2002).
- [15] Melanie Horn, Oliver Mitesser, Thomas Hovestadt, Taishi Yoshii, Dirk Rieger, and Charlotte Helfrich-Förster, “The circadian clock improves fitness in the fruit fly, *Drosophila melanogaster*,” *Frontiers in Physiology* **10**, 1374 (2019).
- [16] Frank AJL Scheer, Michael F Hilton, Christos S Mantzoros, and Steven A Shea, “Adverse metabolic and cardiovascular consequences of circadian misalignment,” *Proceedings of the National Academy of Sciences* **106**, 4453–4458 (2009).
- [17] Selma Masri and Paolo Sassone-Corsi, “The emerging link between cancer, metabolism, and circadian rhythms,” *Nature medicine* **24**, 1795–1803 (2018).
- [18] Lukas Oberreiter, Udo Seifert, and Andre C Barato, “Universal minimal cost of coherent biochemical oscillations,” *Physical Review E* **106**, 014106 (2022).
- [19] Andre C Barato and Udo Seifert, “Coherence of biochemical oscillations is bounded by driving force and network topology,” *Physical Review E* **95**, 062409 (2017).
- [20] Aleksandra Nelson and Evelyn Tang, “Non-reciprocity permits edge states and strong localization in stochastic topological systems,” *arXiv preprint arXiv:2310.16720* (2023).
- [21] Shahid Mehmood, Timothy M Allison, and Carol V Robinson, “Mass spectrometry of protein complexes: from origins to applications,” *Annual review of physical chemistry* **66**, 453–474 (2015).
- [22] Jürgen Schnakenberg, “Network theory of microscopic and macroscopic behavior of master equation systems,” *Reviews of Modern physics* **48**, 571 (1976).
- [23] Feng Liu and Katsunori Wakabayashi, “Novel topological phase with a zero berry curvature,” *Physical review letters* **118**, 076803 (2017).
- [24] Hakuto Kageyama, Taeko Nishiwaki, Masato Nakajima, Hideo Iwasaki, Tokitaka Oyama, and Takao Kondo, “Cyanobacterial circadian pacemaker: Kai protein complex dynamics in the KaiC phosphorylation cycle in vitro,” *Molecular cell* **23**, 161–171 (2006).
- [25] Xu Han, Dongliang Zhang, Lu Hong, Daqi Yu, Zhaolong Wu, Tian Yang, Michael Rust, Yuhai Tu, and Qi Ouyang, “Determining subunit-subunit interaction from statistics of cryo-em images: observation of nearest-neighbor coupling in a circadian clock protein complex,” *Nature Communications* **14**, 5907 (2023).

Reviewers' comments:

Reviewer #2 (Remarks to the Author):

Third review of "A topological mechanism for robust and efficient global oscillations in biological networks" Chongbin Zheng & Evelyn Tang, by reviewer 2.

I appreciate the effort put into revisions, but my recommendation remains the same:

"I feel this should be rejected. It strikes me as an incremental addition to an earlier model. Perhaps there is room for a paper selling these ideas to a new audience, but this paper doesn't do that well either."

In reply to a few comments & changes:

> We have added a clarification in the third paragraph in the introduction to mention the close connection of our model to the PRX one: "We expand on the topological model first proposed in [1] and apply it to the KaiABC system, which regulates the circadian rhythm of cyanobacteria [2]."

This still hides things. Ref [1] also applies this to the KaiABC system -- section IV.A, page 9. What's novel is comparison to other models, I think.

> In both a and b, the curly line from the internal label attaches to the monomer most likely to be modified next, indicating that the monomer is tagged by molecular modifications (e.g. KaiA or KaiB binding) and primed for subsequent change.

Here I think you misunderstand my complaint about graphical choices. It's not that they must all be justified somewhere in the text. It's that the poor reader needs to know as soon as possible to ignore them, since they are in no way part of the model. I object to the use of extraneous detail to hide how the model actually works, and render its emergent behaviours more mysterious than they really are.

Reading everything again, my perhaps my core complaint is this one:

>> The details obscure the core of how it works: The core N-E-S-W-N cycle which drives “external” transition is *designed* to produce edge states, and it does.

It’s very clear to me now why this cycle of internal states is desirable for producing edge states. But is it possible to engineer such a thing out of biologically plausible components? That is, a cycle of states which control the direction of allowed “external” transitions.

This is something the PRX paper does not address, and if the intent is to publish something with more biological detail, then perhaps this paper ought to contain such a mechanism. The only mention of this I can find is this (on page 3, right column) which seems more like speculation on the ingredients nature would have to use, rather than a description of how it's possible to assemble them to produce this quite specific circuit:^[1]_{SEP}

> For the KaiABC system, the internal transitions could correspond to interactions with KaiA and KaiB proteins [51] and conformational changes of a loop-like structure on KaiC called the A-loop (two conformations represented by blue squares and purple circles in Fig. 1 respectively) [52]. These transitions modify the internal state of the system, changing its tendency to phosphorylate or dephosphorylate and whether in T or S.

Response to the Referees for “A topological mechanism for robust and efficient global oscillations in biological networks”

Chongbin Zheng^{1,2}, Evelyn Tang^{1,2,*}

1 Center for Theoretical Biological Physics
Rice University, Houston, Texas 77005, USA

2 Center for Theoretical Biological Physics
Rice University, Houston, Texas 77005, USA

* E-mail: Corresponding e.tang@rice.edu

Third review of “A topological mechanism for robust and efficient global oscillations in biological networks” Chongbin Zheng & Evelyn Tang, by reviewer 2.

I appreciate the effort put into revisions, but my recommendation remains the same:

“I feel this should be rejected. It strikes me as an incremental addition to an earlier model. Perhaps there is room for a paper selling these ideas to a new audience, but this paper doesn’t do that well either.”

We would like to address the concerns about our work being an incremental addition. Indeed, our previous work mentions KaiABC – we know that since we wrote it. However, what we wrote previously was a hypothesis without any connection with KaiABC literature. Our previous work does not relate to any known molecular reactions or experimentally tunable parameters – the model there is **unusable for any real biological system**. The model in the PRX did not specify the biological mechanism for any of its components and transition rates were given without any justification as to their origin. In this as in all the other papers in the field so far on topological models related to biology or stochastic systems (including one published in Nature Communications 2017), **the claim of topology to relate to a biological system remains inspirational at best**, without any guidance of how to measure or access the topological properties.

Our manuscript responds to this glaring lack in the field, by **providing detailed biophysical mechanisms** which are the only path towards any experimental comparison. We furnish a full new phenomenology for the model and **a thoughtful analysis of its properties and consequences in the new context of molecular biology**. Further, we demonstrate its parameter space and how to tune into and out of the topological transition, in dialogue with recent data and experimental knobs. We propose pathways to alter the transition rates (by changing ATP concentration or using a mutant) and hence drive the system away from or closer to the topological phase, which will affect the period and robustness of the oscillation. We explain the delicate interaction of phosphorylation with conformational changes, which is a necessary and sufficient condition for the observed oscillatory cycle, in a simple yet biophysically plausible way – to understand an important system that had until now required very complicated models.

Most excitingly, our work is the first in the field that relates a topological mechanism to a concrete biological system in an operational manner. It provides a blueprint for how the entire model can operate, with clear relationships between the model parameters and actual biological or experimental components. This analysis and insights dramatically open up what is possible, allowing for experimental demonstration and verification. For instance, we now show how to tune the transition rates of γ_{ex} and γ_{in} which can drive the system into the oscillatory and topological regime. In other words, we provide a simple mechanism for dimensional reduction of the network onto the oscillatory cycle, and a proposal for tuning it. Our work is the first to demonstrate how topology can be realized in a biological system, which is a desirable objective given the **many attractive properties of topology such as its robust properties**.

Our connection of topology to real and measurable quantities in a biology system reveals many theoretical and conceptual consequences that go far beyond the specific system we study. Where the measurable outcomes in quantum and other systems are that of electrical conductivity or floppy modes in a spring network, our work demonstrates how to **achieve dimensional reduction in a precise manner**. I.e., how biological networks, notoriously large and messy, can reduce their activity to just a submanifold of their full phase space. We also provide new experimental predictions of a topological state: the first such proposal for observing topological invariants in a biological system. On the theoretical level, we characterize in this manuscript the coherence of the cycle produced, showing that it satisfies theoretical bounds for the most coherent oscillator equivalent to that of a unicycle network – without being a single-use network the way a unicycle is. In short, our work is able to explain long-standing puzzles in biology such as how dimensional reduction is achieved in a robust and flexible manner, and the **necessary role of dissipative motifs such as futile cycles and phosphorylation to produce long and coherent oscillations**.

In reply to a few comments & changes:

> *We have added a clarification in the third paragraph in the introduction to mention the close connection of our model to the PRX one: “We expand on the topological model first proposed in [1] and apply it to the KaiABC system, which regulates the circadian rhythm of cyanobacteria [2].”*

This still hides things. Ref [1] also applies this to the KaiABC system – section IV.A, page 9. What’s novel is comparison to other models, I think.

We regret that this remains a point of confusion. In the revised manuscript, we re-write the introduction to include the points above and delineate the radical novelty of our work, not just for KaiC but for the field more broadly. The revised introduction is given below.

- Towards dimensional reduction onto the network boundary, topological models have been proposed in stochastic systems [1–3]. These are a generalization of topological invariants studied in quantum systems [4–7], which show physical responses on the system edge or boundary. Powerfully, this response is insensitive to various types of disorder or noise [8]. It would be desirable to demonstrate how topology can be realized in a biological system, given the many attractive properties of topology such as its robust response. However, it has not yet been shown to relate to a biological system. Previous works do not explicitly connect to experimentally tunable parameters or known molecular reactions, giving little guidance on how to measure or access the topological properties [1–3]. Our manuscript responds to this lack in the field, by providing detailed biophysical mechanisms towards experimental verification of topological invariants.

Here, we provide the first case study of a topological mechanism in a concrete biological system to render these models operational in the context of experimental knobs – that of the KaiABC system for the circadian rhythm of cyanobacteria [9, 10]. The localized edge currents naturally reproduce the kinetic ordering of KaiC phosphorylation cycles. By demonstrating its parameter space and how to tune into and out of the topological transition, we propose pathways to alter the transition rates (e.g. by changing ATP concentration [11] or using a mutant [12]) which will affect the period and robustness of the oscillation. We explain the delicate interaction of phosphorylation with conformational changes, which is a necessary and sufficient condition for the observed oscillatory cycle, in a simple yet biophysically plausible way [13–16]. This yields key insights towards characterization of an important regulatory system that had until now required quite complicated models especially to obtain the observed kinetic ordering of T and S phosphorylation [17–19].

Our connection of topology to real and measurable quantities in a biology system reveals many theoretical and conceptual consequences that go far beyond the specific system we study. Where the measurable outcomes in quantum and other systems are that of electrical conductivity [20–22] or floppy modes in a spring network [23, 24], our work demonstrates how to achieve dimensional reduction in a precise manner. I.e., how biological networks, notoriously large and messy, can reduce their activity to just a submanifold of their full phase space. On the theoretical level, we characterize in this manuscript the coherence of the cycle produced, showing that it satisfies theoretical bounds [25] for the most coherent oscillator equivalent to that of a unicycle network – without being a single-use network the way a unicycle is. In short, our work is able to explain long-standing puzzles in biology such as how dimensional reduction is achieved in a robust and flexible manner, and the necessary role of dissipative motifs such as futile cycles [26, 27] and phosphorylation, to produce long and coherent oscillations.

> In both a and b, the curly line from the internal label attaches to the monomer most likely to be modified next, indicating that the monomer is tagged by molecular modifications (e.g. KaiA or KaiB binding) and primed for subsequent change.

Here I think you misunderstand my complaint about graphical choices. It's not that they must all be justified somewhere in the text. It's that the poor reader needs to know as soon as possible to ignore them, since they are in no way part of the model. I object to the use of extraneous detail to hide how the model actually works, and render its emergent behaviours more mysterious than they really are.

We appreciate the reviewer's comments on how to best describe the figures and have worked hard to implement their many suggestions thus far, in good faith. In this case, we believe that examining the biophysical mechanisms and how they work in our model, is a worthwhile endeavor. Hence, we would like to keep these specific details in our manuscript.

Reading everything again, my perhaps my core complaint is this one:

>> The details obscure the core of how it works: The core N-E-S-W-N cycle which drives "external" transition is *designed* to produce edge states, and it does.

It's very clear to me now why this cycle of internal states is desirable for producing edge states. But is it possible to engineer such a thing out of biologically plausible components? That is, a cycle of states which control the direction of allowed "external" transitions.

This is something the PRX paper does not address, and if the intent is to publish something with more biological detail, then perhaps this paper ought to contain such a mechanism. The only mention of this I can find is this (on page 3, right column) which seems more like speculation on the ingredients nature would have to use, rather than a description of how it's possible to assemble them to produce this quite specific circuit:

> For the KaiABC system, the internal transitions could correspond to interactions with KaiA and KaiB proteins [51] and conformational changes of a loop-like structure on KaiC called the A-loop (two conformations represented by blue squares and purple circles in Fig. 1 respectively) [52]. These transitions modify the internal state of the system, changing its tendency to phosphorylate or dephosphorylate and whether in T or S.

A key goal of our paper is indeed to clarify the explicit biological mechanism, in order to allow for engineering of the cycle out of biologically plausible components. As this was unclear before, we now highlight the biological mechanisms more directly and early on, also summarizing them in a table for succinctness and immediate comprehension. The following changes respond to the reviewer's desire for a biological mechanism in a clearer and more transparent exposition:

- The speculation of ingredients on page 3 that the reviewer mentions was fleshed out in detail in later pages, on the left column of page 5. The biological mechanisms are now described more directly and early on, in the second paragraph of Sec. II where the internal cycles are first introduced (reproduced below).
 - The "internal" variable s , given by compass directions N-E-S-W, labels the internal state of the system that primes it to go through one of the four possible external transitions next. For example, in the W state, the system is most likely to go through the westward external transition that decrements x . Within each phosphorylation level (x, y) , the system can take any of the four possible internal states N, E, S or W that prime each possible external transition. As shown in Fig. 1b, these internal states are also assumed to transition between each other in a cyclic manner, with internal transition rates γ_{in} and slower reverse rates $\gamma'_{in} \ll \gamma_{in}$ (gray dashed arrows). For the KaiABC system,

Internal transition	Biophysical mechanism
$S \xrightarrow{\gamma_{in}} E$	KaiA binding, which primes T phosphorylation [29–31]
$E \xrightarrow{\gamma_{in}} N$	A-loop burial, which is linked to S phosphorylation [14]
$N \xrightarrow{\gamma_{in}} W$	KaiB binding, which primes T dephosphorylation [31, 32]
$W \xrightarrow{\gamma_{in}} S$	A-loop exposure, which is coupled to S dephosphorylation [16]

TABLE I. **Relevant biophysical mechanisms for the internal transitions and how they prime subsequent phosphorylation/dephosphorylation reactions, as seen in experimental data.**

such cyclic transitions result from the interaction between ligand binding (by KaiA and KaiB proteins) and different conformational states of the A-loop. The A-loop can switch between exposed and buried conformations (represented by blue squares and purple circles respectively in Fig. 1). The A-loop conformational change from buried to exposed facilitates KaiA binding [13] and inhibits KaiB binding [14]. Conversely, the A-loop conformational change from exposed to buried facilitates KaiB binding and inhibits KaiA binding [14]. These reactions correspond to the cycle of internal transitions, as listed in Table I. Crucially, each reaction primes a specific phosphorylation or dephosphorylation in T or S, as seen experimentally. We represent the internal transition associated with KaiA with a curly arrow, since KaiA promotes phosphorylation through rapid association and dissociation, activating a larger stoichiometric amount of KaiC as compared to the typical one-on-one binding like KaiB [28].

- We include a table for succinctness and rapid comprehension; see Table I in this document.
- We move specifications of the model in the second paragraph in Sec. II to the third paragraph, reproduced below.
 - These reactions are repeated for each monomer and hence can be laid out as a lattice. As shown in Fig. 1c, the phosphorylation/dephosphorylation cycles in Fig. 1a (green box) and the internal cycles in Fig. 1b (blue box) both repeat along the x and y axes of T and S phosphorylation. Such a lattice will have edges representing the physical constraints of the system, i.e., $0 \leq x \leq N_x$ and $0 \leq y \leq N_y$. In our case, $N_x = N_y = 6$ since there are 6 sites available on a KaiC hexamer for either T or S phosphorylation. In particular, we assume that the monomer conformational changes occur after S phosphorylation of the same monomer. It follows that the states $(x, y)_E$ and $(x, y)_S$ (for $y > 0$) have $y - 1$ circles and $7 - y$ squares, while the states $(x, y)_W$ and $(x, y)_N$ have y circles and $6 - y$ squares.
- The remaining content from page 5 is merged with the last paragraph in Sec. II, reproduced below.
 - Our topological model presents an alternative way to account for the experimental facts and explain the emergence of the KaiC phosphorylation cycle. As discussed above, in the topological regime where $\rho \gg 0$, the system supports a propagating edge current. Indeed, KaiB binding and possibly its unbinding (with rates γ_{in}) are observed to be slow processes [28, 31, 33, 34] compared to phosphorylation reactions (with rates γ_{ex}) [18]. This is consistent with our premise that there is a separation of timescales. Fig. 2b shows a coarse-grained picture of the edge current, where one of the four internal states is shown for each phosphorylation level (x, y) along the edge, specifically the last state

along the edge (e.g. E states for the bottom edge or S states for the left edge). As we can see, the edge current is equivalent to a global cycle of concerted phosphorylation of the T-sites, followed by the S-sites, then dephosphorylation of the T-sites and, lastly, of the S-sites (also see Supplementary Movie). This provides a mechanism that allows for individual monomers to undergo conformational and other changes, while still producing a global cycle and the experimentally observed phosphorylation sequence that emerges with less fine-tuning.

-
- [1] Arvind Murugan and Suriyanarayanan Vaikuntanathan, “Topologically protected modes in non-equilibrium stochastic systems,” *Nature communications* **8**, 13881 (2017).
 - [2] Kinjal Dasbiswas, Kranthi K Mandadapu, and Suriyanarayanan Vaikuntanathan, “Topological localization in out-of-equilibrium dissipative systems,” *Proceedings of the National Academy of Sciences* **115**, E9031–E9040 (2018).
 - [3] Evelyn Tang, Jaime Agudo-Canalejo, and Ramin Golestanian, “Topology protects chiral edge currents in stochastic systems,” *Physical Review X* **11**, 031015 (2021).
 - [4] Joel E Moore, “The birth of topological insulators,” *Nature* **464**, 194–198 (2010).
 - [5] Ching-Kai Chiu, Jeffrey CY Teo, Andreas P Schnyder, and Shinsei Ryu, “Classification of topological quantum matter with symmetries,” *Reviews of Modern Physics* **88**, 035005 (2016).
 - [6] Horst L Stormer, Daniel C Tsui, and Arthur C Gossard, “The fractional quantum hall effect,” *Reviews of Modern Physics* **71**, S298 (1999).
 - [7] Evelyn Tang and Xiao-Gang Wen, “Interacting one-dimensional fermionic symmetry-protected topological phases,” *Physical Review Letters* **109**, 096403 (2012).
 - [8] J Weis and K Von Klitzing, “Metrology and microscopic picture of the integer quantum hall effect,” *Philosophical Transactions of the Royal Society A: Mathematical, Physical and Engineering Sciences* **369**, 3954–3974 (2011).
 - [9] Masahiro Ishiura, Shinsuke Kutsuna, Setsuyuki Aoki, Hideo Iwasaki, Carol R Andersson, Akio Tanabe, Susan S Golden, Carl H Johnson, and Takao Kondo, “Expression of a gene cluster kaiABC as a circadian feedback process in cyanobacteria,” *Science* **281**, 1519–1523 (1998).
 - [10] Masato Nakajima, Keiko Imai, Hiroshi Ito, Taeko Nishiwaki, Yoriko Murayama, Hideo Iwasaki, Tokitaka Oyama, and Takao Kondo, “Reconstitution of circadian oscillation of cyanobacterial KaiC phosphorylation in vitro,” *science* **308**, 414–415 (2005).
 - [11] Connie Phong, Joseph S Markson, Crystal M Wilhoite, and Michael J Rust, “Robust and tunable circadian rhythms from differentially sensitive catalytic domains,” *Proceedings of the National Academy of Sciences* **110**, 1124–1129 (2013).
 - [12] Ximing Qin, Mark Byrne, Tetsuya Mori, Ping Zou, Dewight R Williams, Hassane Mchaourab, and Carl Hirschie Johnson, “Intermolecular associations determine the dynamics of the circadian KaiABC oscillator,” *Proceedings of the National Academy of Sciences* **107**, 14805–14810 (2010).
 - [13] Yong-Ick Kim, Guogang Dong, Carl W Carruthers Jr, Susan S Golden, and Andy LiWang, “The day/night switch in KaiC, a central oscillator component of the circadian clock of cyanobacteria,” *Proceedings of the National Academy of Sciences* **105**, 12825–12830 (2008).
 - [14] Yong-Gang Chang, Nai-Wei Kuo, Roger Tseng, and Andy LiWang, “Flexibility of the C-terminal, or CII, ring of KaiC governs the rhythm of the circadian clock of cyanobacteria,” *Proceedings of the National Academy of Sciences* **108**, 14431–14436 (2011).
 - [15] Roger Tseng, Yong-Gang Chang, Ian Bravo, Robert Latham, Abdullah Chaudhary, Nai-Wei Kuo, and Andy LiWang, “Cooperative KaiA–KaiB–KaiC interactions affect KaiB/SasA competition in the circadian clock of cyanobacteria,” *Journal of molecular biology* **426**, 389–402 (2014).
 - [16] Lu Hong, Bodhi P Vani, Erik H Thiede, Michael J Rust, and Aaron R Dinner, “Molecular dynam-

- ics simulations of nucleotide release from the circadian clock protein *kaic* reveal atomic-resolution functional insights,” *Proceedings of the National Academy of Sciences* **115**, E11475–E11484 (2018).
- [17] Jeroen S van Zon, David K Lubensky, Pim RH Altena, and Pieter Rein ten Wolde, “An allosteric model of circadian KaiC phosphorylation,” *Proceedings of the National Academy of Sciences* **104**, 7420–7425 (2007).
- [18] Joris Paijmans, David K Lubensky, and Pieter Rein Ten Wolde, “A thermodynamically consistent model of the post-translational Kai circadian clock,” *PLoS computational biology* **13**, e1005415 (2017).
- [19] Paul Smolen, Douglas A Baxter, and John H Byrne, “Modeling circadian oscillations with interlocking positive and negative feedback loops,” *Journal of Neuroscience* **21**, 6644–6656 (2001).
- [20] K v Klitzing, Gerhard Dorda, and Michael Pepper, “New method for high-accuracy determination of the fine-structure constant based on quantized hall resistance,” *Physical review letters* **45**, 494 (1980).
- [21] F Duncan M Haldane, “Model for a quantum hall effect without landau levels: Condensed-matter realization of the” parity anomaly”,” *Physical review letters* **61**, 2015 (1988).
- [22] M Zahid Hasan and Charles L Kane, “Colloquium: topological insulators,” *Reviews of modern physics* **82**, 3045 (2010).
- [23] CL Kane and TC Lubensky, “Topological boundary modes in isostatic lattices,” *Nature Physics* **10**, 39–45 (2014).
- [24] Roman Süsstrunk and Sebastian D Huber, “Observation of phononic helical edge states in a mechanical topological insulator,” *Science* **349**, 47–50 (2015).
- [25] Andre C Barato and Udo Seifert, “Coherence of biochemical oscillations is bounded by driving force and network topology,” *Physical Review E* **95**, 062409 (2017).
- [26] John J Hopfield, “Kinetic proofreading: a new mechanism for reducing errors in biosynthetic processes requiring high specificity,” *Proceedings of the National Academy of Sciences* **71**, 4135–4139 (1974).
- [27] Michael Samoilov, Sergey Plyasunov, and Adam P Arkin, “Stochastic amplification and signaling in enzymatic futile cycles through noise-induced bistability with oscillations,” *Proceedings of the National Academy of Sciences* **102**, 2310–2315 (2005).
- [28] Hakuto Kageyama, Taeko Nishiwaki, Masato Nakajima, Hideo Iwasaki, Tokitaka Oyama, and Takao Kondo, “Cyanobacterial circadian pacemaker: Kai protein complex dynamics in the KaiC phosphorylation cycle in vitro,” *Molecular cell* **23**, 161–171 (2006).
- [29] Yao Xu, Tetsuya Mori, and Carl Hirschie Johnson, “Cyanobacterial circadian clockwork: roles of KaiA, KaiB and the kaiBC promoter in regulating KaiC,” *The EMBO Journal* **22**, 2117–2126 (2003).
- [30] Jenny Lin, Justin Chew, Udaysankar Chockanathan, and Michael J Rust, “Mixtures of opposing phosphorylations within hexamers precisely time feedback in the cyanobacterial circadian clock,” *Proceedings of the National Academy of Sciences* **111**, E3937–E3945 (2014).
- [31] Michael J Rust, Joseph S Markson, William S Lane, Daniel S Fisher, and Erin K O’Shea, “Ordered phosphorylation governs oscillation of a three-protein circadian clock,” *Science* **318**, 809–812 (2007).
- [32] Yohko Kitayama, Hideo Iwasaki, Taeko Nishiwaki, and Takao Kondo, “KaiB functions as an attenuator of KaiC phosphorylation in the cyanobacterial circadian clock system,” *The EMBO journal* **22**, 2127–2134 (2003).
- [33] Jun Abe, Takuya B Hiyama, Atsushi Mukaiyama, Seyoung Son, Toshifumi Mori, Shinji Saito, Masato Osako, Julie Wolanin, Eiki Yamashita, Takao Kondo, *et al.*, “Atomic-scale origins of slowness in the cyanobacterial circadian clock,” *Science* **349**, 312–316 (2015).
- [34] Damien Simon, Atsushi Mukaiyama, Yoshihiko Furuike, and Shuji Akiyama, “Slow and temperature-compensated autonomous disassembly of *kaib*–*kaic* complex,” *Biophysics and Physicobiology* **19**, e190008 (2022).

REVIEWER COMMENTS

Reviewer #1 (Remarks to the Author):

Whether to recommend publication after this many rounds of review hinges to me on whether this paper is a sufficiently novel instance of the model proposed earlier in PRX (which I see as being attractive). I am not an expert in circadian rhythms.

The current paper is a fleshed-out version of section IVA of the earlier paper. There is a lot added in terms of the model, but the conclusions (particularly robustness) are theoretical and not easily testable experimentally (there is essentially one paragraph on page 8).

This is a useful paper, and a nice piece of work. But I'm concluding that it belongs in a more specialized journal.

Reviewer #2 (Remarks to the Author):

Fourth review of "A topological mechanism for robust and efficient global oscillations in biological networks" Chongbin Zheng & Evelyn Tang, by reviewer 2.

I have never before written a 4th review, and I confess that I am tired of being shouted at (in bold) and condescended to ("We regret that this remains a point of confusion"). Where I come from, phrases like "thoughtful analysis" are for others to say, not things you say about yourself.

I do still think that establishing the existence of edge states in a biological system would be exciting.

I do not find the writing of the paper easy to read, the introduction especially seems like a muddle. The abstract mis-uses "parsimonious".

The paper still contains 3 things:

1. a mathematical model which has edge states

2. an analysis of their desirable properties, esp. compared to MWC models
3. a proposed mapping of this model to actual biochemical components.

I have felt since the beginning that the description of the mathematical model, 1, ought to be crystal clear, and not muddled in with the details of 3. The model was clearer (and simpler) in the PRX paper, which I had to read to decode it initially. The presentation here has slowly improved, but it still obscured by many details. Deliberately so, according to the replies.

About point 3, the new draft has Table 1. I spent some time trying to read these references. I remain dubious that listing these amounts to “providing detailed biophysical mechanisms which are the only path towards any experimental comparison”, and believe that “the claim of topology to relate to a biological system remains inspirational at best”.

The mechanism of Fig 1 requires not just a vague “coupling” of components. It requires an internal cycle which is driven in one direction, and controls which external transitions are possible. And it requires that the external transition drive an opposite change in the internal cycle. This is a very specific thing. The four edges of the (x,y) space have four different pairs of fast/slow transitions in sequence.

If this paper claims to have established that such a thing plausibly exists in biology, that would be exciting. But if a general reader is expected to believe this, it needs to be laid out much more explicitly, at much greater length. What exactly are we to take from each of these references? Which transition have been shown to be non-equilibrium, which are hypothesised?

Maybe these questions are obvious to a reader who already knows [42, 44, 57-59] and hence need little explanation here. If so, then the target audience does not include me. And the editor is going to need to find a reviewer within this audience. Or else the authors are going to need to find a more specialized journal.

Reviewer #2 (Remarks on code availability):

Code and data are promised on publication, not now.

Reviewer #4 (Remarks to the Author):

The authors developed an elegant model of biochemical oscillations, showing that the topological mechanism plays an important role in sustaining oscillations. However, the mathematical elegance of this model was already explained in the authors' previous publication (Phys. Rev. X 2021, Ref.31). What is new in the present manuscript is mainly on the two points:

- (1) The authors tried to improve the model to make it testable with experiments on the KaiABC clock.
- (2) The authors analyzed the relation between entropy production and the coherence of oscillations.

In (2), the authors used the band theory argument and showed the nontrivial relation arising from the topological mechanism. I think this is an unexpected novel finding in oscillation models.

However, point (1) is rather embarrassing compared to the following points (A) and (B).

(A) The model is based on the limit of weak coupling between individual subunits in the KaiC hexamer, as the authors wrote that "the positive cooperativity between monomer conformation states is fairly weak [56]" and "individual monomers can independently phosphorylate and shuffle between hexamers." I disagree with this view for the following reasons:

(i) Ref. 56 (Han et al., Nat. Comm. 2023) wrote the opposite. In Ref.56, the cryo-EM data of KaiC hexamers were fitted by the Ising ferromagnet model, where the structural state of each subunit was represented by a binary (i.e., Ising) variable. This variable is cooperatively coupled with the structural state of other subunits to stabilize the two allosteric states of the hexamer, as in the MWC picture. It is natural to consider the fluctuations around the two stable states as discussed in Ref.56, but this does not support the view of the weak coupling limit claimed by the authors.

(ii) The observed KaiA-KaiC stoichiometry shows that a single KaiA dimer binds on the C-terminal side of a KaiC hexamer (Pattanayek et al. EMBO J. 2006; Mori et al. Nat. Comm. 2018). There is no experimental support for the binding of multiple KaiA dimers on a single KaiC hexamer, implying the six subunits change cooperatively, and a single KaiA dimer binds to a bundled A-loops of six subunits. However, the model in the present manuscript assumes each A-loop of six subunits changes independently in the weak coupling limit, allowing independent multiple binding of KaiA dimers on a single KaiC hexamer, which contradicts the observation.

(iii) Monomer shuffling was observed in a specific phase of oscillations (Kageyama et al. Mol. Cell 2006), showing that the variation of the entire hexamer along the oscillation period should change the hexamer stability and the shuffling frequency; therefore, shuffling is not the evidence for the weak coupling limit.

(B) Synchronization/desynchronization among the large number of KaiC hexamers in the solution is essential for predicting experiments, which is the main point of the other models cited by the authors. The authors briefly touched on this point in the paper's last part as the future problem, but I am afraid this aspect should mask the model from the experimental test.

In summary, the proposed model explains many new theoretical insights, such as the novel topological mechanism governing the relationship between entropy production and oscillation coherence and the helpful analogy of the band theory to explain the oscillation coherence, and it is indeed fun if they would be observed in real systems. The proposed test of the model, however, may not reflect the complexity of the real system; each KaiC subunit fluctuates but not independently as in the weak-coupling limit, though they are not in the 100% all/none cooperativity; therefore, a more careful discussion is needed to explain the model validation. Mutations or the ATP concentration change should affect the synchronization of many oscillators; therefore, a more careful proposal of the experimental test should be needed.

Response to the Referees for “A topological mechanism for robust and efficient global oscillations in biological networks”

Chongbin Zheng^{1,2}, Evelyn Tang^{1,2,*}

1 Center for Theoretical Biological Physics
Rice University, Houston, Texas 77005, USA

2 Center for Theoretical Biological Physics
Rice University, Houston, Texas 77005, USA

* E-mail: Corresponding e.tang@rice.edu

I. REVIEWER 1

Whether to recommend publication after this many rounds of review hinges to me on whether this paper is a sufficiently novel instance of the model proposed earlier in PRX (which I see as being attractive). I am not an expert in circadian rhythms.

We thank their reviewer for their comments. Besides the novel biological mechanisms proposed for this model, we have made several new theoretical and computational physics contributions, including:

- A detailed analysis of the thermodynamic properties and new regimes of this model
- A new computational signature for the robustness of the system, based on spectral gaps, that has not previously been suggested for biochemical systems
- Clear and concrete physical mechanisms that support the topological model, which have not previously been proposed or analyzed
- A careful comparison of how coherence compares with other available models in the literature, and a comparison of how each model performs compared to theoretically-predicted bounds from stochastic thermodynamics
- New discussions on how coupling between molecules can lead to higher dimensional manifolds and edge states

The current paper is a fleshed-out version of section IVA of the earlier paper. There is a lot added in terms of the model, but the conclusions (particularly robustness) are theoretical and not easily testable experimentally (there is essentially one paragraph on page 8).

We appreciate the reviewer's viewpoint, and in the revised manuscript provide specific experimental predictions and their implementation modality. One prediction probes a feature uniquely stemming from topology, see Sec. V. Another prediction would distinguish our topological model from the other models in the literature based on the Monod-Wyman-Changeux paradigm – for which there is currently no alternative paradigm [1–3].

This is a useful paper, and a nice piece of work. But I'm concluding that it belongs in a more specialized journal.

To provide more background and accessibility, we note that topology is one of the rare fields in which theory leads experiment. This has happened in many contexts in quantum systems [4–7], including in our own work, where purely theoretical predictions made [8, 9] were subsequently verified years later in several landmark experimental publications [7, 10]. Note that when the theoretical work was published, the experimental systems were not yet developed to test them, as it required years for the correct samples and conditions to be developed.

While these previous successes have been in quantum and other platforms, topology in stochastic and biological systems is nascent. However, the powerful formalism of topology and previous examples show us that topological predictions on the theoretical and computational level have a lot to offer, and can have a huge payoff in terms of experimental consequences. It would be extremely exciting if topology can also be verified in biological systems, as it has been in so many other platforms, and our work provides the first concrete blueprint to do so. We have reworked the Introduction to include this context and background.

II. REVIEWER 2

Fourth review of “A topological mechanism for robust and efficient global oscillations in biological networks” Chongbin Zheng & Evelyn Tang, by reviewer 2.

I have never before written a 4th review, and I confess that I am tired of being shouted at (in bold) and condescended to (“We regret that this remains a point of confusion”). Where I come from, phrases like “thoughtful analysis” are for others to say, not things you say about yourself.

We apologize for these misunderstandings. It was certainly not our intention to shout: the bold writing was meant to highlight key terms and we are sorry if this came across as rudeness. We also did not mean to be condescending – saying “we regret” was a reflection on the limitations in our own writing and nothing else. Please excuse our poor word and stylistic choices and for any ill feelings they may have caused.

I do still think that establishing the existence of edge states in a biological system would be exciting.

I do not find the writing of the paper easy to read, the introduction especially seems like a muddle. The abstract mis-uses “parsimonious”.

Thank you for these comments. We have reworked the introduction to make it easier to read and more accessible. We did indeed misuse “parsimonious” in the abstract and have rewritten those sentences.

The paper still contains 3 things: 1. a mathematical model which has edge states 2. an analysis of their desirable properties, esp. compared to MWC models 3. a proposed mapping of this model to actual biochemical components.

I have felt since the beginning that the description of the mathematical model, 1, ought to be crystal clear, and not muddied in with the details of 3. The model was clearer (and simpler) in the PRX paper, which I had to read to decode it initially. The presentation here has slowly improved, but it still obscured by many details. Deliberately so, according to the replies.

We thank the reviewer for taking the time to read these other references and for their patience. We were not deliberately trying to obscure the presentation, but have been unsure on what to emphasize for biology and physics audiences simultaneously – emphasizing some aspects has at times obscured others. We appreciate the reviewer’s point that the main aspects of the model ought to be clear before diving into more specialized details, and in the revised manuscript have explicitly separated the fundamental mechanisms from detailed biophysics discussions. To this end, we have created a new section on specific mechanisms and their experimental literature: section III – that comes after the section on the mathematical model.

About point 3, the new draft has Table 1. I spent some time trying to read these references. I remain dubious that listing these amounts to “providing detailed biophysical mechanisms which are the only path towards any experimental comparison”, and believe that “the claim of topology to relate to a biological system remains inspirational at best”.

We agree that the table inadequately conveys the mechanisms, and have now created new figures to illustrate key biophysical mechanisms, especially that of the fast and slow steps in the model (see Fig. 1 on the next page), to replace Table I. These aspects are discussed more carefully in the new Section III. Lastly, we have worked out more explicit experimental predictions: one uniquely stemming from topology and another that distinguishes our model for those using the prevailing MWC paradigm.

FIG. 1. (Fig. 2 in the manuscript.) **Biophysical mechanisms for the topological model, which reproduces the KaiABC circadian rhythm.** **a**, Phosphorylation of a monomer relies on two main steps. *Left*: Slow reaction γ_{in} where KaiA promotes ADP release and ATP binding in the CII nucleotide binding pocket (gray oval) [11–13], priming the monomer for phosphorylation. *Right*: Fast reaction γ_{ex} where a phosphate group P_i is transferred from ATP to the phosphorylation site [11, 12]. **b**, Dephosphorylation similarly has two main steps. *Left*: Slow reaction γ_{in} where KaiB binds to KaiC [14, 15] and ATP hydrolyzes to ADP at CII [11, 12], priming the monomer for dephosphorylation. *Right*: Fast reaction γ_{ex} where P_i is transferred back to ADP [11, 12]. In both **a** and **b**, we illustrate what happens for T-sites; additional distinctions between T and S phosphorylation/dephosphorylation are discussed in the main text. **c**, KaiABC exhibits oscillations via a concerted global cycle of phosphorylation and dephosphorylation. During the day, all six KaiC monomers get phosphorylated at the T-sites, and then at the S-sites. Phosphorylation is promoted by interaction with KaiA molecules [16]. By night, phosphorylated KaiC binds to KaiB, which sequesters KaiA from the solution. In the absence of KaiA, all the T-sites get dephosphorylated, followed by the S-sites [14]. Since individual monomers can independently phosphorylate [17], it is unclear why they would perform a concerted phosphorylation cycle that is robust. **d**, A possible solution lies in the topological phase of the model, in which a global cycle emerges that recapitulates the experimentally observed phosphorylation sequence.

The mechanism of Fig 1 requires not just a vague “coupling” of components. It requires an internal cycle which is driven in one direction, and controls which external transitions are possible. And it requires that the external transition drive an opposite change in the internal cycle. This is a very specific thing. The four edges of the (x,y) space have four different pairs of fast/slow transitions in sequence.

If this paper claims to have established that such a thing plausibly exists in biology, that would be exciting. But if a general reader is expected to believe this, it needs to be laid out much more explicitly, at much greater length. What exactly are we to take from each of these references? Which transition have been shown to be non-equilibrium, which are hypothesised?

We thank the reviewer for their thoughtful questions. In the revision, we have created new figures to explicitly lay out the slow reactions that prime the fast reactions: there are two main mechanisms for phosphorylation and dephosphorylation respectively, in addition to further distinctions between S and T. We also provide experimental evidence for non-equilibrium driving that powers the phosphorylation cycle, and for the particular directionality of the transitions (new Section III).

Maybe these questions are obvious to a reader who already knows [42, 44, 57-59] and hence need little explanation here. If so, then the target audience does not include me. And the editor is going to need to find a reviewer within this audience. Or else the authors are going to need to find a more specialized journal.

We appreciate the reviewer helping us to clarify our mapping to biochemical components. These are in addition to the theoretical and computational physics contributions, including

- A detailed analysis of the thermodynamic properties and new regimes of this model
- A new computational signature for the robustness of the system, based on spectral gaps, that has not previously been suggested for such biochemical systems
- A careful comparison of how coherence compares with other available models in the literature, and a comparison of how each model performs compared to theoretically-predicted bounds from stochastic thermodynamics
- New discussions on how coupling between molecules can lead to higher dimensional manifolds and edge states

To provide more background and accessibility, we note that topology is one of the rare fields in which theory leads experiment. This has happened in many contexts in quantum systems [4–7], including in our own work, where purely theoretical predictions made [8, 9] were subsequently verified years later in several landmark experimental publications [7, 10]. Note that when the theoretical work was published, the experimental systems were not yet developed to test them, as it required years for the correct samples and conditions to be developed.

While these previous successes have been in quantum and other platforms, topology in stochastic and biological systems is nascent. However, the powerful formalism of topology and previous examples show us that topological predictions on the theoretical and computational level have a lot to offer, and can have a huge payoff in terms of experimental consequences. It would be extremely exciting if topology can also be verified in biological systems, as it has been in so many other platforms, and our work provides the first concrete blueprint to do so. We have reworked the Introduction to include this context and background.

Reviewer #2 (Remarks on code availability):

Code and data are promised on publication, not now.

III. REVIEWER 4

The authors developed an elegant model of biochemical oscillations, showing that the topological mechanism plays an important role in sustaining oscillations. However, the mathematical elegance of this model was already explained in the authors' previous publication (Phys. Rev. X 2021, Ref.31). What is new in the present manuscript is mainly on the two points:

(1) The authors tried to improve the model to make it testable with experiments on the KaiABC clock. (2) The authors analyzed the relation between entropy production and the coherence of oscillations.

In (2), the authors used the band theory argument and showed the nontrivial relation arising from the topological mechanism. I think this is an unexpected novel finding in oscillation models.

We thank the reviewer for their kind comments on our contributions in point (2).

However, point (1) is rather embarrassing compared to the following points (A) and (B).

(A) The model is based on the limit of weak coupling between individual subunits in the KaiC hexamer, as the authors wrote that "the positive cooperativity between monomer conformation states is fairly weak [56]" and "individual monomers can independently phosphorylate and shuffle between hexamers." I disagree with this view for the following reasons: (i) Ref. 56 (Han et al., Nat. Comm. 2023) wrote the opposite. In Ref.56, the cryo-EM data of KaiC hexamers were fitted by the Ising ferromagnet model, where the structural state of each subunit was represented by a binary (i.e., Ising) variable. This variable is cooperatively coupled with the structural state of other subunits to stabilize the two allosteric states of the hexamer, as in the MWC picture. It is natural to consider the fluctuations around the two stable states as discussed in Ref.56, but this does not support the view of the weak coupling limit claimed by the authors.

We appreciate the referee drawing attention to the specific claims of Han et al., and believe that closer examination of their work actually supports the weak coupling limit. This can be seen from their reported data, where the two states consistent with the MWC picture (all buried or all exposed A-loops) are not the most abundant species observed (see Figure 3a, 3b and 4a in their work). Rather, hexamers with mixed A-loop conformations make up more than 80% of the entire population. Notably, the coupling constant J from the Ising model fit is very small ($J = 0.086$) and the correlation length ξ is short ($\xi_{AA} \approx \xi_{EE} \approx 0.5$ monomer length) – demonstrating weak cooperativity between monomers. Most of all, this conclusion is supported by the authors themselves, who write the following at the bottom of page 5 in their paper: "Given the individual conformational state of each monomer in hexamers, the all-or-none Monod–Wyman–Changeux (MWC) model certainly does not fit the experimental data" [18]. We have included these considerations in the Discussion in the revised manuscript.

(ii) The observed KaiA-KaiC stoichiometry shows that a single KaiA dimer binds on the C-terminal side of a KaiC hexamer (Pattanayek et al. EMBO J. 2006; Mori et al. Nat. Comm. 2018). There is no experimental support for the binding of multiple KaiA dimers on a single KaiC hexamer, implying the six subunits change cooperatively, and a single KaiA dimer binds to a bundled A-loops of six subunits. However, the model in the present manuscript assumes each A-loop of six subunits changes independently in the weak coupling limit, allowing independent multiple binding of KaiA dimers on a single KaiC hexamer, which contradicts the observation.

The reviewer is correct and while our proposed model was intended to be consistent with these data, we apologize that it was explained poorly before. We certainly agree that only a single KaiA dimer binds to the KaiC hexamer. However, a single KaiA dimer binds and unbinds several times during each phosphorylation [19, 20], so that it acts as a catalyst of the reaction, consistent with

the treatment of KaiA in previous models in the literature [1, 2]. To make this clearer in the revised manuscript we have replaced “KaiA binding” with “KaiA catalysis” throughout the text. We also clarify that this catalytic process involves rapid binding and unbinding of KaiA in Sec. II, and that the intermediate KaiA-bound states are not shown (i.e. coarse-grained out of the model); also see the new Fig. 2a (top of page 4 in this document). Lastly, we use a curved arrow to emphasize the catalytic action of KaiA in Fig. 1b, and draw attention to this notation in the caption and in Sec. II.

(iii) Monomer shuffling was observed in a specific phase of oscillations (Kageyama et al. Mol. Cell 2006), showing that the variation of the entire hexamer along the oscillation period should change the hexamer stability and the shuffling frequency; therefore, shuffling is not the evidence for the weak coupling limit.

The reviewer is correct and we have removed the reference to monomer shuffling.

(B) Synchronization/desynchronization among the large number of KaiC hexamers in the solution is essential for predicting experiments, which is the main point of the other models cited by the authors. The authors briefly touched on this point in the paper’s last part as the future problem, but I am afraid this aspect should mask the model from the experimental test.

In summary, the proposed model explains many new theoretical insights, such as the novel topological mechanism governing the relationship between entropy production and oscillation coherence and the helpful analogy of the band theory to explain the oscillation coherence, and it is indeed fun if they would be observed in real systems. The proposed test of the model, however, may not reflect the complexity of the real system; each KaiC subunit fluctuates but not independently as in the weak-coupling limit, though they are not in the 100% all/none cooperativity; therefore, a more careful discussion is needed to explain the model validation. Mutations or the ATP concentration change should affect the synchronization of many oscillators; therefore, a more careful proposal of the experimental test should be needed.

We agree with the reviewer that tests of coherent oscillations are difficult to separate from synchronization effects between many hexamers. In the revised version, we focus on experimental tests that would be insensitive to coherence and synchronization effects. There are three main approaches:

1. First, we predict a response unique to our topological model where decreasing ATP concentration leads to an increase in ADP consumption. This is the opposite of what would typically be expected and is due to the system being driven from the edge localized state into bulk dynamics, that simultaneously increases energy consumption despite a decrease in coherence. This is unique to our topological model and discussed in Fig. 3c. We expect a 5-fold increase in ADP consumption for a 10-fold decrease in ATP concentration, which can be measured via tracking ADP levels [21] or alternatively with heat dissipation [22].
2. Second, we predict that KaiC hexamers should have a wide distribution of different A-loop conformational patterns when imaged over the period of an oscillation, similar to what has been seen in cryo-EM for non-oscillating KaiC hexamers [18]. This would distinguish our model from MWC-type models [1–3], which predict that all six A-loops would have very similar conformations (i.e., mostly buried or mostly exposed).
3. Lastly, mutants that do not oscillate [23] can be used to avoid potential confounds due to synchronization effects. As our model couples S phosphorylation with A-loop burial (while still allowing independent conformational change), this predicts that mutants mimicking phosphorylated S-sites would correlate with hexamers having mostly buried A-loops with the highest frequency. Meanwhile, mutants mimicking unphosphorylated S-sites would correlate with hexamers having mostly exposed A-loops.

We thank the reviewer for their appreciation of our theoretical and computational contributions,

and remain excited about topological predictions to be verified in a biological system. Such verification has happened in many contexts in quantum systems [4–7], including in our own work, where purely theoretical predictions made [8, 9] were subsequently verified years later in several landmark experimental publications [7, 10]. Note that when the theoretical work was published, the experimental systems were not yet developed to test them, as it required years for the correct samples and conditions to be developed. While these previous successes have been in quantum and other platforms, topology in stochastic and biological systems is nascent. However, the powerful formalism of topology and previous examples show us that topological predictions on the theoretical and computational level have a lot to offer, and can have a huge payoff in terms of experimental consequences. It would be extremely exciting if topology can also be verified in biological systems, as it has been in so many other platforms, and our work provides the first concrete blueprint to do so. We have reworked the Introduction to include this context and background.

-
- [1] Jeroen S van Zon, David K Lubensky, Pim RH Altena, and Pieter Rein ten Wolde, “An allosteric model of circadian KaiC phosphorylation,” *Proceedings of the National Academy of Sciences* **104**, 7420–7425 (2007).
 - [2] Jenny Lin, Justin Chew, Udaysankar Chockanathan, and Michael J Rust, “Mixtures of opposing phosphorylations within hexamers precisely time feedback in the cyanobacterial circadian clock,” *Proceedings of the National Academy of Sciences* **111**, E3937–E3945 (2014).
 - [3] Joris Pajmans, David K Lubensky, and Pieter Rein Ten Wolde, “A thermodynamically consistent model of the post-translational Kai circadian clock,” *PLoS computational biology* **13**, e1005415 (2017).
 - [4] F Duncan M Haldane, “Model for a quantum hall effect without landau levels: Condensed-matter realization of the” parity anomaly”,” *Physical review letters* **61**, 2015 (1988).
 - [5] Chao-Xing Liu, Xiao-Liang Qi, Xi Dai, Zhong Fang, and Shou-Cheng Zhang, “Quantum anomalous hall effect in hg 1- y mn y te quantum wells,” *Physical review letters* **101**, 146802 (2008).
 - [6] Cui-Zu Chang, Jinsong Zhang, Xiao Feng, Jie Shen, Zuocheng Zhang, Minghua Guo, Kang Li, Yunbo Ou, Pang Wei, Li-Li Wang, *et al.*, “Experimental observation of the quantum anomalous hall effect in a magnetic topological insulator,” *Science* **340**, 167–170 (2013).
 - [7] Jinhai Mao, Slaviša P Milovanović, Miša Anelković, Xinyuan Lai, Yang Cao, Kenji Watanabe, Takashi Taniguchi, Lucian Covaci, Francois M Peeters, Andre K Geim, *et al.*, “Evidence of flat bands and correlated states in buckled graphene superlattices,” *Nature* **584**, 215–220 (2020).
 - [8] Evelyn Tang, Jia-Wei Mei, and Xiao-Gang Wen, “High-temperature fractional quantum hall states,” *Physical review letters* **106**, 236802 (2011).
 - [9] Evelyn Tang and Liang Fu, “Strain-induced partially flat band, helical snake states and interface superconductivity in topological crystalline insulators,” *Nature Physics* **10**, 964–969 (2014).
 - [10] Linda Ye, Mingu Kang, Junwei Liu, Felix Von Cube, Christina R Wicker, Takehito Suzuki, Chris Jozwiak, Aaron Bostwick, Eli Rotenberg, David C Bell, *et al.*, “Massive dirac fermions in a ferromagnetic kagome metal,” *Nature* **555**, 638–642 (2018).
 - [11] Taeko Nishiwaki and Takao Kondo, “Circadian autodephosphorylation of cyanobacterial clock protein KaiC occurs via formation of ATP as intermediate,” *Journal of Biological Chemistry* **287**, 18030–18035 (2012).
 - [12] Martin Egli, Tetsuya Mori, Rekha Pattanayek, Yao Xu, Ximing Qin, and Carl H Johnson, “Dephosphorylation of the core clock protein KaiC in the cyanobacterial KaiABC circadian oscillator proceeds via an ATP synthase mechanism,” *Biochemistry* **51**, 1547–1558 (2012).
 - [13] Taeko Nishiwaki-Ohkawa, Yohko Kitayama, Erika Ochiai, and Takao Kondo, “Exchange of adp with atp in the cii atpase domain promotes autophosphorylation of cyanobacterial clock protein kaic,” *Proceedings of the National Academy of Sciences* **111**, 4455–4460 (2014).
 - [14] Michael J Rust, Joseph S Markson, William S Lane, Daniel S Fisher, and Erin K O’Shea, “Ordered

- phosphorylation governs oscillation of a three-protein circadian clock,” *Science* **318**, 809–812 (2007).
- [15] Jun Abe, Takuya B Hiyama, Atsushi Mukaiyama, Seyoung Son, Toshifumi Mori, Shinji Saito, Masato Osako, Julie Wolanin, Eiki Yamashita, Takao Kondo, *et al.*, “Atomic-scale origins of slowness in the cyanobacterial circadian clock,” *Science* **349**, 312–316 (2015).
 - [16] Yao Xu, Tetsuya Mori, and Carl Hirschie Johnson, “Cyanobacterial circadian clockwork: roles of KaiA, KaiB and the kaiBC promoter in regulating KaiC,” *The EMBO Journal* **22**, 2117–2126 (2003).
 - [17] Christian Brettschneider, Rebecca J Rose, Stefanie Hertel, Ilka M Axmann, Albert JR Heck, and Markus Kollmann, “A sequestration feedback determines dynamics and temperature entrainment of the KaiABC circadian clock,” *Molecular Systems Biology* **6**, 389 (2010).
 - [18] Xu Han, Dongliang Zhang, Lu Hong, Daqi Yu, Zhaolong Wu, Tian Yang, Michael Rust, Yuhai Tu, and Qi Ouyang, “Determining subunit-subunit interaction from statistics of cryo-em images: observation of nearest-neighbor coupling in a circadian clock protein complex,” *Nature Communications* **14**, 5907 (2023).
 - [19] Hakuto Kageyama, Taeko Nishiwaki, Masato Nakajima, Hideo Iwasaki, Tokitaka Oyama, and Takao Kondo, “Cyanobacterial circadian pacemaker: Kai protein complex dynamics in the KaiC phosphorylation cycle in vitro,” *Molecular cell* **23**, 161–171 (2006).
 - [20] Tetsuya Mori, Shogo Sugiyama, Mark Byrne, Carl Hirschie Johnson, Takayuki Uchihashi, and Toshio Ando, “Revealing circadian mechanisms of integration and resilience by visualizing clock proteins working in real time,” *Nature communications* **9**, 1–13 (2018).
 - [21] Kazuki Terauchi, Yohko Kitayama, Taeko Nishiwaki, Kumiko Miwa, Yoriko Murayama, Tokitaka Oyama, and Takao Kondo, “ATPase activity of KaiC determines the basic timing for circadian clock of cyanobacteria,” *Proceedings of the National Academy of Sciences* **104**, 16377–16381 (2007).
 - [22] Jinhye Bae, Juanjuan Zheng, Haitao Zhang, Peter J Foster, Daniel J Needleman, and Joost J Vlassak, “A micromachined picocalorimeter sensor for liquid samples with application to chemical reactions and biochemistry,” *Advanced Science* **8**, 2003415 (2021).
 - [23] Taeko Nishiwaki, Yoshinori Satomi, Yohko Kitayama, Kazuki Terauchi, Reiko Kiyohara, Toshifumi Takao, and Takao Kondo, “A sequential program of dual phosphorylation of KaiC as a basis for circadian rhythm in cyanobacteria,” *The EMBO journal* **26**, 4029–4037 (2007).

REVIEWERS' COMMENTS

Reviewer #4 (Remarks to the Author):

The authors revised the manuscript by appropriately responding to the reviewer's previous comments. Though the assumption of the weak coupling limit of six subunits in the KaiC hexamer is hypothetical and not convincing, the proposal of this hypothesis is now described clearly in the revised manuscript. A minor comment is that the authors should refer to the paper of Furuike et al. (2022) Science Advances, <https://doi.org/10.1126/sciadv.abm8990>, describing the structural change of individual subunits in the KaiC hexamer upon S or T phosphorylation/dephosphorylation by observing various crystal structures, which should be relevant to the basic assumption of authors' hypothesis.

Response to the Referees for “A topological mechanism for robust and efficient global oscillations in biological networks”

Chongbin Zheng^{1,2}, Evelyn Tang^{1,2,*}

**1 Center for Theoretical Biological Physics
Rice University, Houston, Texas 77005, USA**

**2 Center for Theoretical Biological Physics
Rice University, Houston, Texas 77005, USA**

*** E-mail: Corresponding e.tang@rice.edu**

I. REVIEWER 1

Reviewer #4 (Remarks to the Author):

The authors revised the manuscript by appropriately responding to the reviewer's previous comments. Though the assumption of the weak coupling limit of six subunits in the KaiC hexamer is hypothetical and not convincing, the proposal of this hypothesis is now described clearly in the revised manuscript. A minor comment is that the authors should refer to the paper of Furuike et al. (2022) Science Advances, <https://doi.org/10.1126/sciadv.abm8990>, describing the structural change of individual subunits in the KaiC hexamer upon S or T phosphorylation/dephosphorylation by observing various crystal structures, which should be relevant to the basic assumption of authors' hypothesis.

We thank the reviewer for pointing out the reference, which supports our model assumption and highlights the importance of S phosphorylation/dephosphorylation in driving conformational change of KaiC. We have added a discussion of the reference in the third paragraph of Sec. III.